# The recovery of European freshwater biodiversity has come to a halt

Owing to a long history of anthropogenic pressures, freshwater ecosystems are among the most vulnerable to biodiversity loss[1]. Mitigation measures, including wastewater treatment and hydromorphological restoration, have aimed to improve environmental quality and foster the recovery of freshwater biodiversity[2]. Here, using 1,816 time series of freshwater invertebrate communities collected across 22 European countries between 1968 and 2020, we quantified temporal trends in taxonomic and functional diversity and their responses to environmental pressures and gradients. We observed overall increases in taxon richness (0.73% per year), functional richness (2.4% per year) and abundance (1.17% per year). However, these increases primarily occurred before the 2010s, and have since plateaued. Freshwater communities downstream of dams, urban areas and cropland were less likely to experience recovery. Communities at sites with faster rates of warming had fewer gains in taxon richness, functional richness and abundance. Although biodiversity gains in the 1990s and 2000s probably reflect the effectiveness of water-quality improvements and restoration projects, the decelerating trajectory in the 2010s suggests that the current measures offer diminishing returns. Given new and persistent pressures on freshwater ecosystems, including emerging pollutants, climate change and the spread of invasive species, we call for additional mitigation to revive the recovery of freshwater biodiversity.

Freshwater ecosystems are biodiversity hotspots and provide vital ecosystem services, including drinking water, food, energy and recreation. However, humans have degraded freshwaters for centuries, with impacts sharply increasing after World War II during the great acceleration[3]. Freshwaters are exposed to anthropogenic pressures from agricultural and urban land uses over whole catchments, accumulating pollutants, including phosphorus, organic-rich effluents, fine sediments, pesticides and emergent pollutants (such as nanoplastics and pharmaceuticals)[4,5]. Furthermore, freshwaters have been degraded by hydromorphological alterations, water extraction, invasive species and climate change[6,7]. In response to legislation such as the US Clean Water Act (1972) and the EU Water Framework Directive (2000), key countermeasures designed to improve water quality and restore freshwater habitats were implemented, including better wastewater treatment and controls on the emission of airborne pollutants. These actions resulted in considerable declines in organic pollution and acidification beginning around 1980[8]. Over the past 50 years, such mitigation measures have resulted in quantifiable improvements in freshwater biodiversity in some locations[9], yet the number and impacts of stressors threatening freshwater ecosystems continues to increase worldwide and the biological quality of rivers remains poor globally[10,11].

Freshwater invertebrates are a phylogenetically and ecologically diverse group that contribute to critical ecosystem processes, including decomposing organic matter, filtering water, providing energy to higher trophic levels, and transporting nutrients and energy between aquatic and terrestrial ecosystems[12,13]. Moreover, freshwater invertebrates have long been a cornerstone of water-quality monitoring. The biological traits of freshwater invertebrates are well characterized,

enabling the assessment of functional diversity—the range of functional traits of the organisms in a given ecosystem[14]—an important facet of biodiversity that can be used as a proxy for ecosystem functioning[15,16]. However, trajectories of taxonomic and functional diversity have rarely been investigated simultaneously at larger spatial and temporal scales. Determining the trajectories of taxonomic and functional change could inform the development of evidence-based management strategies that address stressors through mitigation, restoration and conservation. Furthermore, how temporal changes in biodiversity manifest across large spatial scales and vary among taxonomic groups remains equivocal[17–19]. Examining whole ecological groups representative of a particular ecosystem (for example, freshwater invertebrate communities in river ecosystems) may help to clarify discrepancies across studies and identify key drivers of temporal change.

Here we analysed pan-European patterns and drivers of multidecadal trends in abundance and taxonomic and functional diversity of invertebrate communities using a comprehensive dataset of 1,816 time series collected in riverine systems in 22 European countries between 1968 and 2020 (Fig. 1). The dataset comprises 714,698 observations of 2,648 taxa in 26,668 samples. The time series span a mean of 19.2 years with an average of 14.9 sampling years (minimum 8 years, maximum 32 years). We address two research questions: (1) how abundance, taxonomic diversity and functional diversity of freshwater invertebrate communities have changed over the past five decades in European streams and rivers; and (2) what environmental factors have driven these changes. Given that Europe-wide management has resulted in improvements in water quality[2,20], we hypothesize that abundance, taxonomic diversity and functional diversity have increased, consistent with a recovery.

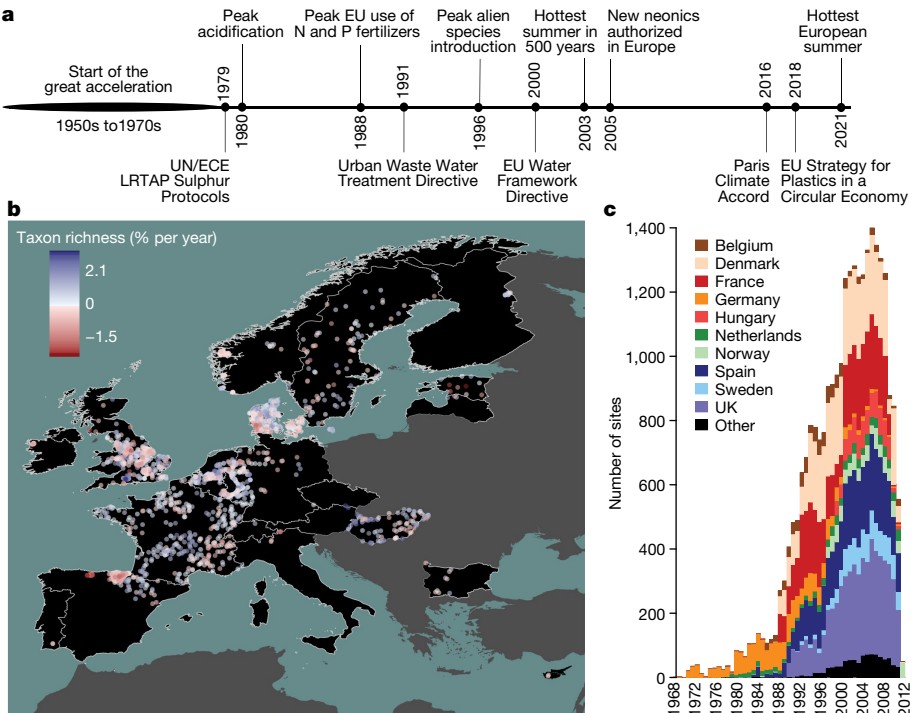

**Fig. 1 | Timeline and data distribution. a**, A timeline of major stressors (above the line) and environmental legislation (below the line) affecting Europe's freshwater ecosystems (citations are provided in Supplementary Table 1). UN/ECE LTRAP, United Nations Economic Commission for Europe Long-Range Transboundary Air Pollution. **b**, The sampling sites (points) and the rate of temporal change in taxon richness of freshwater invertebrate communities (colour of points) across 22 European countries (black). **c**, The distribution of sampling sites over time and countries. 'Other' includes countries with fewer than 50 sampling sites.

We further hypothesize that freshwater invertebrate community recovery was strongest around the end of the previous century after the onset of concerted efforts to mitigate stressor impacts and restore ecosystems, but has slowed in recent years owing to diminishing returns on these actions in addition to remaining and new pressures including climate change, land-use intensification and emerging pollutants. We assessed evidence for negative impacts of multiple human pressures, including dams, urban areas and cropland, and increasing temperatures, while accounting for subcatchment characteristics (such as elevation and stream size). We used hierarchical Bayesian models to estimate trends and identify drivers of change in abundance and taxonomic and functional diversity of Europe's freshwater invertebrate communities, while accounting for temporal autocorrelation, sampling date and sampling variation across studies and countries.

## Recovery of Europe's freshwater invertebrate communities

Across all time series, taxon richness increased by 0.73% per year, whereas abundance increased by 1.17% per year between 1968 and 2020 (Fig. 2a,b), substantiating previous documentation of a recovery process[18,21,22]. The probabilities of trends derived from posterior distributions (that is, the probability of the mean trend being above or below zero) revealed 0.99 and 0.91 probabilities of a mean increase in taxon richness and abundance, respectively. Despite these net-positive trends, taxon richness declined at 30% of sites and abundance declined at 39% of sites. Abundance trends for EPT taxa (mayflies, stoneflies and caddisflies—an indicator group of water quality[23]) and insects increased (EPT, +2.38% per year, 0.97 probability; insects, +1.53% per year, 0.95 probability) at higher net rates than the overall trends. EPT richness (+0.45% per year, 0.82 probability) and insect richness (+0.71% per year, 0.99 probability) trends increased, but at net rates lower than the overall trends (Extended Data Fig. 1).

Freshwater ecosystems are frequently invaded by non-native species[7]. We therefore examined whether changes in abundance and richness were driven by these taxa. Non-native species comprised an average of 4.9% of the species and 8.9% of the individuals at the 1,299 sites for which the taxonomic resolution allowed detection. Thus, native species dominated most communities (with 99.9% of sites comprising >50% native species). When considering only native taxa, trends in richness (+0.64% per year, 0.98 probability) and abundance (+0.26% per year, 0.61 probability) remained positive, but less so than overall net trends (Fig. 2 and Extended Data Fig. 1). For sites at which non-native species were detected (898 out of 1,299 sites), non-native species richness (+3.97% per year, 0.99 probability) and abundance (+3.9% per year, 0.95 probability) increased sharply (Extended Data Fig. 1).

Functional diversity, which describes the value and range of functional traits of the organisms in a given ecosystem[14] (Supplementary Table 4), also increased over the 53-year study period. Functional richness, which quantifies the functional space filled by a community, increased on average by 2.4% per year (0.99 probability of increase; Fig. 2e). Functional redundancy—a measure of overlap in functional trait space—had no strong trend (+0.03% per year, 0.64 probability of increase; Fig. 2f). By contrast, functional evenness declined (−0.22% per year, 0.96 probability of decrease; Fig. 2g), as did taxonomic evenness (−0.54% per year, 0.99 probability; Fig. 2c). Similarly, functional temporal turnover (−0.32% per year, 0.97 probability; Fig. 2h) and taxonomic temporal turnover declined (−0.2% per year, 0.87 probability; Fig. 2d). Together, these results suggest that functional diversity trends largely paralleled those of taxonomic diversity. Model estimates and raw distributions of trends for additional taxonomic and functional metrics are shown in Extended Data Fig. 2.

## Gains in species richness have come to a halt

While overall net trends provide an overview across the entire study period and enable comparison with other long-term biodiversity

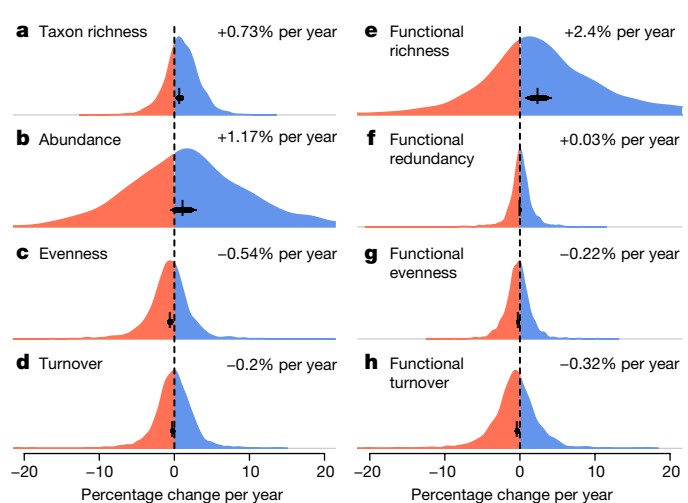

**Fig. 2 | Averages and distributions of trends in taxonomic and functional diversity metrics. a–h**, Overall meta-analysis estimates and distributions of site-level trends for taxonomic metrics of taxon richness (**a**), abundance (**b**), Shannon's evenness (**c**) and turnover (**d**), and functional metrics of richness (**e**), redundancy (**f**), evenness (**g**) and turnover (**h**) across all 1,816 sites. The black error bars and text on each panel show the mean estimates (percentage change per year). The error bars indicate the 80%, 90% and 95% CIs.

studies[17,19,24], they may mask important shorter-term temporal fluctuations in trends. Thus, to provide more nuanced, complementary trend information, we used a ten-year moving-window approach to examine the trajectories of freshwater invertebrate community change over time. Nonlinear trajectories were expected due to temporal variation in pressures and the implementation of mitigation measures[25]. To improve spatial representativity and comparability across years, only years with at least 250 sites from at least 8 countries were included, corresponding to the period of 1990 to 2020.

Although trends in taxon richness were generally positive, indicating increases in local richness through time, this effect became weaker over the decades (mean change in trends = −8.8% per year, 95% credible interval (CI) range: −13.6% to −3.8% per year). Trends in taxon richness started declining around 2010 and then levelled off, reaching an average of net zero around 2013 (Fig. 3a), indicating an end to the preceding recovery period. When considering only the dominant pattern as measured by the proportion of positive trends, the proportion of sites with increasing taxon richness declined after windows centred on the early 2000s (Fig. 3e). Functional richness trends were more variable, with the highest trends evident for windows centred on 2000 and 2010, and near net zero trends after 2010 (Fig. 3c). Functional richness trends had an overall tendency to decline (mean change in trends of functional richness = −5.9% per year, 95% CI range: −12% to +0.1% per year). Temporal changes in the proportion of sites with positive functional richness trends were similar to those reported for taxon richness (Fig. 3e,g). Trends in abundance (Fig. 3b,f) and functional redundancy (Fig. 3d,h) changed little over time (that is, CIs overlapped with zero in an analysis of the change in trend estimates over time), although abundance trends tended to decline from windows centred on 2010 until the end of the study period.

Although similar trends in taxonomic and functional metrics were expected due to functional variation being constrained by taxon richness, functional diversity can be more responsive to environmental gradients[26]. However, changes in functional diversity have rarely been quantified in large-scale investigations of temporal change in biodiversity[27,28]. A switch from primarily positive trends in functional richness in the late 1990s and early 2000s to near-zero trends starting around 2012 (Fig. 3c) may suggest no further improvements in

ecosystem functioning. The concurrent limited change in functional redundancy (Fig. 3d) indicates that the increase in functional richness provided new traits to these communities rather than adding traits that were already present. Both taxonomic and functional trends in evenness and turnover remained near zero or slightly negative over time (Extended Data Fig. 3).

## Environmental drivers of biodiversity change

Identifying the natural and anthropogenic drivers of biotic change is critical to inform effective management strategies. Here we show that climate, dam impacts, and the percentage of upstream urban areas and cropland (both sources of pollution and causes of habitat degradation) can all be linked to trends in taxonomic and functional metrics representing Europe's freshwater invertebrate communities (Fig. 4 and Extended Data Figs. 4 and 5).

Climate strongly influenced freshwater invertebrate communities (Fig. 4). Overall, sites experienced a net increase in air temperature of +0.037 °C per year ± 0.0007 s.e.m. (with 94% of sites warming) and a slight net increase in precipitation of +0.49 mm per year ± 0.12 s.e.m. (with 57% of sites getting wetter) over the studied intervals. Sites in areas with higher mean air temperatures were more likely to gain taxa (Fig. 4) compared with those in cooler areas. This may indicate that climate warming has not yet reached critical values for many European freshwater invertebrates, consistent with previous predictions for ectotherms in temperate regions[29,30]. Alternatively, lower recovery rates for biotic communities in cooler areas could reflect the less severe degradation of northern sites before recovery started. By contrast, more warming over time had negative biodiversity outcomes, with negative effects on long-term trends of taxon richness, abundance and functional richness (Fig. 4). Mean precipitation had a positive effect on long-term trends of functional richness but a negative effect on long-term trends of abundance and functional redundancy, indicating the addition of functionally unique taxa at wet sites. However, greater increases in precipitation over time had a negative effect on long-term trends of both taxonomic and functional richness (Fig. 4). Precipitation can influence invertebrate communities and their functioning by altering flow regimes (and therefore water quality and temperature through changes in runoff, discharge and dilution) and food availability[6].

Biodiversity trends were generally lower at sites downstream of dams and in catchments with a high percentage of urban areas or cropland. High dam impacts (that is, those in systems connected to more dams and/or closer to dams) had negative effects on long-term trends in taxon richness, abundance, functional richness and functional redundancy (Fig. 4). Dams increase sediment loads, reduce longitudinal connectivity, and change river flow and temperature regimes[31–33]. By contrast, high dam impacts had a positive effect on long-term trends of both taxonomic and functional evenness, suggesting that dominant species declined in abundance in communities downstream of dams, whereas richness losses were more pronounced for rare species. Furthermore, increases in functional evenness, accompanied by decreases in functional richness and redundancy, could reflect selection for a subset of traits that confer tolerance, of the conditions downstream of dams, including altered resource availability and hydromorphological homogenization. A greater percentage of upstream cropland had a negative effect on long-term trends in taxonomic and functional richness and abundance. Cropland frequently contributes to nutrient-enriched runoff, leaving primarily tolerant taxa[34]. A greater percentage of upstream urban areas had negative effects on taxon richness long-term trends (Fig. 4), but positive effects on non-native richness long-term trends (Extended Data Fig. 5a), suggesting losses of rare and sensitive native species. Biodiversity trends varied little with stream characteristics, although sites at higher elevations had lower gains in functional richness, potentially due to rising temperatures (as evidenced by a weak positive correlation between temperature trends

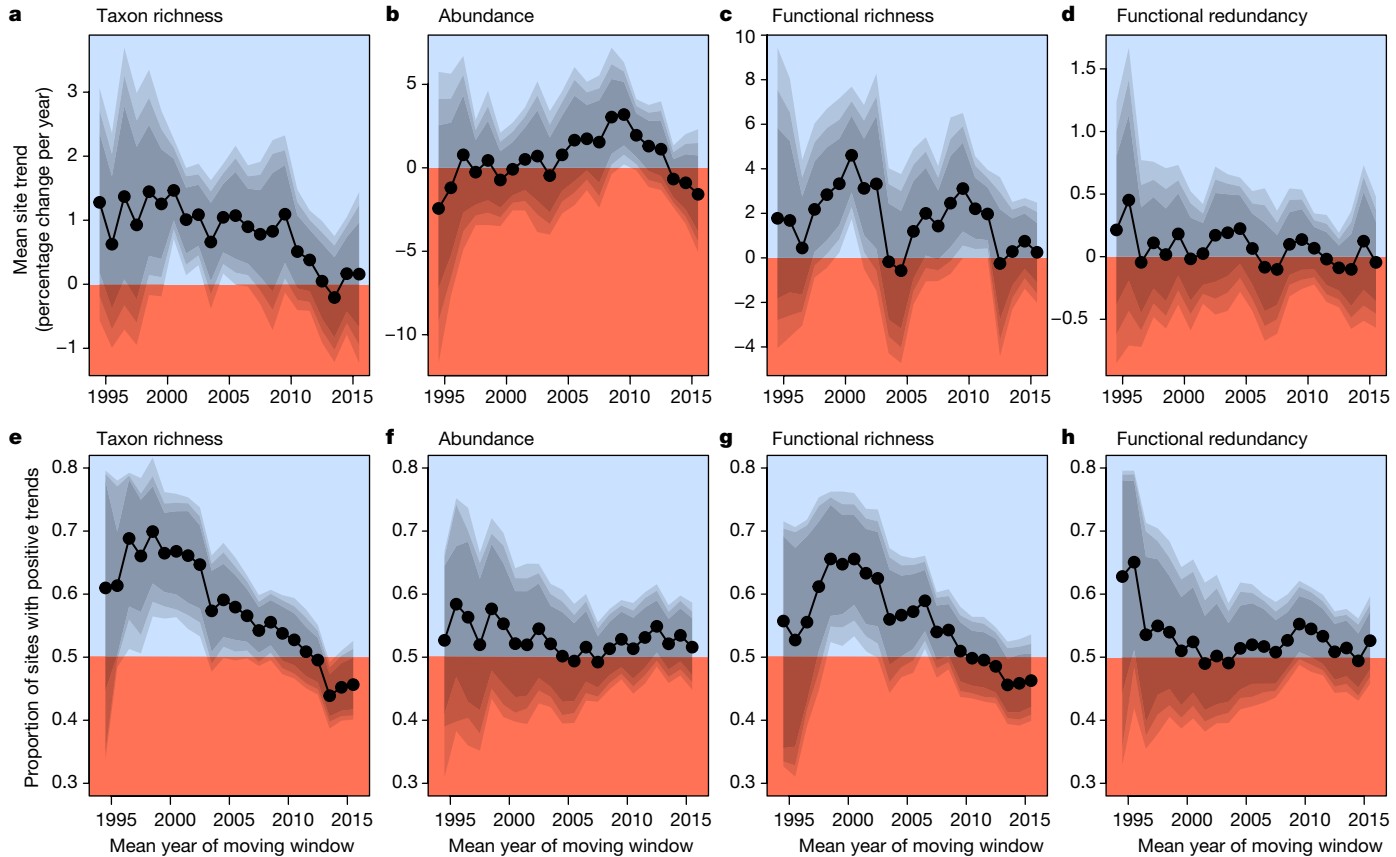

**Fig. 3 | Temporal fluctuations in trend estimates using a moving window.**
**a**–**h**, Modelled trend estimates from moving windows of taxon richness (**a**),
abundance (**b**), functional richness (**c**) and functional redundancy (**d**), and
the proportion of sites with positive trend estimates of taxon richness (**e**),
abundance (**f**), functional richness (**g**) and functional redundancy (**h**). Trend
estimates were calculated from Bayesian mixed-effects models of trends from
at least 250 time series with at least 6 years of data from at least 8 countries
within 10-year moving windows (totalling 21,495 time-series segments). The
proportions are based on whether site-level trend estimates of these
time-series were above zero or not. For trend estimates in **a**–**d**, blue and red
areas indicate the overall positive (>0) and negative (<0) mean trend estimates
for the given 10-year window, respectively, and the grey polygons indicate the
80%, 90% and 95% CIs. For site proportions in **e**–**h**, blue and red areas indicate a
larger proportion of positive (>50% of sites) and negative (<50% of sites)
site-level trend estimates for the given 10-year window, respectively, and the
grey polygons indicate 80%, 90% and 95% CIs.

and elevation; $r = 0.15$)[35]. Larger rivers became relatively more prone to
invasion by non-native species[36] (Extended Data Figs. 6–8).

## Reviving the recovery

Using a comprehensive Europe-wide dataset, we document the recovery of freshwater invertebrate communities over the past 53 years. The taxon richness gains observed across 70% (1,269 out of 1,816) of time series are concurrent with widespread implementation of mitigation measures[8], particularly improvements in wastewater treatment motivated by the EU Urban Waste Water Directive from 1991. However, gains in taxon richness started to decelerate around 2010, which may indicate that progress towards recovery has come to a halt at many sites, while remaining sites may reflect either predominant recovery or ongoing degradation towards the end of the study period. Most of our sites are monitored under the EU Water Framework Directive (WFD) and 60% of WFD-monitored rivers still do not reach 'good ecological status'[37]. Even at 'good' sites, considerable recovery could be needed to reach 'high ecological status', suggesting that improvements documented here represent only a partial recovery of European freshwater ecosystems.

Regardless of the reason for the deceleration, the impacts to Europe's rivers caused by ongoing pressures remain extensive and severe[37,38]. Although our observational data prevent confirmation of the underlying causal processes, our interpretation of the overall recovery being a response to improving water quality aligns with the conclusions of

other studies of European freshwater invertebrate time series[9,39]. Negative effects of poor water quality on biodiversity are supported by our findings that freshwater invertebrate communities downstream of dams, urban areas and cropland were less likely to experience biodiversity recovery. Urban areas produce the majority of micropollutants, are hubs of non-native species invasions (Extended Data Fig. 5a) and generate high-nutrient inputs, whereas croplands are sources of fine sediment[40], pesticides and nutrient-laden runoff[41], and greatly contribute to river salinization[42]. Most European rivers bear a substantial legacy of human impacts on their hydromorphology[8,38], with urban areas being the most affected, despite considerable river restoration in recent decades[43]. The positive effects of higher mean temperatures on long-term trends in invertebrate richness probably reflect the lower initial degradation in northern European countries. This may also reflect the relatively cool temperatures in European countries, whereas decreases in invertebrate richness are currently expected in freshwaters of warmer bioregions, such as tropical regions, which are not represented in our study[44]. However, the negative effects on long-term trends of taxon richness, abundance and functional richness in communities experiencing greater rates of warming are worrying. These effects are likely to worsen as temperatures continue to rise and as climatic extremes including summer droughts and heatwaves become more common[45].

Considering that environmental legislation and policy have insufficiently addressed ongoing and emerging stressors[8], the stalled recovery

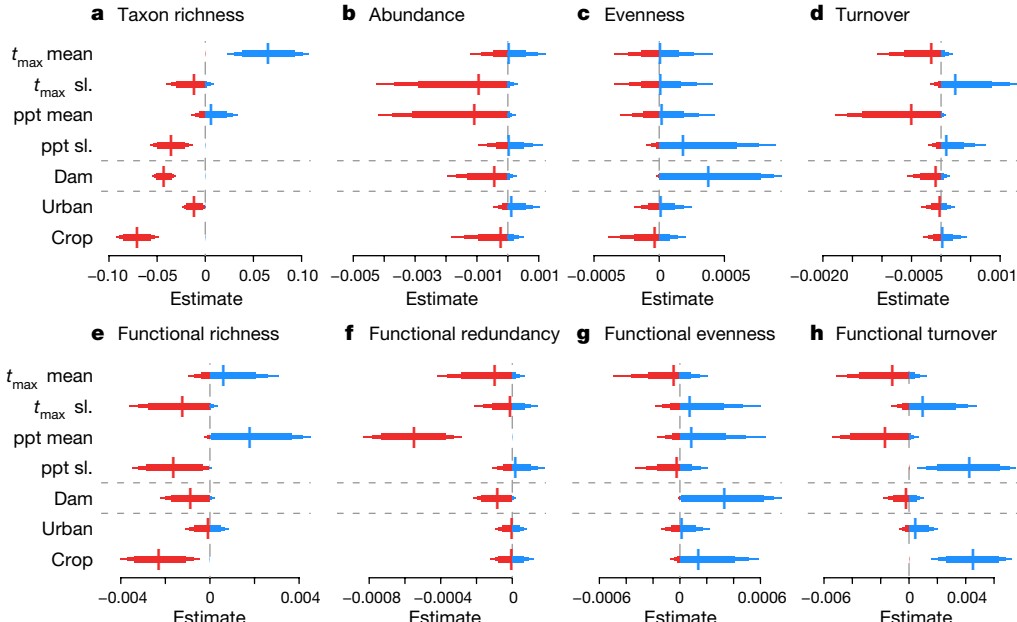

**Fig. 4 | Estimated effects of environmental drivers on biodiversity trends. a–h,** Estimated effects of the mean ($t_{max}$ mean) and trend ($t_{max}$ slope (sl.)) of annual maximum monthly mean temperatures, mean (ppt mean) and trend (ppt sl.) of the annual cumulative precipitation, the dam impact score (dam) and the percentage of the upstream catchment covered by urban areas and cropland on site-level long-term trend estimates for taxon richness (**a**), abundance (**b**), evenness (**c**) and turnover (**d**), and functional richness (**e**), redundancy (**f**), evenness (**g**) and turnover (**h**). $n$ = 1,816 biologically independent sites for all metrics. Positive and negative estimates are shown in blue and red, respectively. For climatic drivers, mean values refer to mean long-term values at each site and represent geographical variation; trends were calculated by regressing annual mean values against year, using the coefficient as an estimate of climatic trend and represent temporal variation. All response variables are site-level trends (that is, change in biodiversity metric over time) and all covariates were standardized to units of s.d. before analysis. A positive coefficient means that sites with higher values of the driver tended to have higher trends, although not necessarily positive trends, compared with sites with lower values of the driver. For example, trends in taxon richness were higher at sites with higher maximum mean temperatures ($t_{max}$ mean) but lower at sites with higher rates of temperature increase ($t_{max}$ sl.; **b**). The bars around the estimates indicate 80%, 90% and 95% CIs. The grey horizontal lines separate the three environmental driver groups: climate, dams and land use. Estimates of stream characteristics (stream order, flow accumulation, elevation and slope) are shown in Extended Data Fig. 6.

is unsurprising. Further management actions to revive the recovery should target sites at greater risk of biodiversity decline, such as those downstream of urban areas, cropland and dams, while maintaining and strengthening protection of the least impacted systems that are refuges of biodiversity. Specifically, substantial, catchment-scale changes in land management must go beyond current legislative requirements and achieve greater reductions in water extraction and inputs of pollutants including fine sediments, pesticides and fertilizers. Substantial investment is needed to upgrade sewage networks and improve wastewater treatment plants to better manage stormwater overflow and more effectively remove micropollutants, nutrients, salts and other contaminants[46]. Adopting a catchment-scale approach that considers barriers to dispersal[47] can further enhance the effectiveness of management, conservation and restoration practices[32,48]. Additional hydromorphological restoration efforts are required to reconnect rivers and floodplains to improve ecosystem functioning, prevent destructive floods, and adapt riverine systems to future climatic and hydrological regimes. Finally, standardized, large-scale and long-term biodiversity monitoring, paired with parallel environmental data collection[49,50], should be prioritized to effectively characterize temporal changes in biodiversity and environmental drivers and identify sites at high risk[51].

Current large-scale measures to address biodiversity loss remain rare, especially for invertebrates. This in part reflects our understanding of biodiversity change, which is limited by unknown historical baseline conditions and complex variation in interacting anthropogenic stressors. Insufficient baseline data present challenges both for characterization of biodiversity trends and ecological status of communities, and evaluation of tolerable levels and effects of stressors[52]. Data on the state

of freshwater communities both before and during the great acceleration are largely lacking, making it unclear when freshwater degradation peaked. Long-term data from the UK suggest freshwater invertebrate biodiversity was lowest at the start of the 1990s[53], but our pre-1990s data are insufficient to determine whether this pattern is Europe-wide (Fig. 1c). Moreover, comparison with unimpacted 'reference' communities, a standard practice in freshwater ecology, is becoming increasingly challenging due to the emergence of new communities[54] resulting from climate change, non-native species invasions and other pressures[55]. Progress towards biodiversity goals needs to recognize these changing pressures through flexible strategies to protect and foster Earth's remaining biodiversity. We call for adaptive environmental management that recognizes conservation and restoration objectives as shifting targets that can be modified to adapt to global change and maximize the protection of biodiversity.

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

Peter Haase[1,2,71] ✉, Diana E. Bowler[3,4,5], Nathan J. Baker[1,6], Núria Bonada[7], Sami Domisch[8], Jaime R. Garcia Marquez[8], Jani Heino[9], Daniel Hering[2], Sonja C. Jähnig[8,10], Astrid Schmidt-Kloiber[11], Rachel Stubbington[12], Florian Altermatt[13,14], Mario Álvarez-Cabria[15], Giuseppe Amatulli[16], David G. Angeler[17,18,19,20], Gaït Archambaud-Suard[21], Iñaki Arrate Jorrín[22], Thomas Aspin[23], Iker Azpiroz[24], Iñaki Bañares[25], José Barquín Ortiz[15], Christian L. Bodin[26], Luca Bonacina[27], Roberta Bottarin[28], Miguel Cañedo-Argüelles[7,29], Zoltán Csabai[30,31], Thibault Datry[32], Elvira de Eyto[33], Alain Dohet[34], Gerald Dörflinger[35], Emma Drohan[36], Knut A. Eikland[37], Judy England[38], Tor E. Eriksen[39], Vesela Evtimova[40], Maria J. Feio[41], Martial Ferréol[32], Mathieu Floury[8,42], Maxence Forcellini[32], Marie Anne Eurie Forio[43], Riccardo Fornaroli[27], Nikolai Friberg[39,44,45], Jean-François Fruget[46], Galia Georgieva[40], Peter Goethals[43], Manuel A. S. Graça[41], Wolfram Graf[11], Andy House[23], Kaisa-Leena Huttunen[47], Thomas C. Jensen[37], Richard K. Johnson[17], J. Iwan Jones[48], Jens Kiesel[8,49], Lenka Kuglerová[50], Aitor Larrañaga[51], Patrick Leitner[11], Lionel L'Hoste[34], Marie-Helène Lizée[21], Armin W. Lorenz[2], Anthony Maire[52], Jesús Alberto Manzanos Arnaiz[24], Brendan G. McKie[17], Andrés Millán[53], Don Monteith[54], Timo Muotka[47], John F. Murphy[48], Davis Ozolins[55], Riku Paavola[56], Petr Paril[31], Francisco J. Peñas[15], Francesca Pilotto[37], Marek Polášek[31], Jes Jessen Rasmussen[39], Manu Rubio[24], David Sánchez-Fernández[53], Leonard Sandin[37], Ralf B. Schäfer[57], Alberto Scotti[28,58], Longzhu Q. Shen[8,59], Stefan Stoll[2,60], Michal Straka[31,61], Henn Timm[62], Violeta G. Tyufekchieva[40], Iakovos Tziortzis[35], Yordan Uzunov[40], Gea H. van der Lee[63], Rudy Vannevel[43,64], Emilia Varadinova[40,65], Gábor Várbíró[66], Gaute Velle[26,67], Piet F. M. Verdonschot[63,68], Ralf C. M. Verdonschot[63], Yanka Vidinova[40], Peter Wiberg-Larsen[69] & Ellen A. R. Welti[1,70,71] ✉

[1]Department of River Ecology and Conservation, Senckenberg Research Institute and Natural History Museum Frankfurt, Gelnhausen, Germany. [2]Faculty of Biology, University of Duisburg-Essen, Essen, Germany. [3]Department of Ecosystem Services, German Centre for Integrative Biodiversity Research (iDiv) Halle-Jena-Leipzig, Leipzig, Germany. [4]Institute of Biodiversity, Friedrich Schiller University Jena, Jena, Germany. [5]Department of Ecosystem Services, Helmholtz Center for Environmental Research—UFZ, Leipzig, Germany. [6]Laboratory of Evolutionary Ecology of Hydrobionts, Nature Research Centre, Vilnius, Lithuania. [7]FEHM-Lab (Freshwater Ecology, Hydrology and Management), Department of Evolutionary Biology, Ecology and Environmental Sciences, Facultat de Biologia, Institut de Recerca de la Biodiversitat (IRBio), University of Barcelona, Barcelona, Spain. [8]Department of Community and Ecosystem Ecology, Leibniz Institute of Freshwater Ecology and Inland Fisheries (IGB), Berlin, Germany. [9]Geography Research Unit, University of Oulu, Oulu, Finland. [10]Geography Department, Humboldt-Universität zu Berlin, Berlin, Germany. [11]Department of Water, Atmosphere and Environment, Institute of Hydrobiology and Aquatic Ecosystem Management, University of Natural Resources and Life Sciences, Vienna, Austria. [12]School of Science and Technology, Nottingham Trent University, Nottingham, UK.

[13]Department of Evolutionary Biology and Environmental Studies, University of Zurich, Zurich, Switzerland. [14]Department of Aquatic Ecology, Eawag: Swiss Federal Institute of Aquatic Science and Technology, Dübendorf, Switzerland. [15]IHCantabria—Instituto de Hidráulica Ambiental de la Universidad de Cantabria, Santander, Spain. [16]School of the Environment, Yale University, New Haven, CT, USA. [17]Department of Aquatic Sciences and Assessment, Swedish University of Agricultural Sciences, Uppsala, Sweden. [18]IMPACT, The Institute for Mental and Physical Health and Clinical Translation, Deakin University, Geelong, Victoria, Australia. [19]Brain Capital Alliance, San Francisco, CA, USA. [20]School of Natural Resources, University of Nebraska-Lincoln, Lincoln, NE, USA. [21]INRAE, UMR RECOVER Aix Marseille Univ, Centre d'Aix-en-Provence, Aix-en-Provence, France. [22]Agencia Vasca del Agua, Vitoria-Gasteiz, Spain. [23]Wessex Water, Bath, UK. [24]Ekolur Asesoría Ambiental SLL, Oiartzun, Spain. [25]Departamento de Medio Ambiente y Obras Hidráulicas, Diputación Foral de Gipuzkoa, Donostia-San Sebastián, Spain. [26]LFI—The Laboratory for Freshwater Ecology and Inland Fisheries, NORCE Norwegian Research Centre, Bergen, Norway. [27]Department of Earth and Environmental Sciences—DISAT, University of Milano-Bicocca, Milan, Italy. [28]Institute for Alpine Environment, Eurac Research, Bolzano, Italy. [29]FEHM-Lab, Institute of Environmental Assessment and Water Research (IDAEA), CSIC, Barcelona, Spain. [30]Department of Hydrobiology, University of Pécs, Pécs, Hungary. [31]Department of Botany and Zoology, Faculty of Science, Masaryk University, Brno, Czech Republic. [32]INRAE, UR RiverLy, Centre de Lyon-Villeurbanne, Villeurbanne, France. [33]Fisheries Ecosystems Advisory Services, Marine Institute, Newport, Ireland. [34]Environmental Research and Innovation Department, Luxembourg Institute of Science and Technology, Esch-sur-Alzette, Luxembourg. [35]Water Development Department, Ministry of Agriculture, Rural Development and Environment, Nicosia, Cyprus. [36]Centre for Freshwater and Environmental Studies, Dundalk Institute of Technology, Dundalk, Ireland. [37]Norwegian Institute for Nature Research (NINA), Oslo, Norway. [38]Environment Agency, Wallingford, UK. [39]Norwegian Institute for Water Research, Oslo, Norway. [40]Department of Aquatic Ecosystems, Institute of Biodiversity and Ecosystem Research, Bulgarian Academy of Sciences, Sofia, Bulgaria. [41]Department of Life Sciences, University of Coimbra, Marine and Environmental Sciences Centre, ARNET, Coimbra, Portugal. [42]Univ Lyon, Université Claude Bernard Lyon 1, CNRS, ENTPE, UMR 5023 LEHNA, Villeurbanne, France. [43]Department of Animal Sciences and Aquatic Ecology, Ghent University, Ghent, Belgium. [44]Freshwater Biological Section, University of Copenhagen, Copenhagen, Denmark. [45]water@leeds, School of Geography, University of Leeds, Leeds, UK. [46]ARALEP—Ecologie des Eaux Douces, Villeurbanne, France. [47]Department of Ecology and Genetics, University of Oulu, Oulu, Finland. [48]School of Biological and Behavioural Sciences, Queen Mary University of London, London, UK. [49]Department of Hydrology and Water Resources Management, Christian-Albrechts-University Kiel, Institute for Natural Resource Conservation, Kiel, Germany. [50]Department of Forest Ecology and Management, Swedish University of Agricultural Sciences, Umeå, Sweden. [51]Department of Plant Biology and Ecology, University of the Basque Country, Leioa, Spain. [52]Laboratoire National d'Hydraulique et Environnement, EDF Recherche et Développement, Chatou, France. [53]Department of Ecology and Hydrology, University of Murcia, Murcia, Spain. [54]UK Centre for Ecology & Hydrology, Lancaster Environment Centre, Lancaster, UK. [55]Institute of Biology, University of Latvia, Riga, Latvia. [56]Oulanka Research Station, University of Oulu Infrastructure Platform, Kuusamo, Finland. [57]Institute for Environmental Science, RPTU Kaiserslautern-Landau, Landau, Germany. [58]APEM, Stockport, UK. [59]Institute for Green Science, Carnegie Mellon University, Pittsburgh, PA, USA. [60]Department of Environmental Planning / Environmental Technology, University of Applied Sciences Trier, Birkenfeld, Germany. [61]T.G. Masaryk Water Research Institute, Brno, Czech Republic. [62]Chair of Hydrobiology and Fishery, Centre for Limnology, Estonian University of Life Sciences, Elva vald, Estonia. [63]Wageningen Environmental Research, Wageningen University and Research, Wageningen, The Netherlands. [64]Flanders Environment Agency, Aalst, Belgium. [65]Department of Geography, Ecology and Environment Protection, Faculty of Mathematics and Natural Sciences, South-West University 'Neofit Rilski', Blagoevgrad, Bulgaria. [66]Department of Tisza River Research, Centre for Ecological Research, Institute of Aquatic Ecology, Debrecen, Hungary. [67]Department of Biological Sciences, University of Bergen, Bergen, Norway. [68]Institute for Biodiversity and Ecosystem Dynamics, University of Amsterdam, Amsterdam, The Netherlands. [69]Department of Ecoscience, Aarhus University, Aarhus, Denmark. [70]Conservation Ecology Center, Smithsonian National Zoo and Conservation Biology Institute, Front Royal, VA, USA. [71]These authors contributed equally: Peter Haase, Ellen A. R. Welti. ✉e-mail: peter.haase@senckenberg.de; mischiefmao@gmail.com

## Methods

### Time series

We assembled a database of time series of riverine invertebrate communities following a data call targeting European ecologists and environmental managers. We included only time series that (1) included abundance estimates; (2) documented whole freshwater invertebrate communities (including all sampled macroinvertebrates, for example, Coleoptera, Crustacea, Diptera, Ephemeroptera, Hirudinea, Mollusca, Odonata, Oligochaeta, Plecoptera, Trichoptera, Tricladida); (3) identified most taxa to family, genus or species; (4) had ≥8 sampling years (not necessarily consecutive); (5) used the same sampling method and taxonomic resolution throughout the sampling period; and (6) had consistent sampling effort per site (for example, the number of samples or area sampled) in all years.

Only one sampling event per year was included for each time series, where a sampling event was defined as the sample or samples collected within a single day. For time series with multiple sampling seasons within or among years, we included only one sampling season (defined as three consecutive months), preferentially using the season with the longest time series. No time series had multiple sampling events per season. Sensitivity analyses indicated limited effects of season on trend estimates (Extended Data Fig. 10). We removed taxa that are not freshwater invertebrates, including terrestrial and semi-aquatic taxa, and vertebrates, in addition to freshwater invertebrates that were recorded inconsistently owing to their small size (such as mites, copepods and cladocerans).

Between 13 and 516 taxa were sampled per site across all sampling years. Communities from 42% of sites were identified to species, 30% were identified to mixed (species-to-family) taxonomic levels and 28% were identified primarily to family. In total, 2,648 taxa from 959 genera, 212 families and 47 groups (primarily orders) were recorded. We list time-series locations, durations and characteristics in Supplementary Table 2 and list the number of sites sampled per year and country in Supplementary Table 3.

Our compiled time series represent different stream types and stream orders from a large geographical area of Europe. Data were collected for purposes including research projects and regulatory biomonitoring, although detailed information on the purpose is unavailable for some time series. These data were not selected randomly but were collected from available studies that met our six criteria. As these data were collected from sites exposed to varying and unquantified levels of anthropogenic impacts, we cannot rule out biases arising from unequal representation of sites exposed to different impact levels from severely impacted to least impacted.

### Community metrics

We calculated taxonomic and functional diversity metrics representing freshwater invertebrate communities across sites and over time. We also examined different community subsets: native and non-native species, and insects and EPT taxa (Ephemeroptera, Plecoptera, Trichoptera, that is, mayflies, stoneflies, caddisflies, grouped as an indicator of water quality[56]).

**Taxonomic diversity.** We calculated total abundance, taxon richness, Shannon's diversity, Shannon's evenness, rarefied richness (calculated on the basis of standardized numbers of individuals) and temporal turnover for each site and year. As sampling effort was standardized within time series before metric calculation, individual-based rarefied richness was used to estimate the number of taxa per given number of individuals, based on the lowest number of individuals per sampling year in each time series[17]. We calculated temporal turnover as the ratio of taxa gained or lost to the total number of taxa present between two timepoints using the R package codyn[57]. All other taxonomic metrics were calculated using the R package vegan[58].

**Functional diversity.** Traits were extracted from the European databases freshwaterecology.info (v.7.0)[59] and DISPERSE[60]. First, we downloaded trait data for all taxa. We considered biological traits that influence both a taxon's response to and its effects on its environment[61,62]. Specifically, we compiled data on 10 biological traits (with 53 trait modalities): respiration type, resistance form, dispersal type, aquatic stage, life cycle duration, reproduction type, maximum potential body size, wing form, propensity to drift and feeding type[60,63]. For taxa with multiple aquatic life stages (primarily beetles), whenever available from the trait databases, functional roles were assigned for each life stage, otherwise adult traits were used. We included only traits for which information was available for >85% of all taxa. All traits were fuzzy coded across multiple modalities depending on the information available; for example, the trait 'maximum potential body size' contains seven modalities ranging from ≤0.25 cm to >8 cm. Within each trait, we scaled affinities to different component modalities between 0 and 1 (summing to 1 across modalities for each taxon), so that each taxon was assigned an affinity score for each modality[64], to recognize potential trait plasticity.

We took the following steps to fill in gaps due to missing trait data. First, when trait data were not available at the original identification level (15.9% trait coverage across taxa), we used genus-level trait data, resulting in 48.2% coverage. Genus-level trait data are generally sufficient to represent most interspecific variation among freshwater invertebrates and thus taxon responses to environmental variability[61]. Next, when genus-level trait data were not available for taxa identified to genus, we replaced missing values in trait modalities with the median of trait profiles of all species within a genus from the full taxon list, resulting in 61.3% coverage. For taxa identified to family level with no available data for a given trait, we replaced missing values in trait modalities with the median value of trait profiles of all genera within a family, resulting in 90.5% coverage across all taxa. The lack of accurate phylogenies for many invertebrate taxa, low trait coverage at the species level and mixed taxonomic resolution across sampling sites prevented the use of other gap-filling approaches, but taxonomic aggregation generally aligns well with expert trait assignments[65].

We analysed functional diversity separately for each site by calculating six distance-based metrics chosen to describe multiple facets of community niche space and to align with taxonomic diversity metrics: functional richness, functional redundancy, functional evenness, functional turnover, functional divergence and Rao's quadratic entropy (definitions and citations are provided in Supplementary Table 4). All functional metrics except for functional redundancy and turnover were calculated using the dbFD function in the R package FD[66]. In calculations of functional richness and divergence, we used six principal coordinate analysis axes (the dbFD 'm' argument), according to current recommendations[67]. To enable calculation of functional turnover, we calculated community-weighted means of each functional trait category weighted by taxa abundance, then calculated turnover of the community-weighted means using the R package codyn, as for taxonomic temporal turnover[57,68]. We calculated abundance-weighted functional redundancy using the uniqueness function in the R package adiv[69]. We calculated redundancy according to a previous report[70]: community uniqueness ($U$) was calculated as quadratic diversity divided by Simpson diversity and functional redundancy was calculated as $1 - U$. The trait input matrix was based on Euclidean distances bound between 0 and 1 and the tolerance threshold was $10^{-8}$.

**Non-native species.** Non-native species were defined as introduced species (that is, those present due to human activities, not natural range expansion) at the country level (for example, a species native to Bulgaria could be non-native in the UK). To identify non-native species, we used two databases: DAISIE[71] and the Global Alien First Record Database

(GAFRD) (v.2)[72,73]. DAISIE contains non-native species in addition to native species defined as invasive because they cause economic loss (that is, pest species). GAFRD includes only non-native species but is limited to species and countries for which the approximate year of introduction is known. From each database, we first extracted all species listed for each European country in our dataset. We determined each species' country of origin using the Global Biodiversity Information Facility[74] or peer-reviewed publications, both to eliminate native species listed in DAISIE and to check whether species listed as non-native in one European country were also non-native elsewhere (for example, a North American species marked as non-native in Germany in GAFRD would be non-native in all European countries in which it occurred).

In total, we identified 61 non-native species. The initial analysis of native and non-native species was restricted to the 1,299 sites at which taxa were identified to species or a mixed taxonomic resolution; we excluded the remaining 517 sites due to the coarse (primarily family level) taxonomic resolution, which does not allow for reliable identification of non-native species. Estimates of trends in non-native species richness and abundance were restricted to the 898 (of 1,299) sites at which non-native species were recorded. The two most abundant non-native species were the New Zealand mud snail, *Potamopyrgus antipodarum* (≥1 individual present in ≥1 year at 81% of sites) and the North American bladder snail, *Physella acuta* (34% of sites).

### Stream characteristics and environmental predictors

**Stream network.** We used the MERIT Hydro[75] digital elevation model (DEM) to delineate the high-resolution Hydrography90m stream network[76]. To achieve a high spatial accuracy, we used an upstream contributing area of 0.05 km² as the stream channel initialization threshold using the r.watershed and r.stream.extract modules in GRASS GIS[77]. We next calculated the subcatchments for each segment of the stream network, that is, the area contributing laterally to a given stream reach between two nodes, using the r.basins module. Coordinates indicating a site's location did not always occur in the delineated stream network due to spatial inaccuracy of either the DEM or the coordinates. To ensure that point occurrences matched the DEM-derived stream network and therefore the network topology, we first identified the subcatchment in which each point occurrence was located, then moved all points to the corresponding stream segments using the v.net module within the given subcatchment. From each point, we calculated the network (as the fish swims) distance (km) using the v.net.distance module, and the Euclidean (as the crow flies) distance to all other point occurrences using the v.distance module. The distance was set to NA when sites were located in different drainage basins, and therefore not connected through the network.

**Environmental predictors.** We calculated stream topographical and topological predictors using the MERIT Hydro DEM[76]. Using the r.univar module in GRASS GIS, we computed the average elevation (m), elevation difference between the site and the upstream subcatchment (m), slope and the upstream contributing area (or flow accumulation, km²) for each subcatchment. To create a proxy for dam impacts, we calculated the network distance between each site and each upstream dam using the Global Reservoir and Dam Database (v.1.3)[78]. For dam impact score calculations, see Supplementary equation (1).

We extracted monthly climatic predictors from the TerraClimate dataset[79] for 1967–2020, which covered all sites and years. For each site, we identified the sampling month and computed the mean monthly climatic value for the corresponding subcatchment. We calculated climatic predictors of cumulative annual precipitation (mm) and maximum monthly temperature (°C) for each 12 month period preceding the mean sampling month at each site. Trend values in precipitation and maximum temperature over the period covered by each

time series were calculated using Bayesian models fitted using the R package brms[80]. These models were similar to those used to calculate site-level biodiversity metric trends, in which a trend was estimated as the coefficient of a continuous year effect. The TerraClimate dataset is associated with uncertainties in areas of complex terrain, but our large number of sampling sites, relatively good station coverage and the low physiographical complexity of most site locations should have minimized error in our analyses.

We calculated the proportion of land cover categories in each subcatchment using the ESA CCI Land Cover time series[81] for each year from 1992 to 2018. Land cover data were available for 92% of analysed site and year combinations and for 99% of sites. We computed the entire upstream catchment for each point occurrence using the r.water.outlet module and calculated the percentage cover of each land cover category within this area. The areas of cropland and urban land were calculated as the percentage of the upstream area averaged across the sampled years at each site.

A list of the stream characteristics and environmental drivers, their units and sources is provided in Supplementary Table 5.

### Statistical analysis

**Trend analysis.** Temporal trends in each taxonomic (abundance, richness, Shannon's diversity, Shannon's evenness, individual-based rarefied richness and temporal turnover), functional (redundancy, richness, evenness, turnover, divergence and Rao's quadratic entropy) and community subset (taxon richness and abundance of native species, non-native species, EPT taxa and insects only) metric were assessed using a two-step approach. First, we calculated site-level trends for each metric using Bayesian linear models fitted using the R package brms[80]. In these models, a biodiversity metric was the response variable and year was the continuous predictor variable of which the coefficient represented the temporal trend estimate.

The form of the model was: bf(BiodiversityMetric ~ cYear + ar(time = iYear, $p$ = 1, cov = TRUE)).

Fixed-year variables were centred to improve model convergence (cYear) and year in the temporal autocorrelation term was included as a count with the first year of sampling considered year 1 (iYear). The models accounted for any residual temporal autocorrelation using an ar(1) term[82] and included day of year as an additional predictor when variation in sampling dates at a site was >30 days.

The form of the model was: bf(BiodiversityMetric ~ cday_of_year + cYear + ar(time = iYear, $p$ = 1, cov = TRUE)).

The models assumed normally distributed errors, which were checked visually using histograms. Taxonomic evenness, functional richness, total abundance and subset abundance (non-native, native, EPT and insect abundance) were $\log_{10}$-transformed, and functional divergence was squared to meet the normality assumption.

We ran linear mixed-effects models (LMM) in the brms package to synthesize site-level data and estimate overall mean trends. The LMM included site-level trend estimates as the response, and an overall intercept and two random effects (country and study identity) as predictors. These random effects accounted for data heterogeneity due to unequal numbers of sites among studies and countries. Site-level trends were normally distributed; we therefore assumed normal errors. Site-level trends were combined in a meta-analysis model to estimate the mean trend across studies, including the uncertainty (represented by the s.d.) of the trend estimates, using brms[80].

The form of the model was: brm(estimate|se(sd_trend_estimate) ~ 1 + (1|study_id) + (1|country), data = response_stan, iter = 5000, inits = 0, chains = 4, prior = c(set_prior("normal(0,3)", class = "Intercept")), control = list(adapt_delta = 0.90, max_treedepth = 12)).

For each response metric, we calculated the proportion of the posterior distribution of the mean trend estimate (that is, the overall LMM intercept) above or below zero, that is, the probability of an increasing or decreasing mean trend.

In Bayesian models, we mostly used default brms settings, including four chains, which were run for 5,000 iterations (50% burn-in). We used default priors except for trend estimates, for which we selected a narrower prior to diminish the influence of biologically unrealistic trend estimates. Specifically, we used normally distributed priors with a mean of zero and an s.d. of 10 (for site-level trends) or 3 (for mean site-level trends). We compared our meta-analysis model of trends with and without including the uncertainty of site-level trend estimates. To optimize model fit, unweighted models were used for non-native and EPT abundance, and for EPT taxon richness. Functional turnover was fitted using beta models as values were bound between 0 and 1. The percentage change per year was calculated by back-transforming model estimates. Back-transformation calculations varied according to the originally modelled transformations of response variables (see the 'equationsToPercChangePerYr.xlsx' file in the 'plots/Fig2_DensityPlots' folder at https://github.com/Ewelti/EuroAquaticMacroInverts). We further tested a one-stage synthesis approach in which mean trends were estimated in one large mixed-effect model of the observed data, including random intercepts and slopes. Overall, these models produced similar trend results (see figure 16 in the 'Online Figures.docx' file in the 'plots' folder at https://github.com/Ewelti/EuroAquatic-MacroInverts).

**Moving-window analysis.** To assess how estimates of trends in abundance and taxonomic and functional diversity changed over time, we used a moving-window approach. We used a similar two-stage process as described above. For each year of the analysis, we calculated trends within a ten-year window in which all time series with ≥6 sampling years and from ≥8 countries were included. A ten-year window was chosen according to current recommendations regarding times-series length[83,84] and six was chosen as the number of sampling years covering >50% of each ten-year period. This analysis was restricted to the period between a first moving window from 1990 to 1999, in which any time series with ≥6 sampling years was included, to a final window from 2011 to 2020. After estimating site-level trends centred on each year of the moving window, we ran a Bayesian LMM for each year to estimate the overall mean trends across sites in that time period. These models followed the same form as used to calculate trend estimates, containing the predictor variables of trends including an error term to account for uncertainty, an overall intercept, and study identity and country as random effects (see the equation in the 'Trend analysis' section).

To test for an overall linear change in the trajectory of moving-window trends, we modelled the effect of year on moving-window trend estimates using brms[80].

The form of the model was: brm(MovingWindowTrend|se(sd_trend_estimate) ~ year, data = moving_window_trends, iter = 5000, inits = 0, chains = 4, prior = c(set_prior("normal(0,3)", class = "Intercept")), control = list(adapt_delta = 0.90, max_treedepth = 12)).

These models identified a linear decline in trends in taxon richness and a tendency for decline in functional richness trends over time (see figure 21 in the 'Online Figures.docx' file in the 'plots' folder at https://github.com/Ewelti/EuroAquaticMacroInverts).

We examined the proportion of sites with positive trends and how this proportion changed through time for our key biodiversity metrics of taxon richness, abundance, functional richness and functional redundancy. To do this, we used site-level moving-window trends and estimated the proportion of sites with positive trends in each year. We repeated this calculation for each posterior draw to propagate through site-level uncertainty to the overall mean proportion and estimated 80%, 90% and 95% CIs. To ensure this proportion was not driven by studies with especially large numbers of sampling sites, we weighted each site by the inverse of the number of sites in each study. This complements the moving-window analysis by examining whether the emerging mean trends are typical of site-level patterns. This analysis

was based only on trend direction and not trend magnitude and was therefore less affected by any noise contributed by studies with trends at the extremes.

An important caveat of the moving-window analysis is that different sites are included in different moving windows. Supplementary Table 6 lists the number of sites per window in each country. Although we accounted for the heterogeneity of site distribution across studies and countries within years, models cannot correct for the changing number of sampled sites across years. We cannot fully discount the possibility that biases in the characteristics of sites sampled across time affected trajectory results. We therefore conducted two additional moving-window analyses to investigate this, the first limited to sites with long-term data and the second limited to sites with species-level taxonomic resolution. The first additional analysis initially included only sites with ≥20 sampling years between 1990–2020, although moving windows with start years of 1990 and 1991 were excluded as they included <200 sites. This analysis included 308 sites from 8 countries. The second analysis included sites with species-level taxonomic data and windows covering 1990–2020 with >200 sites, resulting in windows from 1994–2003 to 2011–2020. The species-level moving-window analysis included 717 sites from 14 countries. Apart from the sites included, models were identical to our original moving-window analyses described above. These alternative moving-window analyses found similar declines in the trend of taxon richness over time (see figures 22–25 in the 'Online Figures.docx' file in the 'plots' folder at https://github.com/Ewelti/EuroAquaticMacroInverts).

**Analysis of environmental predictors.** We assessed responses of biodiversity metrics to climate (both the mean and the trend over the time series' durations) and upstream land cover (as the annual mean cover value during the sampling period), dam impact score and subcatchment characteristics (Supplementary Table 5). We did not include upstream land-use trends as most sites exhibited low variation: cropland cover changed by a mean of −0.002% per year ± 0.11 s.e.m., with no change detected at 634 sites; urban cover changed by 2.48% per year ± 0.14 s.e.m., but with no change detected at 803 sites. To examine relationships between environmental drivers and biodiversity trends, we modelled trend estimates using an LMM, incorporating trend errors as for the calculation of the overall trend, including all predictor variables as fixed effects, and study identity and country as random effects.

The form of the model was: brm(estimate|se(sd) ~ Precip-Trend + TempTrend + PrecipMean + TempMean + StreamOrder + Accumulation + Elevation + Slope + Urban + Crop + DamScore + (1|study_id) + (1|country), = response_stan, iter = 5000, chains = 4, prior = prior1, control = list(adapt_delta = 0.90, max_treedepth = 12)).

We ran models using the R package brms[80]. We standardized predictor variables to unit s.d. to facilitate comparison of their relative importance. We used regularizing horseshoe priors on environmental covariates that pull unimportant covariate effects towards zero to avoid overfitting. Our analysis of drivers focused on site-level variation in long-term trends, and not temporal variation in short-term trends examined in the moving-window analysis. Thus, our driver analysis cannot be used to understand recent changes in trends. To further examine whether biodiversity trends were positive or negative across the range of driver values, we used R package marginaleffects[85] to visualize responses to drivers while holding other driver covariates at their median. Predicted trends complement the effects on trends shown in Fig. 4 (see figures 28–34 in the 'Online Figures.docx' file in the 'plots' folder at https://github.com/Ewelti/EuroAquaticMacroInverts).

**Model checking.** All models run to quantify biodiversity trends and responses to drivers were evaluated by plotting the posterior samples to confirm chain convergence, examining $R$-hat values (<1.1)[86] and estimating Pareto shape parameters using the argument pareto_k_table

in the R package loo[87]. For trend models and across the 20 examined biodiversity metrics, an average of 99.5% of the 1,816 sites had shape parameter estimates of $k < 0.7$ (a threshold for good model performance). For environmental driver models, an average of 99% of the 1,816 sites had shape parameter estimates of $k < 0.7$.

**Sensitivity analysis.** To check the robustness of our results to analytical decisions, we ran multiple sensitivity analyses for all biodiversity metrics. We tested the effects on trend estimates of (1) taxonomic resolution, by rerunning meta-analysis models with resolution (family, mixed, and species) as an additional fixed factor; (2) sampling season, by rerunning meta-analysis models (described in the 'Trend analysis' section) with season (winter, spring, summer and fall) as an additional fixed factor; and (3) country, using a jackknife resampling analysis in which the meta-analysis was rerun after sequentially removing countries. Models were otherwise similar to those presented above. Scripts for sensitivity analyses are available at GitHub (https://github.com/Ewelti/EuroAquaticMacroInverts (HPC_Sensitivity_analysis.R and HPC_Meta_analysis_country_jackknife).

Some caution is advised when inferring conclusions from a dataset including different levels of taxonomic resolution or different seasons. However, intra-site sampling was consistently within one season or taxonomic resolution, so intra-site trends were not affected by these differences. Neither taxonomic resolution nor season had strong directional effects on trend estimates, with error bars generally overlapping. Patterns across taxonomic resolutions and sampling seasons were generally similar to those presented in Fig. 2 (Extended Data Figs. 9 and 10). Trends of taxonomic richness were robust to one-country removal but abundance trends became more strongly positive on removal of data from some countries, suggesting geographical variability in abundance trends (see figure 17 in the 'Online Figures.docx' file in the 'plots' folder at https://github.com/Ewelti/EuroAquaticMacroInverts).

We analysed the effect of the number of sampling years in a time series on observed trends using simple linear regression. The number of sampling years did not affect trend estimates of taxon richness ($R^2 < 0.001$), abundance ($R^2 < 0.001$), functional richness ($R^2 = 0.004$) or functional redundancy ($R^2 < 0.001$) (see figure 14 in the 'Online Figures.docx' file in the 'plots' folder at https://github.com/Ewelti/EuroAquaticMacroInverts).

### Reporting summary

Further information on research design is available in the Nature Portfolio Reporting Summary linked to this article.

## Data availability

All data needed to reproduce analyses including metadata, site characteristics and values of each metric (for example, species richness, functional richness) for each site and year are available at Figshare (https://doi.org/10.6084/m9.figshare.22227841). Biodiversity composition data are available at GitHub (https://github.com/Ewelti/EuroAquaticMacroInverts/raw-data).

## Code availability

Annotated R code is available at GitHub (https://github.com/Ewelti/EuroAquaticMacroInverts).

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

**Acknowledgements** N. Kaffenberger helped with initial data compilation. Funding for authors and data collection and processing was provided by the EU Horizon 2020 project eLTER PLUS (grant agreement no. 871128); the German Federal Ministry of Education and Research (BMBF; 033W034A); the German Research Foundation (DFG FZT 118, 202548816); Czech Republic project no. P505-20-17305S; the Leibniz Competition (J45/2018, P74/2018); the Spanish Ministerio de Economía, Industria y Competitividad—Agencia Estatal de Investigación and the European Regional Development Fund (MECODISPER project CTM 2017-89295-P); Ramón y Cajal contracts and the project funded by the Spanish Ministry of Science and Innovation (RYC2019-027446-I, RYC2020-029829-I, PID2020-115830GB-100); the Danish Environment Agency; the Norwegian Environment Agency; SOMINCOR—Lundin mining & FCT—Fundação para a Ciência e Tecnologia, Portugal; the Swedish University of Agricultural Sciences; the Swiss National Science Foundation (grant PP00P3_179089); the EU LIFE programme (DIVAQUA project, LIFE18 NAT/ES/000121); the UK Natural Environment Research Council (GLiTRS project NE/V006886/1 and NE/R016429/1 as part of the UK-SCAPE programme); the Autonomous Province of Bolzano (Italy); and the Estonian Research Council (grant no. PRG1266), Estonian National Program 'Humanitarian and natural science collections'. The Environment Agency of England, the Scottish Environmental Protection Agency and Natural Resources Wales provided publicly available data. We acknowledge the members of the Flanders Environment Agency for providing data. This article is a contribution of the Alliance for Freshwater Life (www.allianceforfreshwaterlife.org).

**Author contributions** P.H. conceived the study. N.B., J.H., D.H., S.C.J. and A.S.-K. contributed to initial ideas. E.A.R.W. performed data cleaning. E.A.R.W., D.E.B. and N.J.B. designed and conducted analyses. S.D. and J.R.G.M. provided environmental data with assistance from G.A., J.K. and L.Q.S. E.A.R.W. and P.H. wrote the first draft with assistance from D.E.B., and all of the

authors contributed to additional versions. F.A., M.Á.-C., D.G.A., G.A., I.A.J., T.A., I.A., I.B., J.B.O., C.L.B., L.B., N.B., R.B., M.C.-A., N.J.B., Z.C., T.D., E.d.E., A.D., G.D., E.D., K.A.E., J.E., T.E.E., V.E., M.J.F., M. Ferréol, M. Floury, M. Forcellini, M.A.E.F., R.F., N.F., J.-F.F., G.G., P.G., M.A.S.G., W.G., A.H., P.H., K.-L.H., T.C.J., R.K.J., J.I.J., L.K., A.L., P.L., L.L. M.-H.L., A.W.L., A. Maire, J.A.M.A., B.G.M., A. Millán, D.M., T.M., J.F.M., D.O., R.P., F.J.P., F.P., M.P., P.P., J.J.R., M.R., D.S.-F., L.S., R.B.S., A.S.-K., A. Scotti, A. Skuja, S.S., M.S., R. Stubbington, H.T., V.G.T., I.T., Y.U., G.H.v.d.L., R.V., E.V., G. Várbíró, G. Velle, P.F.M.V., R.C.M.V., Y.V. and P.W.-L. provided invertebrate data.

**Competing interests** The authors declare no competing interests.

**Additional information**
**Correspondence and requests for materials** should be addressed to Peter Haase or Ellen A. R. Welti.

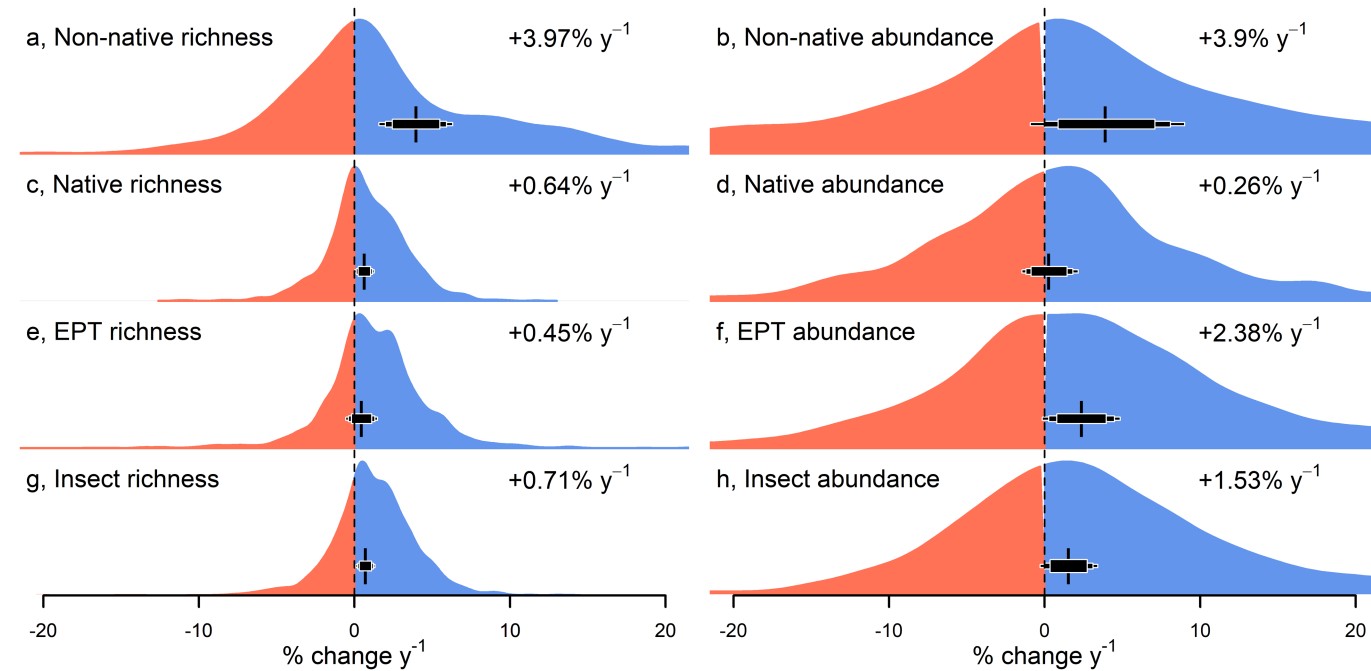

**Extended Data Fig. 1 | Trend estimates for community subsets.** Overall estimates and distributions of trends in **a**, non-native species richness, **b**, non-native abundance, **c**, native taxon richness, **d**, native abundance, **e**, Ephemeroptera, Plecoptera, and Trichoptera (EPT) taxon richness, **f**, EPT abundance, **g**, insect taxon richness, and **h**, insect abundance. Bars around estimates indicate 80%, 90%, and 95% credible intervals. Trend estimates for native taxa (c, d) are restricted to the 1,299 sites at which taxa were identified to species or a mixed taxonomic resolution. Trend estimates for non-native species (a, b) are restricted to the 898 (of 1,299) sites at which non-native species were detected. Incorporating the remaining 394 (30.1%) of the 1,299 sites (i.e. those with no detected non-native species) as having trends = 0 resulted in an average increase of 2.75% y$^{-1}$ in richness and 2.79% y$^{-1}$ in abundance.

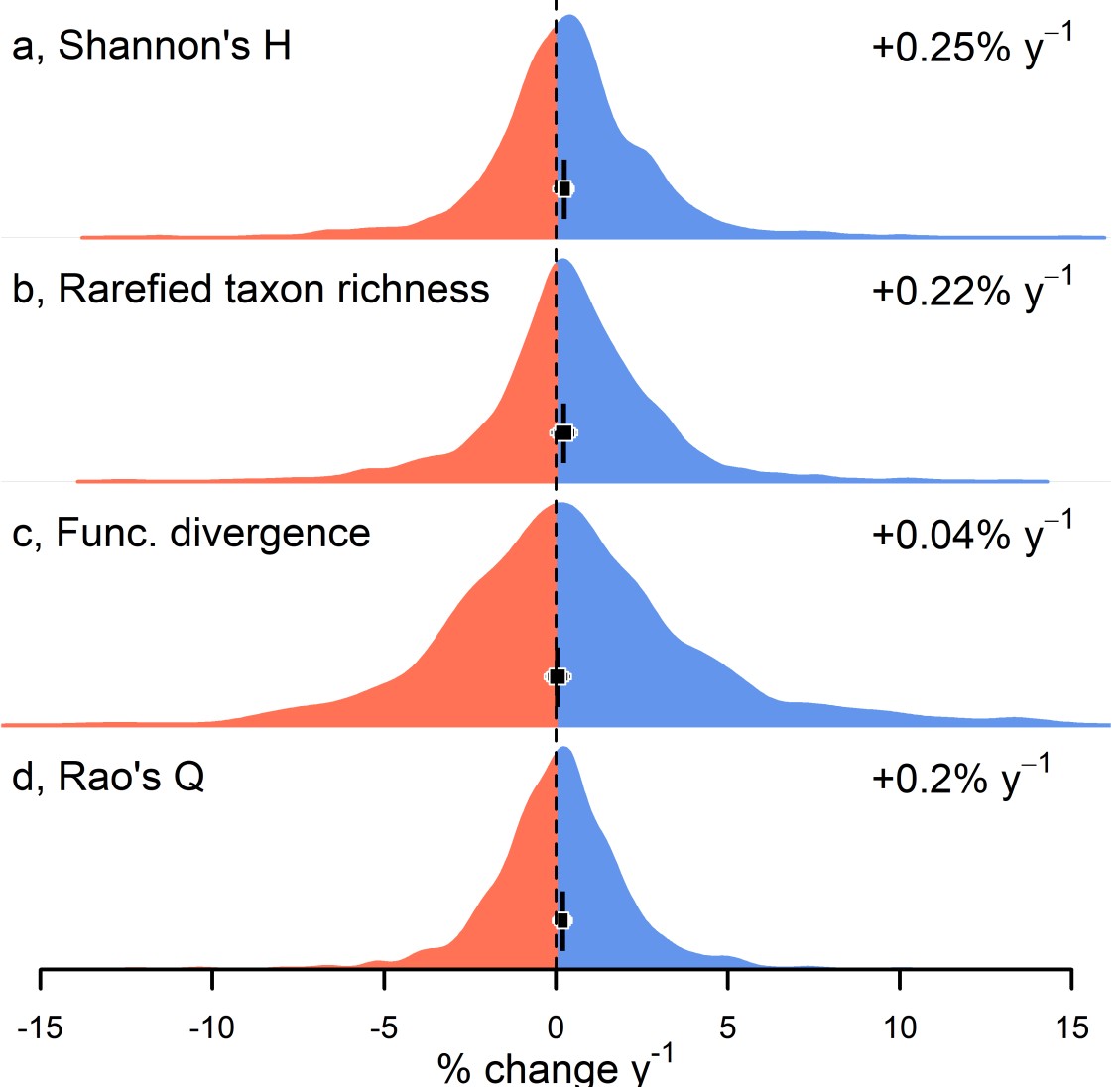

a, Shannon's H    +0.25% y$^{-1}$

b, Rarefied taxon richness    +0.22% y$^{-1}$

c, Func. divergence    +0.04% y$^{-1}$

d, Rao's Q    +0.2% y$^{-1}$

-15    -10    -5    0    5    10    15

% change y$^{-1}$

**Extended Data Fig. 2 | Trend estimates for additional biodiversity metrics.**
Overall estimates and distributions of trends in **a**, Shannon's diversity,
**b**, rarefied taxon richness, **c**, functional divergence, and **d**, Rao's quadratic
entropy (n = 1,816 biologically independent sites for all metrics). Bars around
estimates indicate 80%, 90%, and 95% credible intervals.

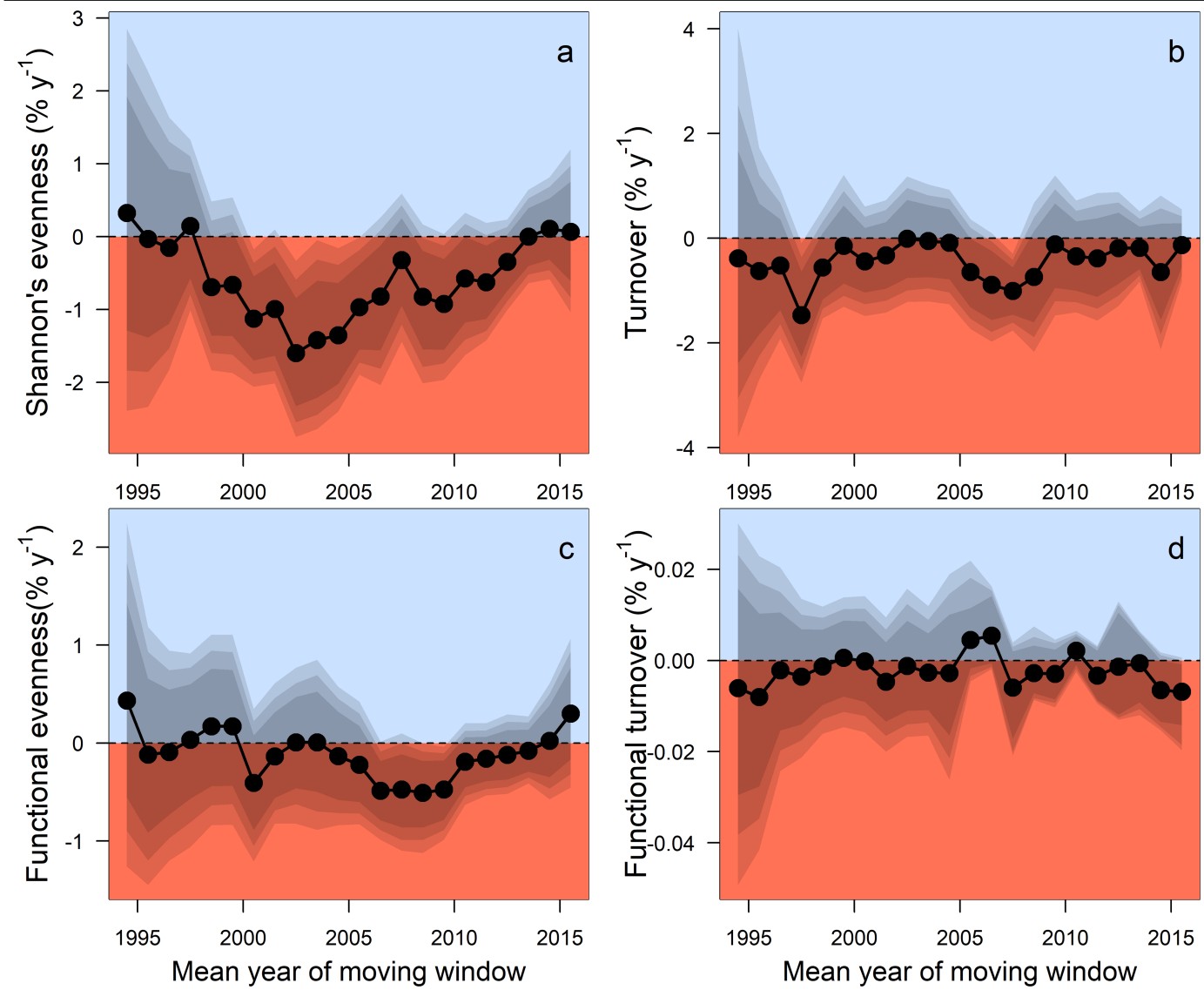

**Extended Data Fig. 3 | Moving window trends for additional biodiversity metrics.** Estimated trends in **a**, Shannon's evenness, **b**, taxonomic turnover, **c**, functional evenness, and **d**, functional temporal turnover. Estimates were calculated from Bayesian mixed-effects models of trends from ≥250 time series with ≥6 years of data from ≥8 countries within 10-year moving windows. Grey polygons indicate 80, 90, and 95% credible intervals.

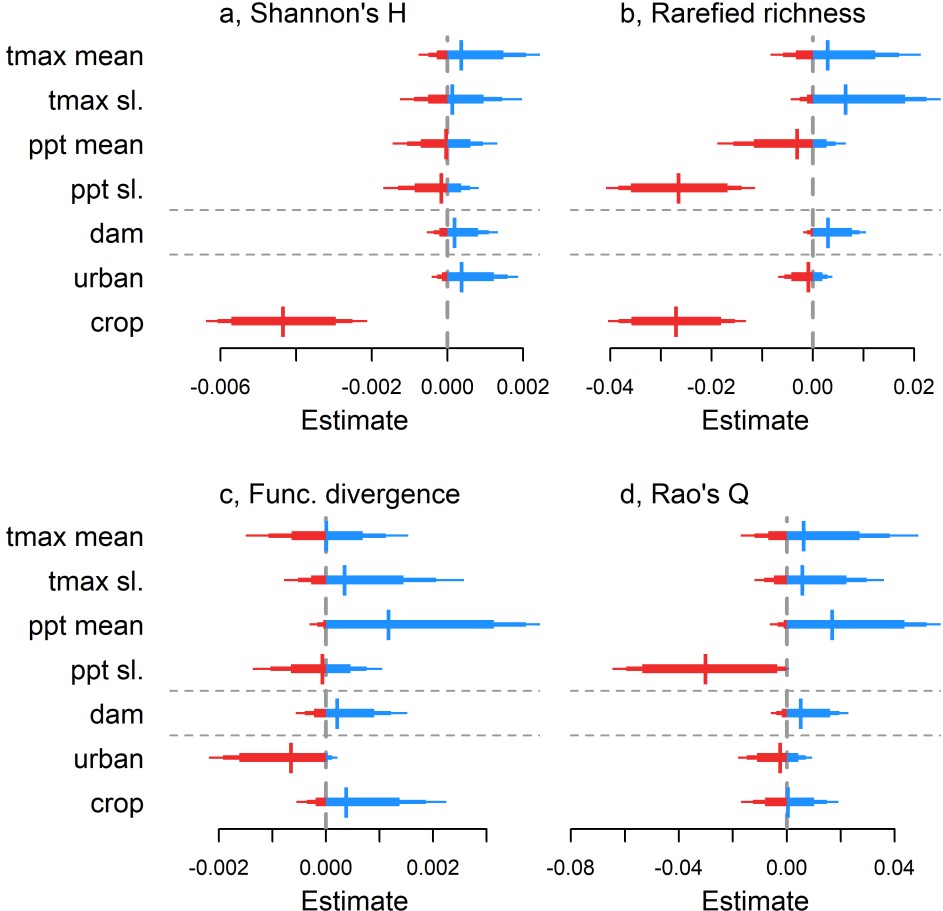

**Extended Data Fig. 4 | Estimated effects of environmental drivers on temporal trends in additional biodiversity metrics.** Estimated effects of the mean (tmax mean) and trend (tmax sl. [slope]) of annual maximum temperature, mean (ppt mean) and trend (ppt sl.) of annual precipitation, dam impacts (dam), and the percentage of the upstream catchment covered by urban areas and cropland on temporal trends in **a**, Shannon's diversity, **b**, rarefied taxon richness, **c**, functional (func.) divergence, and **d**, Rao's quadratic entropy (Q) (n = 1,816 biologically independent sites for all metrics). Bars around estimates indicate 80%, 90%, and 95% credible intervals. Grey, horizontal lines separate the three environmental driver groups: climate, dams, and land use.

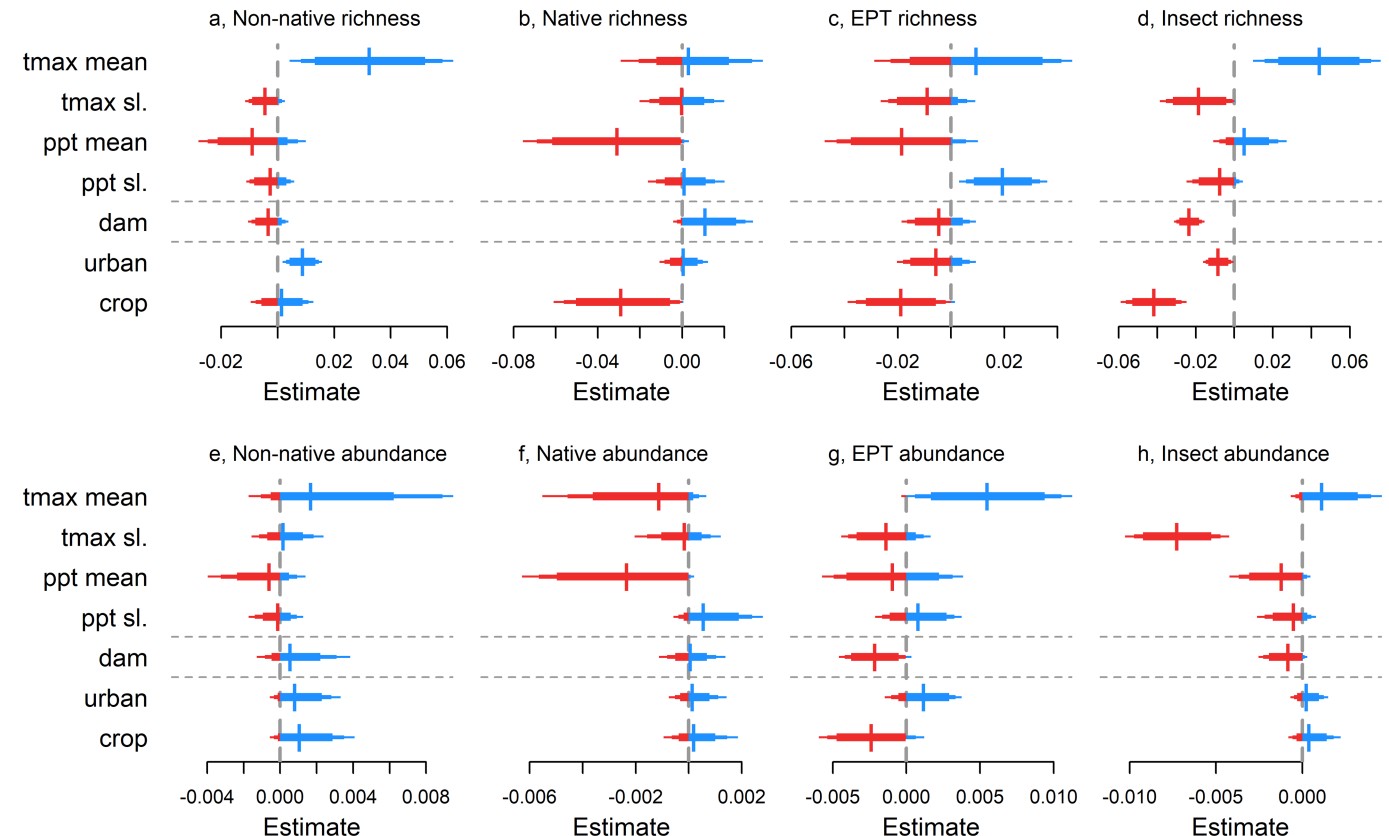

**Extended Data Fig. 5 | Estimated effects of environmental drivers on biodiversity metrics representing community subsets.** Estimated effects of the mean (tmax mean) and trend (tmax sl. [slope]) of annual maximum temperature, mean (ppt mean) and trend (ppt sl. [slope]) of annual precipitation, dam impacts (dam), and the percentage of the upstream catchment covered by urban areas and cropland on temporal trends in **a**, non-native species richness, **b**, native taxon richness, **c**, EPT richness, **d**, insect richness, **e**, non-native abundance, **f**, native abundance, **g**, EPT abundance, and **h**, insect abundance.

Trend estimates for native taxa (b, f) are restricted to 1,299 sites at which taxa were identified to species or a mixed taxonomic resolution. Trend estimates for non-native species (a, e) are restricted to the 898 (of 1,299) sites at which non-native species were detected. Bars around estimates indicate 80%, 90%, and 95% credible intervals. Bars around estimates indicate 80%, 90%, and 95% credible intervals. Grey, horizontal lines separate the three environmental driver groups: climate, hydrology, and land use.

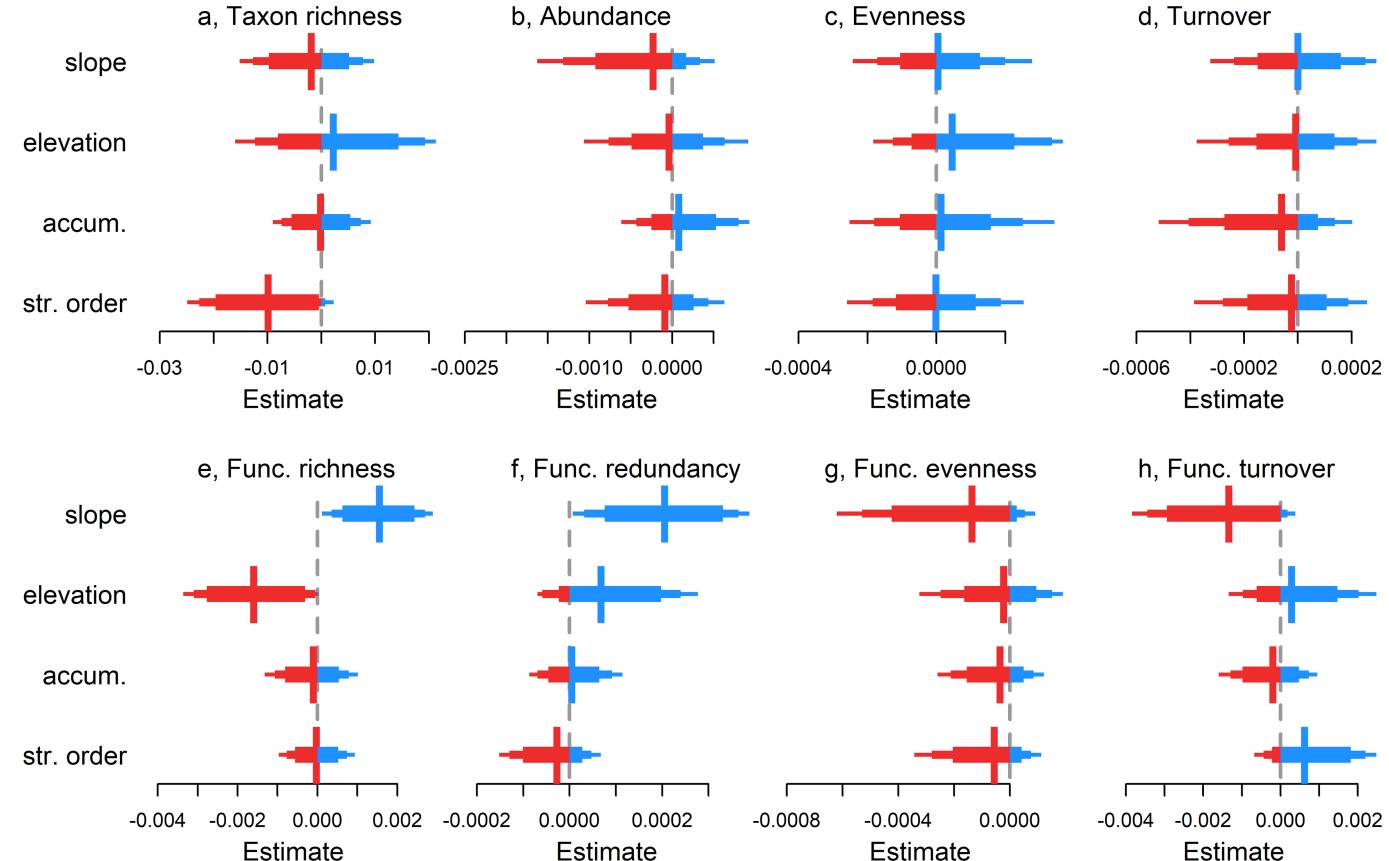

**Extended Data Fig. 6 | Estimated effects of stream characteristics on biodiversity metrics.** Estimated effects of slope, elevation, flow accumulation (accum.) and Strahler stream order (str. order) on temporal trends in **a**, taxon richness, **b**, abundance, **c**, evenness, **d**, turnover, and functional (func.) **e**, richness, **f**, redundancy, **g**, evenness, and **h**, turnover (n = 1,816 biologically independent sites for all metrics). Bars around estimates indicate 80%, 90%, and 95% credible intervals.

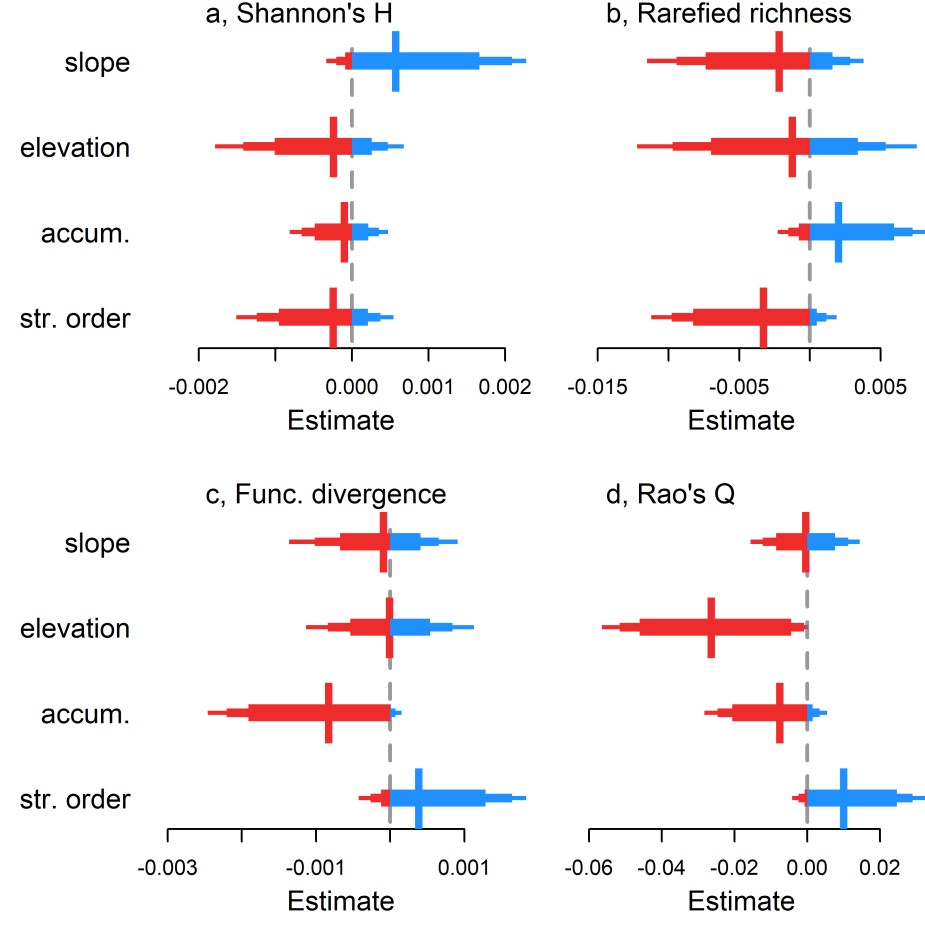

**Extended Data Fig. 7 | Estimated effects of stream characteristics on additional biodiversity metrics.** Estimated effects of stream characteristics of slope, elevation, flow accumulation (accum.) and Strahler stream order (str. order) on temporal trends in **a**, Shannon's diversity, **b**, rarefied taxon richness, **c**, functional (func.) divergence, and **d**, Rao's quadratic entropy (n = 1,816 biologically independent sites for all metrics). Bars around estimates indicate 80%, 90%, and 95% credible intervals.

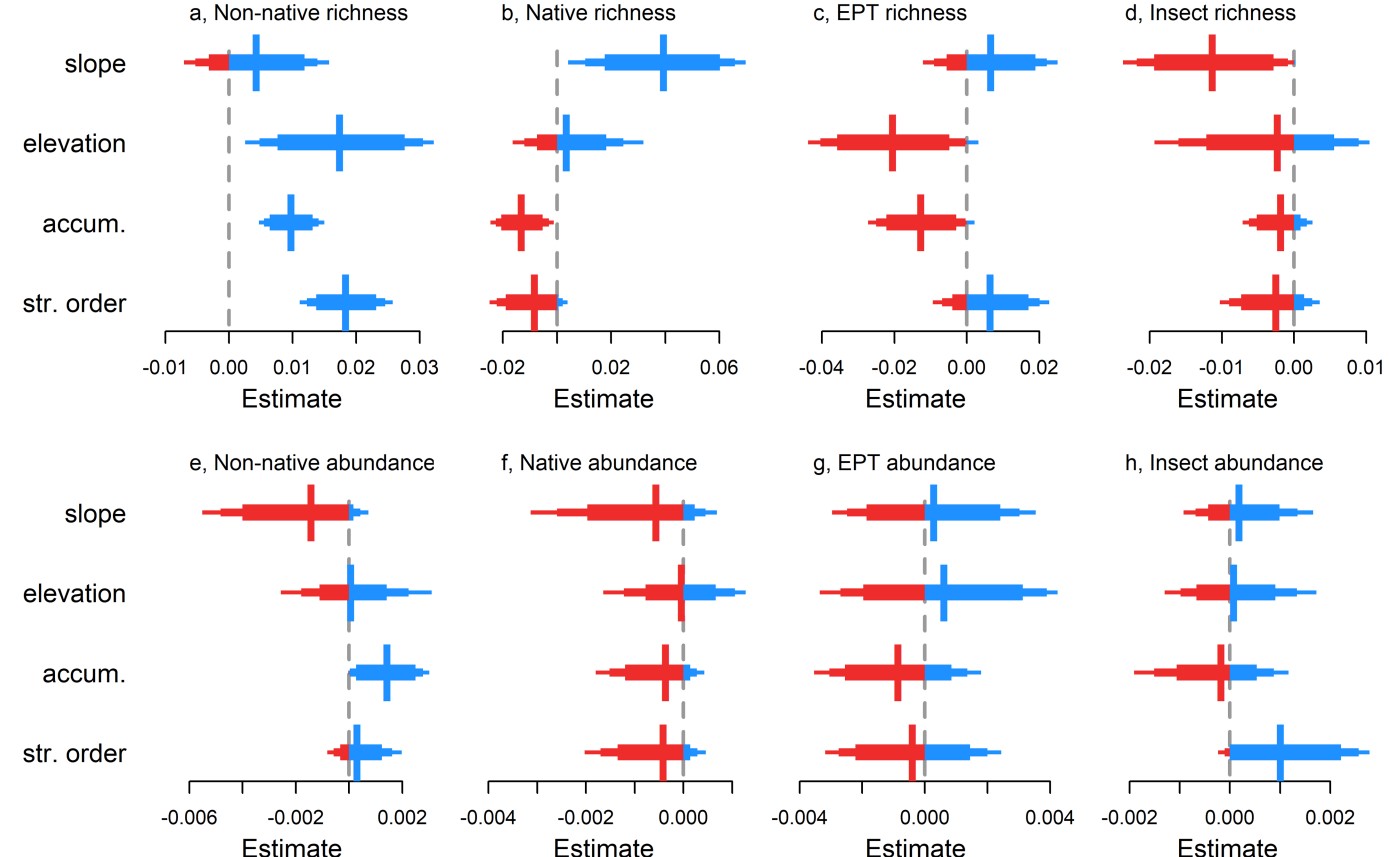

**Extended Data Fig. 8 | Estimated effects of stream characteristics on taxon richness and abundance of taxa subsets.** Estimated effects of slope, elevation, flow accumulation (accum.) and Strahler stream order (str. order) on temporal trends in **a**, non-native species richness, **b**, native taxon richness, **c**, EPT richness, **d**, insect richness, **e**, non-native abundance, **f**, native abundance, **g**, EPT abundance, and **h**, insect abundance. Trend estimates for native taxa (b, f) are restricted to 1,299 sites at which taxa were identified to species or a mixed taxonomic resolution. Trend estimates for non-native species (a, e) are restricted to the 898 (of 1,299) sites at which non-native species were detected. Bars around estimates indicate 80%, 90%, and 95% credible intervals.

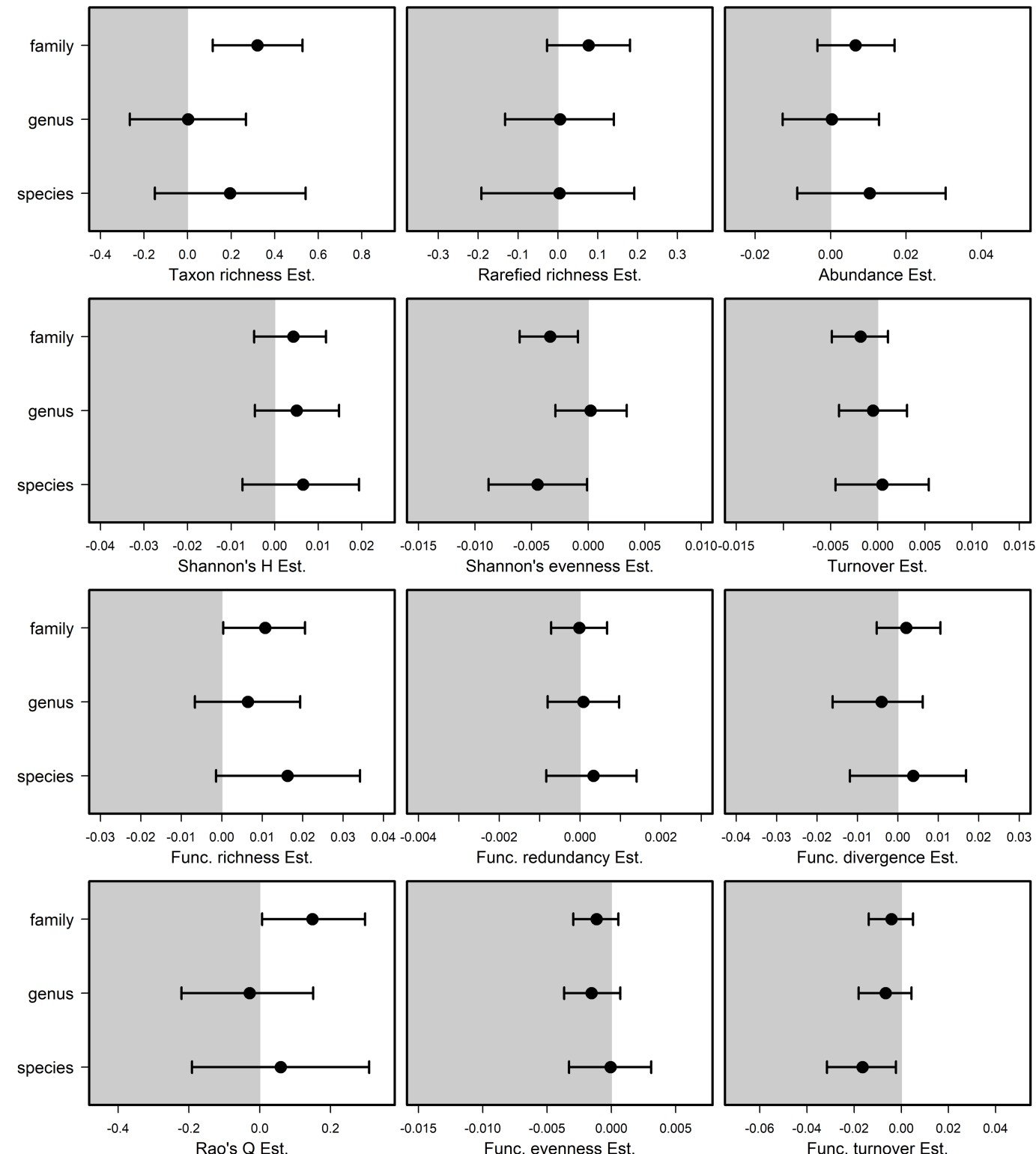

**Extended Data Fig. 9 | Sensitivity of biodiversity metric responses to taxonomic identification level.** Error bars represent 95% credible intervals. Overlapping error bars indicate comparable trend estimates for analyses at species (n = 762), genus/mixed (n = 537) and family (n = 517) taxonomic levels; Func., functional; Est., estimated trend.

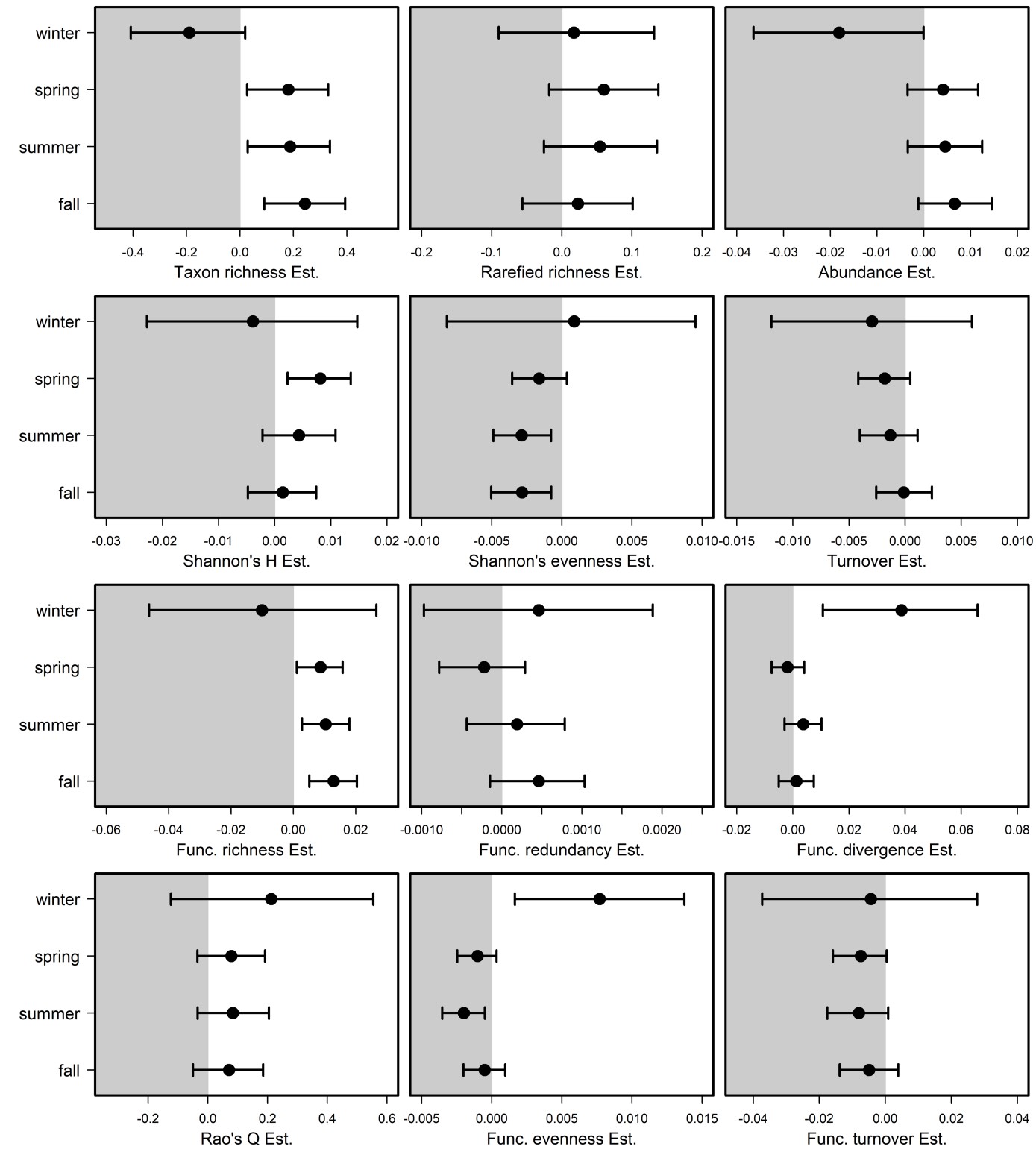

**Extended Data Fig. 10 | Sensitivity of biodiversity metric responses to sampling season.** Error bars represent 95% credible intervals. The largest differences between seasons were found for winter, which likely reflects the low number of sites sampled in this season (winter n = 5, spring n = 623, summer n = 473, fall n = 715). Func. refers to functional; Est. refers to trend estimates.

# Reporting Summary

## Statistics

For all statistical analyses, confirm that the following items are present in the figure legend, table legend, main text, or Methods section.

| n/a | Confirmed | |
|---|---|---|
| ☐ | ☒ | The exact sample size (*n*) for each experimental group/condition, given as a discrete number and unit of measurement |
| ☐ | ☒ | A statement on whether measurements were taken from distinct samples or whether the same sample was measured repeatedly |
| ☐ | ☒ | The statistical test(s) used AND whether they are one- or two-sided *Only common tests should be described solely by name; describe more complex techniques in the Methods section.* |
| ☐ | ☒ | A description of all covariates tested |
| ☐ | ☒ | A description of any assumptions or corrections, such as tests of normality and adjustment for multiple comparisons |
| ☐ | ☒ | A full description of the statistical parameters including central tendency (e.g. means) or other basic estimates (e.g. regression coefficient) AND variation (e.g. standard deviation) or associated estimates of uncertainty (e.g. confidence intervals) |
| ☐ | ☒ | For null hypothesis testing, the test statistic (e.g. $F$, $t$, $r$) with confidence intervals, effect sizes, degrees of freedom and $P$ value noted *Give P values as exact values whenever suitable.* |
| ☐ | ☒ | For Bayesian analysis, information on the choice of priors and Markov chain Monte Carlo settings |
| ☐ | ☒ | For hierarchical and complex designs, identification of the appropriate level for tests and full reporting of outcomes |
| ☒ | ☐ | Estimates of effect sizes (e.g. Cohen's *d*, Pearson's *r*), indicating how they were calculated |

*Our web collection on statistics for biologists contains articles on many of the points above.*

## Software and code

Policy information about availability of computer code

| | |
|---|---|
| Data collection | No software was used in data collection. |
| Data analysis | Annotated R (ver. 4.2.2); all scripts are available at GitHub: https://github.com/Ewelti/EuroAquaticMacroInverts |

For manuscripts utilizing custom algorithms or software that are central to the research but not yet described in published literature, software must be made available to editors and reviewers. We strongly encourage code deposition in a community repository (e.g. GitHub). See the Nature Portfolio guidelines for submitting code & software for further information.

## Data

Policy information about availability of data

All manuscripts must include a data availability statement. This statement should provide the following information, where applicable:
- Accession codes, unique identifiers, or web links for publicly available datasets
- A description of any restrictions on data availability
- For clinical datasets or third party data, please ensure that the statement adheres to our policy

Metadata, site characteristics, and trend estimates are available on GitHub: https://github.com/Ewelti/EuroAquaticMacroInverts. Raw biodiversity data will be made available in the same GitHub repository and to the BioTime database following acceptance and a six-month embargo.

## Research involving human participants, their data, or biological material

Policy information about studies with human participants or human data. See also policy information about sex, gender (identity/presentation), and sexual orientation and race, ethnicity and racism.

| | |
|---|---|
| Reporting on sex and gender | Does not apply to our study |
| Reporting on race, ethnicity, or other socially relevant groupings | Does not apply to our study |
| Population characteristics | Does not apply to our study |
| Recruitment | Does not apply to our study |
| Ethics oversight | Does not apply to our study |

Note that full information on the approval of the study protocol must also be provided in the manuscript.

## Field-specific reporting

Please select the one below that is the best fit for your research. If you are not sure, read the appropriate sections before making your selection.

☐ Life sciences    ☐ Behavioural & social sciences    ☒ Ecological, evolutionary & environmental sciences

For a reference copy of the document with all sections, see nature.com/documents/nr-reporting-summary-flat.pdf

## Ecological, evolutionary & environmental sciences study design

All studies must disclose on these points even when the disclosure is negative.

| | |
|---|---|
| Study description | The study is a meta-analysis of 1,816 time series of freshwater macroinvertebrate communities to examine biodiversity trends over time and across Europe. Overall estimates of slopes of biodiversity metrics were calculated using a Bayesian hierarchical model (2-step model). Step 1 involved calculating individual slopes for each time series. Step 2 involved an calculating overall estimate and an overall intercept and two random effects (country and study identity) as predictors. |
| Research sample | Data were collected from previous studies and assembled from a data call. |
| Sampling strategy | No sample-size calculation was preformed. Time series were included in analyses when they met selection criteria, resulting in a collection of 1,816 time series. |
| Data collection | Data were assembled from a data call to European ecologists and environmental managers. Peter Haase put out the data call and Ellen Welti assembled data from data providers. |
| Timing and spatial scale | All of the 1,816 time series contain annual sampling of a minimum of 8 years of data. All time series combined span the period of 1968-2020. |
| Data exclusions | All time series obtained in the data call were included if they met the pre-selected criteria of: 1) inclusion of abundance estimates, 2) surveyed whole freshwater invertebrate communities (not restricted to certain taxonomic groups, such as insects), 3) identified most major taxa to family, genus or species, 4) had a minimum of eight sampling years (not necessarily consecutive), 5) had no changes in sampling method or taxonomic resolution during the sampling period, and 6) had consistent sampling effort per site (e.g. number of samples or area of river sampled) across years. |
| Reproducibility | No new experiments were performed in this meta-analysis. All code, meta-data, and slope estimates are provided on Github: https://github.com/Ewelti/EuroAquaticMacroInverts |
| Randomization | The study is a meta-analysis of pre-collected time series data, and does not including new experimental designs requiring randomization. When testing for overall estimates of change in biodiversity metrics over time, study and country were included in models as random effects. |
| Blinding | Blinding was not relevant to this study. |

Did the study involve field work?    ☐ Yes    ☒ No

## Reporting for specific materials, systems and methods

We require information from authors about some types of materials, experimental systems and methods used in many studies. Here, indicate whether each material, system or method listed is relevant to your study. If you are not sure if a list item applies to your research, read the appropriate section before selecting a response.

## Materials & experimental systems

| n/a | Involved in the study |
|-----|----------------------|
| ☒ | ☐ Antibodies |
| ☒ | ☐ Eukaryotic cell lines |
| ☒ | ☐ Palaeontology and archaeology |
| ☐ | ☒ Animals and other organisms |
| ☒ | ☐ Clinical data |
| ☒ | ☐ Dual use research of concern |
| ☒ | ☐ Plants |

## Methods

| n/a | Involved in the study |
|-----|----------------------|
| ☒ | ☐ ChIP-seq |
| ☒ | ☐ Flow cytometry |
| ☒ | ☐ MRI-based neuroimaging |

# Animals and other research organisms

Policy information about studies involving animals; ARRIVE guidelines recommended for reporting animal research, and Sex and Gender in Research

| | |
|---|---|
| Laboratory animals | The study did not involve laboratory organisms. |
| Wild animals | Data include time series from previous studies of field collections of freshwater macroinvertebrates. Macroinvertebrates were killed to identify specimens in these studies. Details are provided in the Methods and Supplemental Tables. |
| Reporting on sex | Does not apply to our study. |
| Field-collected samples | Data include time series from previous studies of field collections of freshwater macroinvertebrates. Macroinvertebrates were killed to identify specimens in these studies. Details are provided in the Methods and Supplemental Tables. |
| Ethics oversight | No ethical approval or guidance was required as data were collected only from previous studies. |

Note that full information on the approval of the study protocol must also be provided in the manuscript.

