## [Peer Review File · Nature]

Manuscript Title: The recovery of European freshwater biodiversity has come to a halt

Reviewer Comments & Author Rebuttals

Reviewer Reports on the Initial Version:

Referees' comments:

Referee #1 (Remarks to the Author):

This paper analyzes a massive data set on freshwater invertebrate biodiversity in Europe and its change through time. The manuscript found that in general, invertebrate biodiversity was increasing through time as inferred by a variety of biodiversity metrics. Trends at sites were linked to land-use variables, with more impacted sites not increasing as much.

Originality and significance. I appreciated the impressive dataset, the work to do spatial analyses on all of the sites, and the overall exploration of trends and watershed/climate factors. It is an impressive paper. I also think the narrative of increasing biodiversity and the success of environmental policies is one that does not receive sufficient attention in environmental science.

I did have several concerns or critiques about the paper. Some of these concerns are about the interpretation of the data. Other critiques are more minor in nature. These are outlined below.

1. My main concern was with the robustness of the key finding—that “recovery of European freshwater biodiversity has come to a halt” (title, section “Biodiversity gains have come to a halt”). This key finding does not appear to be that strongly supported by the analysis. This finding references Figure 3, which shows the results of a moving window analysis, where trends were run through a sliding window of 10 years of data. First, throughout the time series, the confidence intervals (even the 80% CI) overlap with values that are greater than 0 (increases of freshwater biodiversity). Thus, the analysis doesn't appear to be able to rule out the possibility that freshwater biodiversity continues to increase. Second, the eye is drawn to the points on the far right that drop below zero. However, these points are based on increasingly small sample sizes as that shifting window runs out of data points and thus are not strictly comparable. I also question that ongoing “recovery” is a reasonable assumption given that presumably some of the 1800+ sites were not that degraded in the first place. For these reasons, I don't think the data convincingly demonstrate that recovery has halted.

2. One of the aspects of the manuscript that could be strengthened is to clarify the degree to which sensitivity of findings are to data variability. With such a big dataset, it is not surprising that the data are not consistent across space, time, and resolution. However, it is challenging to assess whether these differences influence the patterns described. I suggest that there are sensitivity analyses that could help address these potentials.

a. The dataset had a mixed of taxonomic resolutions in it. Many of the response variables--taxon richness, taxon turnover, etc. could be strongly influenced by this resolution. I appreciated Extended Data Fig. 9, that did examine include taxonomic resolution as a factor, but I think it would be helpful to subset the data and rerun the main analyses to directly test this potential (e.g., those that are shown in Fig. 3 and 4). I also note that figure caption of Extended Data Fig. 9 was a bit skimpy and could have used a bit more detail to aid interpretation.

b. Similarly, the dataset also had mixed coverage across sites (e.g., Fig 1c). It might be helpful to rerun with the subset of the data with data that was fully spanning the data to rule out the potential that nonrandom patterns of site coverage across space and time contributed to patterns.

3. The paper emphasized the importance of functional diversity. One challenge for with this was

that the approach didn't consider specific functions, but rather aggregated all of the traits, that ranged from body size to life-history. This makes this concept of functional diversity fairly abstract. I also note that many traits had to be inferred based on other species. Thus, while the functional diversity analyses seemed fine, from my perspective they did not add that much to the paper. This is not a very strong suggestion from me, so feel free to disregard.

4. Figure 4. The predictor variables are a mixture of static and dynamic variables, which makes it hard to interpret. Line 196-198 should be revised to be more precise to reflect that some of the relationships could be that the trend differs across the variable (e.g., sites with dams had lower trends in richness through time).

5. The paper made some assertions about conservation/management that should be considered. For example:

a. Line 41-43. "Determining the trajectories . . . is a prerequisite. ." I disagree with this assertion. First, much management is driven by single-species approaches or other conservation values. Second, management can be informed by a variety of knowledge or data, it doesn't always have to be reactive.

b. Line 249-250. This felt a bit dramatic and also it wasn't clear what was precisely meant by "flexible management strategies".

6. Analysis. The analysis I think struggled a bit to deal with time, space, and different environmental conditions. I think there probably could have been a more elegant approach that used a hierarchical Bayesian time series approach that then incorporated site-level dynamic (e.g., precipitation) and static (e.g., dams) variables as potential modifiers of the trend. Instead, what was run was a site-specific linear trend analysis and then a post-hoc analysis of the site-specific linear trend estimates. It does not appear that error was fully propagated across these steps (i.e., the posterior distributions of the trends were not used as inputs into the secondary analyses), which is a potential issue. This current approach was ok (and impressive in its own right) but I think it made it a bit hard to provide an integrated perspective into how biodiversity is changing across space and time.

Referee #2 (Remarks to the Author):

Haase et al. describes how river macroinvertebrate communities have changed across a large swathe of Europe in recent decades. It reveals a complex picture in which different facets of taxonomic and functional diversity have changed in different ways through time, among river types and in response to different pressures. Evidence is presented that increases in taxon and functional richness in particular have slowed over the last c. 25 years, with no net change in the last 5 years or so. Assessing functional diversity alongside taxonomic diversity is a valuable addition, revealing more nuanced (and complex) changes.

The core result - the slowing down or cessation of richness gain in the 2000s - fits with earlier studies (e.g. Vaughan & Ormerod 2012, *Global Change Biology* 18, 2184-2194; Outhwaite et al 2020, *Nature Ecol. Evol.* 4, 384-392) but expands it to both a wider range of diversity measures and, most importantly, identifies it at a continental scale suggesting a more general mechanism.

There is little question that this is an important topic and timely assessment, given the evidence of disproportionately large biodiversity declines in freshwaters (e.g. the Living Planet Index for freshwaters) and the ongoing debate around insect/invertebrate declines and evidence for increases in freshwaters cf. terrestrial systems (e.g. van Klink et al. 2020, *Science* 368, 417-420). The authors are to be commended on compiling such a large and impressive international data set, allowing for a potentially important contribution to this debate.

With just a few minor exceptions, the manuscript is well written and the methodology is clear to follow.

There are a few areas that I think would benefit from some more consideration. I've tried to group these together under related headings:

Recovery

There's an implicit assumption that the overall average increases in diversity represent recovery and that the slowing down of annual changes indicates that recovery has halted. I agree that it is increasingly difficult to find pristine reference streams to help to contextualise changes (line 245), but your analysis raised a couple of issues in my mind:

1. An alternative interpretation of the results might be that recovery is nearing completion when viewed across Europe as a whole (with specific exceptions highlighted in Fig 4), so that there is no further recovery to 'revive' (line 202+). Is it possible to refute this idea with the current data?
2. I was interested in your findings that (1) increases in diversity were smaller, or declines were evident, in catchments with cropland or urban land use – exactly the sorts of locations where, if anything, we might expect larger recoveries on the back of historically poorer water quality (Hypothesis 2) – and (2) the percentage increases in EPT abundance and richness were smaller than insects as a whole – wouldn't you expect recovery from historical stressors to be characterised by larger increases amongst the relatively sensitive EPT taxa compared to insects more generally? Taking both of these together, can you be confident that this really is recovery as opposed to a more general biodiversity change driven by, for example, climate?

Assessment of change through time

I think the first analysis of the paper – the overall mean trends through time – needs some more thought. Presenting a mean value per year implies that change is more or less linear through time, yet the second analysis (= main message of the paper) contradicts that, as does evidence of a strongly nonlinear change through time over a similar timeframe (1970 onwards) from at least one of the constituent countries (UK; Outhwaite et al 2020). Indeed, in the case of the latter, despite large changes in freshwater invertebrates through time, there was negligible net change between the start and end of the time series. An explicit consideration of the potential for nonlinear change since 1968 is important and could provide further insights from your data.

Following on from this, the apparent end of improvements is such an important result: is there a more compelling image that can be used to demonstrate this slow down? e.g. tracking an abundance index through time to demonstrate the plateau. This might also help to address my previous point.

Other recent studies have revealed the influence of time series length upon the resulting average trends (e.g. Leung et al. 2020; Nature 588, 267-271). Are you able to rule out this issue here?

Not surprisingly, there was very restricted coverage geographically prior to the 1990s. How much does the first ~20 year of the time series influence the final trend estimates?

It's good to see that the potential problems of data heterogeneity through time are considered (e.g. line 562). Could you assess this more directly e.g. leaving different countries out of the analysis in the manner of a k-fold cross-validation so see what some of the effects are?

Similarly, it's good to see the inclusion of various sensitivity analysis. If I understand correctly, in

the case of varying taxonomic resolution, different data sets are used for the different taxonomic levels, presumably representing the resolution at which they were recorded. Would it not be better to use the species-level data throughout, coarsening the taxonomic resolution, rather than using different data sets? This would eliminate other sources of variation and be a purer assessment of the effects of resolution.

Other general points

Functional diversity needs some more explanation in the main text, both in terms of technical aspects (e.g. what 'functional space' is) and giving an overview of what the different functional measures tell us (+ how they complement one another)

I couldn't find any mention of potential biases arising from the sampling strategies used in the constituent data sets e.g. were sampling locations selected randomly, focused on particular stream orders, purposefully chosen to monitor the effects of wastewater treatment outfalls, etc? I think it would be extremely difficult to account for such factors in the analysis, but important to provide some consideration of this issue and its potential impacts.

Freshwater macroinvertebrates are often viewed as indicators for environmental quality as a whole, not just in river systems because – as you say – they integrate processes occurring across their catchments. Are you able to comment any more on this e.g. the extent to which the observed changes are telling us about the terrestrial environment in addition to the aquatic one?

Specific minor points

Abstract: Are the net increases across the whole time series or just prior to the mid-2010s? It seems a bit strange say that increases stopped by the mid-2010s, then present the annual increases

line 98, functional diversity: Needs a more accessible definition for a general audience. Also it would be useful to provide some more explanation of the meaning and benefits of the multiple metrics of this diversity e.g. for general readers, the interpretation and importance of functional evenness is unlikely to be obvious

line 120, non-linear trajectories. Could non-linearity also be generated by falling recovery rates as recovery/gains from mitigation measures are exhausted?

line 147 – urban areas are also likely to be a proxy for point source pollution (cf. diffuse)

line 152, percent increases in temperatures will only be meaningful if expressing the change in Kelvins. Do you need to present it in this way? Degrees C would be more interesting and informative

line 158, lower recovery rates in cooler areas due to less severe degradation. Good point - it would be interesting to see how change was related to existing biodiversity e.g. smaller/no gains in functional richness in streams which were already rich

line 166. Precipitation could also play a role via varying water quality (runoff, dilution...)

line 215, positive effects of rising temperatures at high latitudes. Why not test that directly here?

line 505 – say why you used gls in this instance

lines 592+ – more details needed to describe the sensitivity analyses

Referee #3 (Remarks to the Author):

The extensive data set analysed in this paper is a considerable strength – I know of no other paper with equivalent depth of coverage of biodiversity data from a large biogeographic region. Its focus on invertebrates is laudable, and a welcome change from analyses that consider only vertebrates. The paper's take home message – of initial recovery, subsequently halted – has important policy implications. As far as I am aware, this is an original finding.

With papers of this sort, the details of the analyses are paramount. I found it difficult, in places, to fully understand what was done, or what the raw data looked like. Line 370 states that the authors used time series that included the whole invertebrate community. But we are not told which groups were sampled at each site. This information should be included in Supplementary Table 2. Biodiversity analyses are notoriously vulnerable to variation in sampling effort. The authors note that they standardised the data by including only one sampling season (line 378). They further note (line 398) that they computed rarefied taxon richness. It is not clear whether they mean a sample-based rarefaction (as implied above) or some other form of rarefaction. This leads me to wonder to what extent change in the various metrics is underpinned by shifts in abundance. Clearly a more comprehensive description of data handling and analysis is needed here. At present there is insufficient information to replicate the analyses.

With regard to functional diversity I wondered how the authors dealt with taxa at different developmental stages (given that the functional roles of aquatic invertebrates can shift markedly as they mature).

The authors used a variety of statistical methods to estimate diversity trends. A headline result is that gains in biodiversity have been reversed in the last decade. The moving window analysis appears to be key here, but as the authors mention, there are many caveats involved. A complicating factor (evident from Figure 1c) is the much higher number of sites post 2000. The authors note this, and have tried to deal with it by focussing the analysis on a subset of sites. Nonetheless, reassurance that change in sampling effort has not influenced the results is needed, and that either increases in the types of sites included in the data set, or spatial biases, cannot account for the conclusions.

From a conceptual standpoint, Figure 3 is interesting. It shows 4 metrics for which the earlier values are higher than the later ones (but with the credible intervals overlapping zero to a large extent). (Extended data Figure 3 is the equivalent plot for 4 more metrics, but – the final data point apart – are much less convincing of a shift in diversity). Should we expect gains in diversity to be sustained continuously? Surely these systems have some 'natural' baseline level of diversity around which levels will fluctuate but not necessarily show directional change? To what extent are the observed trends indicative of a flattening off in recovery as it reaches some equilibrium, as opposed to a reversal of gains? Is the shift a consequence of climate change? – the paper notes that warming is associated with an increase in diversity (a result that tallies with other work in the field). Figure 4 seems to be an analysis of overall trends in relation to various drivers, rather than one specifically focused on shifts in trends in the last decade. It left me wondering to what extent these drivers account for the reported downturn in diversity. Can these not be explicitly investigated?

A general observation I had is that the paper can't quite decide if it is an analysis of overall trends, or is focussed mainly on the break point in trends. It contains many very interesting analyses and is potentially influential, but in places it needs to provide more information, and additional analyses, to support the conclusions it draws.

Referee #4 (Remarks to the Author):

This report concerns only the statistics aspect of the article, I do not comment on the other scientific aspects.

Overall, this paper uses statistical models to analyse time series data on riverine invertebrate communities. The statistical models they used was based on a two stage approach: first stage obtains a trend from time series separately for each time series, then the second stages use these trends as response to combine spatially separate time series to obtain an estimate of an overall trend. The approach does not account for all the uncertainty in the estimates, potentially leading to an over-confidence of changes in trend.

Overall, I find that the statistical writing can be improved, expressions (particularly conclusions) can be made more precise and the details of the statistical models should be provided. As the manuscript stands now, I found it difficult to follow the statistical techniques used. Below I give specific comments and some suggestions on how to make the statistics sections easier to read.

L77, when the authors say taxon richness increased by a net mean of 0.94%, what do they actually mean by this? Is it the differences between the first and last observed values, or the fitted estimate of trend? Why not provide confidence intervals to indicate whether this change is statistically significant?

L79-86 "posterior distributions revealed 99% and 99.3% probabilities of a net increase...".

- firstly, percentages are not probabilities, the authors mean 0.99 probability of;
- secondly what is the meaning of this sentence? Posterior distributions of what? What is the confidence interval for these probabilities?

L117, the use of a 10-year moving average window is not well justified, as we know, the width of the window has large impact on the results.

L116, are the conclusions in this section based on visual observations of the 10-year moving average smoothing? Is there any statistical tests that were carried to test for statistically significant decline? e.g., in Fig 3a.

L151 – 167, the affects of climate on the invertebrate communities discussed here are based on which model? and when reporting the results, it is best to give 95% confidence interval and state which increases or decreases are statistically significant, SE alone does not tell you that.

L416-428, it was described the method used to impute missing trait data. However, it seems to me that only 15.9% are available at the original identification level, the rest are proxies. Even if the genus-level trait data is a reasonable substitute, there's still more than 50% missing. What assurances can the authors demonstrate that their approach for imputing the missing data does not lead to potentially wrong conclusions?

L515, the authors should precisely write down (using equations) the statistical model used, both for the first stage and the second stage using LMM and the trend estimate from the first stage as response.

L523, it was mentioned that an AR(1) model was used, can the authors justify this choice?

L536, it is stated that the probability of mean increase is calculated from the posterior distribution, how is it calculated?

L539, what priors were actually used? A more precise expression should be given here.

L541, moving-window analysis, as I mentioned before, why a 10 year width of window is chosen? Additionally, why not use something like the splines, which can mitigate some of the issues mentioned by the authors in terms of boundary bias.

L560 (and later), when the authors say, "weighted by their inverse error to account for uncertainty", what do they mean by this? How is this done?

L585, should R-hat value be around 1? (why <1.1 ?)

Author Rebuttals to Initial Comments:

Response to Reviewers

Referee #1 (Remarks to the Author):

This paper analyzes a massive data set on freshwater invertebrate biodiversity in Europe and its change through time. The manuscript found that in general, invertebrate biodiversity was increasing through time as inferred by a variety of biodiversity metrics. Trends at sites were linked to land-use variables, with more impacted sites not increasing as much.

Originality and significance. I appreciated the impressive dataset, the work to do spatial analyses on all of the sites, and the overall exploration of trends and watershed/climate factors. It is an impressive paper. I also think the narrative of increasing biodiversity and the success of environmental policies is one that does not receive sufficient attention in environmental science.

I did have several concerns or critiques about the paper. Some of these concerns are about the interpretation of the data. Other critiques are more minor in nature. These are outlined below.

1. My main concern was with the robustness of the key finding—that “recovery of European freshwater biodiversity has come to a halt” (title, section “Biodiversity gains have come to a halt”). This key finding does not appear to be that strongly supported by the analysis. This finding references Figure 3, which shows the results of a moving window analysis, where trends were run through a sliding window of 10 years of data. First, throughout the time series, the confidence intervals (even the 80% CI) overlap with values that are greater than 0 (increases of freshwater biodiversity). Thus, the analysis doesn’t appear to be able to rule out the possibility that freshwater biodiversity continues to increase. Second, the eye is drawn to the points on the far right that drop below zero. However, these points are based on increasingly small sample sizes as that shifting window runs out of data points and thus are not strictly comparable. I also question that ongoing “recovery” is a reasonable assumption given that presumably some of the 1800+ sites were not that degraded in the first place. For these reasons, I don’t think the data convincingly demonstrate that recovery has halted.

Author response:

We agree with the reviewer that this key finding (recovery has come to a halt) requires strong justification. In the revised manuscript and associated files, to provide additional support that there is a declining trend in taxon richness, we have 1) conducted an additional statistical test of the overall linear changes in trends within the moving window analyses, 2) examined changes in the proportion of sites with positive (compared to negative) trends in biodiversity metrics through time to examine dominant patterns, 3) tested an alternative moving window analyses that shows changes in the mean site trends through time using a higher data-inclusion threshold, and 4) removed the points based on the shorter windows at the end of the time series in our main moving window analysis. We discuss each of these revisions in more detail as follows:

(1) We modeled the effect of year on moving window trends using a simple Bayesian model of:

Moving_window_trend|se(Moving_window_trend_error) ~ Moving_window_year

to test for a directional change with time in our trend analysis. The script for this analysis can be found at <https://github.com/Ewelti/EuroAquaticMacroInverts> in folder R: MetaMeta_movingwindowEsts_Yr.R. This analysis was also included in response to comments from Reviewer 4. These models identified declines in taxon richness trends (Estimate = -0.0192, SD = 0.006) with 95% CI not overlapping zero. Functional richness trends also tended to decline (Estimate = -0.0004, SD = 0.0002), with 95% CI barely overlapping zero. No directional change was found for abundance trends (Estimate = 0.00009, SD = 0.0002) or functional richness trends (Estimate = -0.00003, SD = 0.00003) as 95% CIs overlapped zero. We now include estimates converted into change in trends per year in the main manuscript (lines 133-148):

Although trends in taxon richness were generally positive, indicating increases in local richness through time, this effect became weaker over the decades (mean change in trends of taxon richness = $-8.8\% \text{ y}^{-1}$, 95% CI range: -13.6% to $-3.8\% \text{ y}^{-1}$). Trends in taxon richness started declining around 2010 and then levelled off, reaching an average of net zero around 2013 (Fig. 3a), indicating an end to the preceding recovery period. When considering only the dominant pattern as measured by the proportion of positive trends, the number of sites with increasing taxon richness increased in the late 1990s, but then sharply declined after windows centred in the early 2000s (Fig. 3e). Functional richness trends were more variable, with the highest trends evident for windows centred on 2000 and 2010, and near net zero trends post-2010 (Fig. 3c). Functional richness trends had an overall tendency to decline (mean change in trends of functional richness = $-5.9\% \text{ y}^{-1}$, 95% CI range: -12% to $+0.1\% \text{ y}^{-1}$). Temporal changes in the proportion of sites with positive functional richness trends were similar to those reported for taxon richness (Fig. 3e, 3g). Trends in abundance (Fig. 3b, 3f) and functional redundancy (Fig. 3d, 3h) changed little over time (i.e. credible intervals overlapping zero in an analysis of the change in estimates over time), although abundance trends tended to decline from windows centred on 2010 until the end of the study period.

- (2) As another way to validate these results, we examined the proportion of sites with positive trends in the biodiversity metrics. We now explain this in the Methods (lines 662-669):

We examined the proportion of sites with positive trends and how this proportion changed through time for our key biodiversity metrics of taxon richness, abundance, functional richness, and functional redundancy. This complements the moving window analysis by asking whether the emerging mean trends are typical of the site-level patterns. This analysis was based only on trend direction and not trend magnitude and hence was less affected by any noise contributed by studies with trends at the extremes. We calculated the 95% confidence intervals for the proportion of sites with positive trends from the trend posteriors.

We now include this analysis in our revised Fig. 3, which depicts the moving window analysis. The proportions for taxon richness and functional richness peaked between 2000 and 2005, suggesting that more sites had positive trends than negative trends during this period. After 2000-2005, this proportion declined to around equal proportions of trends (values of 0.5) for taxon richness, functional richness and functional redundancy, indicating

that increases were fully balanced by decreases in later years. The revised figure is shown below (panels e-h show the proportion of positive trends):

With few exceptions (such as trends for Germany and the Netherlands), similar patterns were typically found when repeating this analysis for each country, indicating that changes in the proportion were not explained by geographic variation in the availability of data (see Figures below and <https://github.com/Ewelti/EuroAquaticMacroInverts> under plots/Online Figures.docx: Figures 24-27).

The figures below show country-level proportions of sites with positive trends (each point represents the proportion within a 10-y moving window) in **taxon richness**:

Country-level proportions of sites with positive trends (each point represents the proportion within a 10-y moving window) in **abundance**:

Country-level proportions of sites with positive trends (each point represents the proportion within a 10-y moving window) in **functional richness**:

Country-level proportions of sites with positive trends (each point represents the proportion within a 10-y moving window) in **functional redundancy**:

(3) We tested an alternative moving window analyses that shows changes in the mean site trends through time using a higher data-inclusion threshold. In this additional sensitivity analysis, we included sites with ≥ 20 sampling years during our full main period of 1990-2020 but removed the first two windows (resulting in windows running from 1992-2001 to 2011-2020) because they included < 200 sites. The results of this analysis are provided below.

This additional moving window analysis includes fewer sites, and thus lower spatial coverage than the moving window analysis presented in the main manuscript. Hence, there is more uncertainty. However, the mean taxon richness trend is positive in the years 1996-2008 and changes to a near-zero trend for the years 2009-2018. Thus, the key result – a decline in the mean trend of taxon richness, including positive mean trends during early years and close to zero mean trends during later years – is robust. Since we are limited by the number of figures allowed in the Extended Data, we include these results on our public github repository: <https://github.com/Ewelti/EuroAquaticMacroInverts> under plots/Online Figures.docx: Figures 21 & 22.

(4) In response to the comment about shorter windows in the last few years, we removed these years (new Figure 3 in main manuscript). Each data point is now based on comparable 10-year moving windows. This change does not affect our main result – that the mean trend in taxon richness has shifted from positive to negative. The mean moving window trends for abundance are now more stable over the years and hence we revised the discussion of this result.

Our *Gains in species richness have come to a halt* section now reads (lines 133-148):

Although trends in taxon richness were generally positive, indicating increases in local richness through time, this effect became weaker over the decades (mean change in trends of taxon richness = $-8.8\% \text{ y}^{-1}$, 95% CI range: -13.6% to $-3.8\% \text{ y}^{-1}$). Trends in taxon richness started declining around 2010 and then levelled off, reaching an average of net zero around 2013 (Fig. 3a), indicating an end to the preceding recovery period. When considering

only the dominant pattern as measured by the proportion of positive trends, the number of sites with increasing taxon richness increased in the late 1990s, but then sharply declined after windows centred in the early 2000s (Fig. 3e). Functional richness trends were more variable, with the highest trends evident for windows centred on 2000 and 2010, and near net zero trends post-2010 (Fig. 3c). Functional richness trends had an overall tendency to decline (mean change in trends of functional richness = $-5.9\% \text{ y}^{-1}$, 95% CI range: -12% to $+0.1\% \text{ y}^{-1}$). Temporal changes in the proportion of sites with positive functional richness trends were similar to those reported for taxon richness (Fig. 3e, 3g). Trends in abundance (Fig. 3b, 3f) and functional redundancy (Fig. 3d, 3h) changed little over time (i.e. credible intervals overlapping zero in an analysis of the change in estimates over time), although abundance trends tended to decline from windows centred on 2010 until the end of the study period.

Additionally, we agree with the reviewer that ongoing recovery is not a reasonable assumption. We have rephrased the text to avoid stating “ongoing recovery” in our manuscript; instead, we explicitly “*hypothesise that freshwater invertebrate community recovery was strongest around the end of the previous century following onset of concerted efforts to mitigate and restore ecosystems, but has slowed in recent years due to diminishing returns of these actions in addition to remaining and new pressures including climate change, land use intensification, and emerging pollutants*” (lines 64–68). This hypothesis is supported by considerable recent research that has documented the extensive, severe, ongoing impacts on Europe's rivers caused by pressures including instream barriers (Beletti et al. 2020) and agriculture (Whelan et al. 2022). Moreover, the most recent report from the European Commission states that 60% of European streams and rivers still fail to achieve ‘good ecological status’ with even higher failure rates in the western European countries in which most of our time series were located, indicating that many sites have not recovered. Even those sites that reached ‘good ecological status’ have not necessarily fully recovered; the highest status class (high ecological status, indicative of unimpacted conditions) is currently only met by <10% of European rivers and streams.

We agree with the reviewer that some of our sites may never have substantially been degraded, but these sites would not contribute to a decelerating trend as their metric values would be similar over time. However, biodiversity metrics may have increased to a level indicative of unimpacted conditions at some sites by the end of the study period, which may also have led to the observed deceleration of the overall recovery process. We revised the following sentences to the 1st. paragraph of “Reviving the recovery” (lines 239ff):

However, gains in taxon richness started to decelerate around 2010, which may reflect recovery of some sites towards the end of the study period and indicate that progress toward recovery has come to a halt at other sites. Most of our sites are monitored under the EU Water Framework Directive (WFD) and 60% of WFD-monitored rivers still fail to reach ‘good ecological status’³⁸. Even at ‘good’ sites, considerable recovery would be needed to reach the ‘high ecological status’, the improvements documented here represent only a partial recovery of European freshwater ecosystems.

2. One of the aspects of the manuscript that could be strengthened is to clarify the degree to which sensitivity of findings are to data variability. With such a big dataset, it is not surprising that the data are not consistent across space, time, and resolution. However, it is challenging to assess whether these differences influence the patterns described. I suggest that there are sensitivity analyses that could help address these potentials.

a. The dataset had a mixed of taxonomic resolutions in it. Many of the response variables--taxon richness, taxon turnover, etc. could be strongly influenced by this resolution. I appreciated Extended Data Fig. 9, that did examine include taxonomic resolution as a factor, but I think it would be helpful to subset the data and rerun the main analyses to directly test this potential (e.g., those that are shown in Fig. 3 and 4).

Author response:

Following the reviewer's suggestion, we have run a series of additional sensitivity analyses using subsets of data with different taxonomic resolutions and estimated the mean trend for each resolution. The results are similar to our previous analysis (in which we included taxonomic resolution as a fixed effect) and are not strongly directionally skewed by taxonomic resolution. Although we expect some differences between estimates due to geographic/environmental differences among sites with different taxonomic resolutions, resolution showed no clear directional effect on the trends (e.g. species estimate > genus estimate > family estimate), reducing concerns about biases introduced by taxonomic resolution. The figure is included below and on our public github repository (<https://github.com/Ewelti/EuroAquaticMacroInverts>, plots: sensitivity: Sensitiv_Split_taxoresTrends.tiff).

This figure shows the overall mean trend for the different biodiversity metrics presented in Fig. 2 split by taxonomic resolution (species level, $n = 762$; genus/mixed taxonomic level, $n = 537$; family level, $n = 517$) in comparison to the overall trend (“all”). Error bars represent 95% credible intervals. Taxonomic resolution did not have strong directional effects on trend estimates, as indicated by generally overlapping error bars.

As suggested, we have also repeated our main moving window analyses using only the subset of sites that reported data at the species level. The results show a similar pattern to our overall moving window analysis, i.e. a decline in the trend of taxon richness over time. This figure is shown below and is included in our public github repository (<https://github.com/Ewelti/EuroAquaticMacroInverts>, plots/Online Figures.docx: Fig. 23).

Additionally, we repeated our analysis of environmental drivers using the 762 sites with species-level data (see figure below). While many of the results are similar in our full analysis of the 1,816 sites and this species-level analysis, precipitation and dams had different effects on biodiversity trends between these two analyses. These differences are likely to reflect geographic bias, as the species-level sites generally had more positive trends for increasing precipitation (mean precipitation trend estimate at species-level resolution sites: 0.83 ± 0.21 SE; mean precipitation trend estimate at all sites: 0.49 ± 0.12 SE) and experienced relatively low dam impacts (mean dam impact score at species-level resolution sites: 0.035 ± 0.2007 SE; mean dam impact score at all sites: 0.081 ± 0.007 SE).

Finally, we examined the number of sites with each taxonomic resolution, across all years and in each moving window period. These figures are shown below and are included in our public github repository (<https://github.com/Ewelti/EuroAquaticMacroInverts>, plots/Online Figures.docx: Fig. 18 & Fig. 20).

Plot of taxonomic resolution in each year of the study period:

Plot of taxonomic resolution in each mean year of each moving window period:

The number of sites at each taxonomic resolution is reasonably comparable across the study years and each moving window period. While the proportion of family-level sites does go down, this is limited to the last two moving window periods, whereas the observed decline in the trend in taxon richness begins much earlier, around a mean year of 2010. Hence, we can identify no evidence for directional change in taxonomic resolution that could bias our results. In addition, our inclusion only of sites that maintained the same taxonomic resolution throughout the time series ensures that within-site trends are not driven by changes in taxonomic resolution.

We have addressed this issue by adding information and links to additional resources to the manuscript (lines 718-721 and lines 727-737):

To check the robustness of our results to analytical decisions, we ran multiple sensitivity analyses for all biodiversity metrics. We tested the effects on trend estimates of: (a) taxonomic resolution, by rerunning meta-analysis models with resolution (family, mixed, and species) as an additional fixed factor...

Scripts for sensitivity analyses are available at:

<https://github.com/Ewelti/EuroAquaticMacroInverts> in the folder R and named HPC_Sensitivity_analysis.R and HPC_Meta_analysis_country_jackknife.

Neither taxonomic resolution nor season had strong directional effects on trend estimates, with error bars generally overlapping, but the sign (positive/negative) of some estimates changed depending on taxonomic resolution or season. Some caution is advised when inferring conclusions from a dataset including different levels of taxonomic resolution or different seasons. However, intra-site sampling was consistently within one season or taxonomic resolution, so within-site trends were not affected by these differences. Patterns

across taxonomic resolutions and sampling seasons were generally similar to those presented in Fig.2 (Extended Data Figs. 9–10).

I also note that figure caption of Extended Data Fig. 9 was a bit skimpy and could have used a bit more detail to aid interpretation.

Author response:

We thank the reviewer for pointing out the lack of information in this figure legend. We have added detail to aid interpretation, and added additional levels of detail in this figure. It now reads (lines 960-963):

Extended Data Fig. 9: Sensitivity of biodiversity metric responses to the taxonomic identification level. Error bars represent 95% credible intervals. Overlapping error bars indicate comparable trend estimates for analyses at species ($n = 762$), genus/mixed ($n = 537$) and family ($n = 517$) taxonomic levels; *Func.*, functional; *Est.*, estimated trend.

b. Similarly, the dataset also had mixed coverage across sites (e.g., Fig 1c). It might be helpful to rerun with the subset of the data with data that was fully spanning the data to rule out the potential that nonrandom patterns of site coverage across space and time contributed to patterns.

Author response:

This is a good point, but no time series spans all sampling years. The plot below shows the number of sites with at least a given amount of sampling years; all 1,816 sites have at least 8 sampling years, but this declines as the number of sampling years increases. Thus, spatial and temporal coverage are a tradeoff that we had to balance in our analyses.

As suggested by the reviewer, we have rerun the moving window analyses using higher inclusion thresholds, which are described in detail in response to the reviewer's first comment. Additionally, below, we show the change in the number of sites per country included in each window of the moving window analysis. This figure shows that our moving window analysis has good spatial representation over time. This figure is included in our public github repository (<https://github.com/Ewelti/EuroAquaticMacroInverts>, plots/Online Figures.docx: Fig. 19).

We have also now conducted a jackknifing analysis to check the influence of data from each country on our overall results. The results show that our main result of a positive trend in taxon richness is robust to the removal of the sites of any individual country. Abundance trends became more strongly positive on removal of data from some countries (e.g. Austria, Ireland, and France). We show the figure below and now mention this at lines 737-740:

Trends of taxonomic richness were robust to one-country removal but abundance trends became more strongly positive on removal of data from some countries, suggesting geographical variability in abundance trends
 (<https://github.com/Ewelti/EuroAquaticMacroInverts>, plots/Online Figures.docx: Fig. 17).

3. The paper emphasized the importance of functional diversity. One challenge for with this was that the approach didn't consider specific functions, but rather aggregated all of the traits, that ranged from body size to life-history. This makes this concept of functional diversity fairly abstract. I also note that many traits had to be inferred based on other species. Thus, while the functional diversity analyses seemed fine, from my perspective they did not add that much to the paper. This is not a very strong suggestion from me, so feel free to disregard.

Author response:

In our study, we sought to complement our taxonomic analysis with analogous functional metrics. This approach was informed by other studies findings that taxonomic changes do not necessarily equate to functional changes which would have implications for ecosystem functions and services (e.g. Brown et al. 2018 Nature Ecology & Evolution). We recognise that there are different ways to represent traits in functional metrics. While we appreciate the idea to investigate individual traits or trait groups that could be more directly related to certain functions, conversely, this should/could also be done for taxonomic diversity, by examining trends in the abundance of individual taxa. Such analyses would be interesting but are beyond the scope of our study.

4. Figure 4. The predictor variables are a mixture of static and dynamic variables, which makes it hard to interpret. Line 196-198 should be revised to be more precise to reflect that some of the relationships could be that the trend differs across the variable (e.g., sites with dams had lower trends in richness through time).

Author response:

All predictor variables were site-specific (i.e., one value per site and hence 'static' in one sense). We assume the reviewer's reference to 'dynamic' relates to some of the climate variables, which included both site-specific mean values (annual averages across the duration of each time series) and site-specific trend estimates in temperature/precipitation (mean annual change in each variable over the time series' period). Mean values were included in the model to account for geographic climate variation (e.g., we found that increases in taxon richness were greater at sites with higher mean temperatures). Climate trends were calculated in a similar way to previous research (e.g. Antão et al. 2020 Nature Ecology & Evolution), by regressing annual mean temperature against year and using the coefficient as an estimate of the trend (in units of mean change in °C per year). The climatic trend variables are intended to test the impacts of climate change on the biodiversity metrics. Analysis of the climate trend effects showed that sites with greater increases in precipitation and temperature tended to have reduced gains in taxon richness. We have added a few sentences as suggested by the reviewer to clarify the interpretation of Figure 4 and the climate covariates (lines 224-230):

For climatic drivers, mean values are the mean long-term values at each site (time series) and account for geographic variation; trends were calculated by regressing annual mean values against year and using the coefficient as an estimate of trend. All response variables are trends (i.e. change in biodiversity metric over time). For example, taxon richness increased over time at sites with higher maximum mean temperatures (tmax mean) but decreased over time at sites where temperatures had higher rates of increase (tmax sl.); b).

5. The paper made some assertions about conservation/management that should be considered. For example:

a. Line 41-43. “Determining the trajectories . . . is a prerequisite. . .” I disagree with this assertion. First, much management is driven by single-species approaches or other conservation values. Second, management can be informed by a variety of knowledge or data, it doesn’t always have to be reactive.

Author response:

We agree with the reviewer and have revised this sentence accordingly. It now reads (lines 46-48):

Determining the trajectories of taxonomic and functional change could inform the development of evidence-based management strategies that address stressors through mitigation, restoration, and conservation.

b. Line 249-250. This felt a bit dramatic and also it wasn’t clear what was precisely meant by “flexible management strategies”.

Author response:

We revised this text. It now reads (lines 297-301):

Progress towards biodiversity goals needs to recognise these changing pressures through flexible strategies to protect and foster Earth’s remaining biodiversity. We call for adaptive environmental management that recognises conservation and restoration objectives as shifting targets that can be modified to adapt to global change and maximise the protection of biodiversity.

6. Analysis. The analysis I think struggled a bit to deal with time, space, and different environmental conditions. I think there probably could have been a more elegant approach that used a hierarchical Bayesian time series approach that then incorporated site-level dynamic (e.g., precipitation) and static (e.g., dams) variables as potential modifiers of the trend. Instead, what was run was a site-specific linear trend analysis and then a post-hoc analysis of the site-specific linear trend estimates. It does not appear that error was fully propagated across these steps (i.e., the posterior distributions of the trends were not used as inputs into the secondary analyses), which is a potential issue. This current approach was ok (and impressive in its own right) but I think it made it a bit hard to provide an integrated perspective into how biodiversity is changing across space and time.

Author response:

When designing our analysis, we considered both a one-stage synthesis model (along the lines the reviewer suggests) as well as our two-stage approach, which first estimates site-level trends and then combines them in a meta-analysis to estimate the overall average trend, which includes all the uncertainty from the first step of the model. We decided on a two-stage approach for the following reasons:

(1) We wanted to ensure that the results of our overall synthesis reflected the average site-level pattern. In a one-stage synthesis, the overall patterns can be driven by a subset of well-sampled sites, which could give a misleading impression of the sample size underlying the overall results. In our two-stage approach, we were able to check that our results were robust to variation in site-level sampling – whether sites were weighted in the synthesis by the confidence of the trend estimate as per a standard meta-analysis model (usually from

well-sampled sites, with lower errors in the trend estimates), or whether sites were unweighted, allowing them all to contribute equally to the overall mean. To incorporate this weighting, our second-stage models included both the trend estimates and their uncertainty (specifically, the standard deviation of the trend estimate posteriors) as a form of measurement error (using the `se` function in the `brms` R package, see Stan Development Team. 2021. Stan Modeling Language User's Guide and Reference Manual, <https://mc-stan.org>). The form of the model was (as described in the Methods: Statistical analysis: (a) Trend analysis):

```
brm(estimate|se(sd_trend_estimate) ~ 1 + (1|study_id) + (1|country), data =  
response_stan, iter = 5000, inits = 0, chains = 4, prior = prior1, control =  
list(adapt_delta = 0.90, max_treedepth = 12))
```

where `prior1` referred to:

```
prior1 = c(set_prior("normal(0,3)", class = "Intercept"))).
```

As shown below, our results were almost identical in the weighted and unweighted analyses, increasing our confidence in the generality of our findings across our study region.

(2) The site-specific trends were the basis for different parts of our later analyses – we used them to calculate the mean trends across all sites and countries, and used them in our analysis of drivers of local trends. Hence, it made sense to estimate the site-level trends only once, so that we had a consistent set of site-level trend estimates for use in our subsequent analyses.

(3) As suggested by the reviewer, we have run an additional one-stage model and compared the results with our weighted and unweighted meta-analysis models. The plot below shows that the results are robust to this analytical choice.

We can see there are arguments both for and against a one-stage or two-stage synthesis, but both ultimately ask the same question and the biggest difference is simply computational.

In response to the reviewer's comment, we have added scripts to the github repository to compare our weighted meta-analysis model of trends with unweighted meta-analysis models and the one-stage synthesis approach (<https://github.com/Ewelti/EuroAquaticMacroInverts>): folder: R, scripts: HPC_onestage_models.R & HPC_Meta_analysis.R and the plot is provided at plots/Online Figures.docx : Fig.16.

We added the following sentences to the Methods section (lines 633-636):

We compared our weighted meta-analysis model of trends with unweighted meta-analysis models, and the one-stage synthesis approach. Overall, these models produced similar trend results (<https://github.com/Ewelti/EuroAquaticMacroInverts>, plots/Online Figures.docx: Fig. 16).

Referee #2 (Remarks to the Author):

Haase et al. describes how river macroinvertebrate communities have changed across a large swathe of Europe in recent decades. It reveals a complex picture in which different facets of taxonomic and functional diversity have changed in different ways through time, among river types and in response to different pressures. Evidence is presented that increases in taxon and functional richness in particular have slowed over the last c. 25 years, with no net change in the last 5 years or so. Assessing functional diversity alongside taxonomic diversity is a valuable addition, revealing more nuanced (and complex) changes.

The core result - the slowing down or cessation of richness gain in the 2000s - fits with earlier studies (e.g. Vaughan & Ormerod 2012, *Global Change Biology* 18, 2184–2194; Outhwaite et al 2020, *Nature Ecol. Evol.* 4, 384-392) but expands it to both a wider range of diversity measures and, most importantly, identifies it at a continental scale suggesting a more general mechanism.

There is little question that this is an important topic and timely assessment, given the evidence of disproportionately large biodiversity declines in freshwaters (e.g. the Living Planet Index for freshwaters) and the ongoing debate around insect/invertebrate declines and evidence for increases in freshwaters cf. terrestrial systems (e.g. van Klink et al. 2020, *Science* 368, 417-420). The authors are to be commended on compiling such a large and impressive international data set, allowing for a potentially important contribution to this debate.

With just a few minor exceptions, the manuscript is well written and the methodology is clear to follow.

Author response:

We thank the reviewer for their positive and constructive comments.

There are a few areas that I think would benefit from some more consideration. I've tried to group these together under related headings:

Recovery

There's an implicit assumption that the overall average increases in diversity represent recovery and that the slowing down of annual changes indicates that recovery has halted. I agree that it is increasingly difficult to find pristine reference streams to help to contextualise changes (line 245), but your analysis raised a couple of issues in my mind:

1. An alternative interpretation of the results might be that recovery is nearing completion when viewed across Europe as a whole (with specific exceptions highlighted in Fig 4), so that there is no further recovery to 'revive' (line 202+). Is it possible to refute this idea with the current data?

Author response:

We suggest that it is possible to refute the idea that recovery is nearing completion, based on considerable recent, data-driven research documenting the extensive, severe, ongoing impacts to Europe's rivers and their biodiversity caused by pressures including instream barriers (Beletti et al. 2020 *Nature*) and agriculture (Whelan et al. 2022 *Science of the Total Environment*). Moreover, legacy stressors (e.g. phosphorus accumulated in soil and

groundwater) will impede complete recovery for some time to come. Accordingly, as shown in Figure 2 (and Extended Data Figure 2), many sites also still show declining trends in taxonomic and functional diversity metrics (red shaded areas of the plots), suggesting incomplete recovery. Furthermore, most streams and rivers investigated in our study were also monitored under the EU Water Framework Directive (WFD). As stated at lines 242-243, 60% of all European rivers still fail to reach the WFD target of ‘good ecological status’ (with even higher failure rates in the western European countries, from which most of our data came). In addition, ‘good ecological status’ indicates slight deviation from unimpacted conditions, meaning that considerable recovery to ‘high ecological status’ is still possible. It is most unlikely that many of the sites have fully recovered, although some sites may have improved considerably, thus slowing the rate of recovery per year. To address this issue (and comparable feedback from reviewer 1), we rephrased the following sentences of the first paragraph of “Reviving the recovery” (lines 239-246):

However, gains in taxon richness started to decelerate around 2010, which may reflect recovery of some sites towards the end of the study period and indicate that progress toward recovery has come to a halt at other sites. Most of our sites are monitored under the EU Water Framework Directive (WFD) and 60% of WFD-monitored rivers still fail to reach ‘good ecological status’³⁸. Even at ‘good’ sites, considerable recovery would be needed to reach the ‘high ecological status’, the improvements documented here represent only a partial recovery of European freshwater ecosystems.

Additionally, we mention in lines 185-186 that:

Alternatively, lower recovery rates for biotic communities in cooler areas could reflect the less severe degradation of northern sites before recovery started.

2. I was interested in your findings that (1) increases in diversity were smaller, or declines were evident, in catchments with cropland or urban land use – exactly the sorts of locations where, if anything, we might expect larger recoveries on the back of historically poorer water quality (Hypothesis 2) – and (2) the percentage increases in EPT abundance and richness were smaller than insects as a whole – wouldn’t you expect recovery from historical stressors to be characterised by larger increases amongst the relatively sensitive EPT taxa compared to insects more generally? Taking both of these together, can you be confident that this really is recovery as opposed to a more general biodiversity change driven by, for example, climate?

Author response:

Regarding (1): Although our hypotheses were informed by the overall improvement in European water quality, we did not expect greater recovery in catchments with high % of cropland and/or urban areas. The overall improvements in water quality mainly reflect the installation of wastewater treatment plants (WTPs) throughout Europe. WTPs reduce point-source pollution – mainly organic pollution, but also inorganic nutrient and chemical pollution. However, diffuse pollution from cropland and urban areas are still affecting freshwater ecosystems with an increasing intensity (Liess et al. 2021 Water Research). Whelan et al. (2022 Science of the Total Environment) recently showed that in Britain, "river quality in catchments with intensive agriculture is likely to remain worse now than before the 1960s". Furthermore, streams draining cropland and urban areas are much more likely to be canalised, culverted, dredged and/or embanked than streams in grasslands or forests, reducing

their ecological resilience to water quality impacts. This is now captured in the revised lines 247-254:

Regardless of the reason for the deceleration, the impacts to Europe's rivers caused by ongoing pressures remain extensive and severe^{37,38}. While our observational data prevent confirmation of underlying causal processes, our interpretation of the overall recovery being a response to increasing water quality aligns with the conclusions of other studies of European freshwater macroinvertebrate time series^{9,39}. Negative effects of poor water quality on biodiversity are supported by our findings that freshwater invertebrate communities downstream of dams, urban areas, and cropland have experienced particularly pronounced biodiversity declines.

Regarding (2): Abundance trends were higher for EPT than insects (+1.02% y⁻¹; +0.66% y⁻¹, respectively), but EPT had lower trends in richness compared to insects (+0.45% y⁻¹; +0.71% y⁻¹, respectively). Although being smaller compared to insects as a whole, EPT richness still increases annually. Over the entire observation period of 53 years, we regard this annual average as considerable, supporting our inference of an underlying recovery process. Further, we did not expect that increases would be greater for EPT than for all insects, because – despite some improvements in water quality – many stressors still persist and negatively affect freshwater biodiversity, particularly more sensitive taxa, such as many EPT. To address the reviewer's comment, we added the following sentence (lines 248-251):

While our observational data prevent confirmation of underlying causal processes, our interpretation of the overall recovery being a response to increasing water quality aligns with the conclusions of other studies of European freshwater macroinvertebrate time series^{9,39}.

Assessment of change through time

I think the first analysis of the paper – the overall mean trends through time – needs some more thought. Presenting a mean value per year implies that change is more or less linear through time, yet the second analysis (= main message of the paper) contradicts that, as does evidence of a strongly nonlinear change through time over a similar timeframe (1970 onwards) from at least one of the constituent countries (UK; Outhwaite et al 2020). Indeed, in the case of the latter, despite large changes in freshwater invertebrates through time, there was negligible net change between the start and end of the time series. An explicit consideration of the potential for nonlinear change since 1968 is important and could provide further insights from your data.

Author response:

We agree that presenting the overall mean trends over time requires careful thought, but respectfully suggest that there is value in presenting both the mean trend and the non-linearity of trends through time. The mean trend is one of the most simple and commonly used summary metrics of change in many monitoring schemes (e.g. bird monitoring schemes in Europe all report mean trend estimates: <https://pecbms.info/trends-and-indicators/species-trends/>) as well as many other recent studies on temporal biodiversity change (e.g., Blowes et al. 2019 Science; van Klink et al. 2020 Science). Hence, by taking a similar approach, we make our findings directly comparable with those of many other studies. Given that we know trends change through time, the mean trend estimate is best interpreted as the long-term mean annual change, averaging out shorter-term temporal fluctuations.

We also agree that it is important to explicitly consider nonlinear change – and respectfully suggest that we do so. We tested for the non-linearity of change, inspired by the findings of recent studies demonstrating changes in the magnitude and direction of trends through time in the freshwater realm (e.g. Outhwaite et al., as mentioned by the reviewer; Baranov et al. 2020, Conservation Biology). While examining non-linearities adds complexity to our study, we felt this analysis was especially important to ensure our paper’s relevance for freshwater policy. For instance, while we show a mean positive long-term trend in richness, we also show that trends have become less positive in recent years via our nonlinear (moving window) trend analysis. In summary, we agree with the reviewer that both careful presentation of mean trends through time and explicit consideration of nonlinear change is required, and we are satisfied that both ways of looking at trends is warranted, because each provides complementary information. We now clarify our rationale in the paper on lines 124-128 as follows:

While overall net trends provide an overview across the entire study period and enable comparison with other long-term biodiversity studies^{17,19,24}, they may mask important shorter-term temporal fluctuations in trends. Therefore, to provide more nuanced, complementary trend information, we used a 10-y moving window approach to examine trajectories of freshwater invertebrate community change over time.

Following on from this, the apparent end of improvements is such an important result: is there a more compelling image that can be used to demonstrate this slow down? e.g. tracking an abundance index through time to demonstrate the plateau. This might also help to address my previous point.

Author response:

As sampling methods differ among sites, we do not use raw abundance or other raw biodiversity values in our analyses. Instead, we used trends in the biodiversity metrics. Focusing on trends facilitates comparison of trends of different biodiversity metrics in different units. We have now added text to the Fig. 3 legend to guide readers’ interpretation of the figure (lines 166-168):

For trend estimates (a–d), points in blue and red areas indicate overall positive (>0) and negative (<0) trend estimates for the given 10-y window, respectively, and grey polygons indicate 80, 90, and 95% CIs.

Also in response to reviewer 1’s feedback, we have now run an alternative analysis of these changes using the proportion of positive trends within 10-y windows over time. This analysis is only based on trend direction and not trend magnitude and hence is less affected by any noise contributed by studies with trends at the extremes. As another way to validate these results, we have now examined the proportion of sites with positive trends in the biodiversity metrics. We now explain this in the Methods (lines 662-669):

We examined the proportion of sites with positive trends and how this proportion changed through time for our key biodiversity metrics of taxon richness, abundance, functional richness, and functional redundancy. This complements the moving window analysis by asking whether the emerging mean trends are typical of the site-level patterns. This analysis was based only on trend direction and not trend magnitude and hence was less affected by any noise contributed by studies with trends at the extremes. We calculated the

95% confidence intervals for the proportion of sites with positive trends from the trend posteriors.

We now represent this analysis in our revised Fig. 3, which depicts the results of the moving window analysis. The proportions for taxon richness and functional richness peaked between 2000 and 2005, suggesting that more sites had positive trends than negative trends during this period. However, since then, this proportion has declined to around equal proportions of trends (values of 0.5) for taxon richness, functional richness and functional redundancy, indicating that increases were fully balanced by decreases in later years. The revised figure is shown below (panels e-h show the proportion of positive trends):

Other recent studies have revealed the influence of time series length upon the resulting average trends (e.g. Leung et al. 2020; Nature 588, 267-271). Are you able to rule out this issue here?

Author response:

To examine the influence of time series length on biodiversity trends, we conducted an additional analysis (see description below). This showed no influence of time series length on our four main biodiversity metrics, as illustrated below. Code to generate plots and plots are included in our github repository (<https://github.com/Ewolti/EuroAquaticMacroInverts>): folder: R, script: HPC_modelchecking.

Below, we show a sensitivity check to examine effects of time series length (i.e. years sampled) on biodiversity trend estimates. Points falling in the white area represent positive trends and those in the grey area represent negative trends. The red line shows the linear regression relationship between time series length and trend estimates and the 95% CI are shown as a pink polygon with dashed lines as borders. We include it in our public github repository (<https://github.com/Ewolti/EuroAquaticMacroInverts> in plots/Online Figures.docx: Fig. 14).

We have added the following text to the Methods to describe this additional analysis and its results (lines 741-746):

We analysed the effect of the number of sampling years in a time series on observed trends using simple linear regression. The number of sampling years did not affect trend estimates of taxon richness ($R^2 < 0.001$), abundance ($R^2 < 0.001$), functional richness ($R^2 = 0.004$), or functional redundancy ($R^2 < 0.001$). Further results from our sensitivity analyses are available at <https://github.com/Ewelti/EuroAquaticMacroInverts>, under plots/Online Figures.docx: Fig. 14.

Not surprisingly, there was very restricted coverage geographically prior to the 1990s. How much does the first ~20 year of the time series influence the final trend estimates?

Author response:

We acknowledge that we have limited geographical coverage before ca. 1990. Most data in this period comes from Germany, and less data from Germany are available in later time series (as shown in Fig. 1). Therefore, a jackknife analysis, in which Germany was removed from the dataset was done and is shown in response to the reviewer's next comment. The removal of Germany did not have strong effects on overall trends in biodiversity metrics (see Figure in response to next comment). Additionally, although most data are from prior to 1990, the moving window analysis explicitly excluded these early years and therefore the results of this analysis are unaffected by this potential geographical bias.

It's good to see that the potential problems of data heterogeneity through time are considered (e.g. line 562). Could you assess this more directly e.g. leaving different countries out of the analysis in the manner of a k-fold cross-validation so see what some of the effects are?

Author response:

As suggested by the reviewer, we have run a jackknife analysis in which each country was sequentially removed, to examine the robustness of our meta-analysis trend results. While there is some variability, our main result for taxon richness is robust to leave-one-out country removal. We provide this plot below and include it in our public repository

<https://github.com/Ewelti/EuroAquaticMacroInverts> in plots/Online Figures.docx: Fig. 17.

Similarly, it's good to see the inclusion of various sensitivity analysis. If I understand correctly, in the case of varying taxonomic resolution, different data sets are used for the different taxonomic levels, presumably representing the resolution at which they were recorded. Would it not be better to use the species-level data throughout, coarsening the taxonomic resolution, rather than using different data sets? This would eliminate other sources of variation and be a purer assessment of the effects of resolution.

Author response:

In any synthesis such as this, there is a trade-off between inclusion criteria and sample size. We defined our inclusion criteria to analyze only sites with fixed taxonomic resolution throughout the time series and a minimum of family-level identification. We chose to use studies reporting data at different taxonomic resolutions to enable inclusion of a considerably greater number of studies (i.e. $n = 1,816$ rather than $n = 762$). But importantly, we only included a study (or site within a study) if data were collected and processed in the same way and the same level of taxonomic resolution was achieved across all sampling years (see section "Statistical analyses"). Hence, although taxonomic resolution varies among sites, it is standardised within sites, and changes in taxonomic resolution therefore do not affect site-level trends. We conducted sensitivity analyses to check whether taxonomic resolution explained differences in trends among studies, and did not detect any strong directional effects of taxonomic resolution on our results. We also note that total abundance, one of our main metrics, is not affected by taxonomic resolution. Differences in abundance estimates when splitting the dataset by taxonomic resolution therefore are due to other factors across the site locations.

Following the reviewer's suggestion, we ran a series of further sensitivity analyses using subsets of data with different taxonomic resolutions and estimated the mean trend for each resolution. The results are similar to our previous analysis, in which we included taxonomic resolution as a fixed effect, and are not strongly directionally skewed by taxonomic resolution. While we expect some differences between estimates due to geographic differences among sites with different taxonomic resolutions, the lack of strong evidence of directional changes (e.g. species estimate > genus estimate > family estimate) reducing concerns about taxonomic resolution biases. The figure is included below and on our public github repository (<https://github.com/Ewelti/EuroAquaticMacroInverts>, plots: sensitivity: Sensitiv_Split_taxoresTrends.tiff).

The figure below shows the overall mean trend results for the different biodiversity metrics (i.e., those presented in Fig. 2) split by taxonomic resolution (species level, $n = 762$; genus/mixed taxonomic level, $n = 537$; family level, $n = 517$) in comparison to the overall trend ("all"). Error bars represent 95% credible intervals. Taxonomic resolution did not have strong directional effects on trend estimates, with error bars generally overlapping. We add some information and suggest caution in interpretation on lines 730-740:

Neither taxonomic resolution nor season had strong directional effects on trend estimates, with error bars generally overlapping, but the sign (positive/negative) of some estimates changed depending on taxonomic resolution or season. Some caution is advised when inferring conclusions from a dataset including different levels of taxonomic resolution or different seasons. However, intra-site sampling was consistently within one season or taxonomic resolution, so within-site trends were not affected by these differences. Patterns across taxonomic resolutions and sampling seasons were generally similar to those presented in Fig.2 (Extended Data Figs. 9–10). Trends of taxonomic richness were robust to one-

country removal but abundance trends became more strongly positive on removal of data from some countries, suggesting geographical variability in abundance trends (https://github.com/Ewelti/EuroAquaticMacroInverts, plots/Online Figures.docx: Fig. 17).

We have also repeated our main moving window analysis using only the subset of sites that reported species-level data. This shows a similar pattern to our overall moving window analysis in terms of a decline in the trend of taxon richness over time. This figure is included in our public github repository (<https://github.com/Ewelti/EuroAquaticMacroInverts>, plots/Online Figures.docx: Fig. 23).

Additionally, we repeated our analysis of environmental drivers using the 762 sites with species-level data (see figure below). While many of the results are similar in our full analysis of the 1,816 sites and this species-level analysis, precipitation and dams had different effects on biodiversity trends between these two analyses. These differences are likely to reflect geographic bias, as the species-level sites generally had more positive trends for increasing precipitation (mean precipitation trend estimate at species-level resolution sites: 0.83 ± 0.21 SE; mean precipitation trend estimate at all sites: 0.49 ± 0.12 SE) and experienced relatively low dam impacts (mean dam impact score at species-level resolution sites: 0.035 ± 0.2007 SE; mean dam impact score at all sites: 0.081 ± 0.007 SE).

Finally, we examined the number of sites with each taxonomic resolution, across all years and in each moving window period. These figures are shown below and are included in our public github repository (<https://github.com/Ewelti/EuroAquaticMacroInverts>, plots/Online Figures.docx: Fig. 18 & Fig. 20).

Plot of overall changes in taxonomic resolution over time:

Plot of changes over time in taxonomic resolution per mean year of moving window periods:

The number of sites at each taxonomic resolution is reasonably comparable across the study years and each moving window period. While the proportion of family-level sites does go down, this is limited to the last two moving window periods, whereas the observed decline in the trend in taxon richness begins much earlier, around a mean year of 2010. Hence, we can identify no evidence for directional change in taxonomic resolution that could bias our results. In addition, our inclusion only of sites that maintained the same taxonomic resolution throughout the time series ensures that within-site trends are not driven by changes in taxonomic resolution.

We have addressed this issue by adding information and links to additional resources to the manuscript (lines 718-721 and lines 727-737):

To check the robustness of our results to analytical decisions, we ran multiple sensitivity analyses for all biodiversity metrics. We tested the effects on trend estimates of: (a) taxonomic resolution, by rerunning meta-analysis models with resolution (family, mixed, and species) as an additional fixed factor...

Scripts for sensitivity analyses are available at:

<https://github.com/Ewelti/EuroAquaticMacroInverts> in the folder R and named HPC_Sensitivity_analysis.R and HPC_Meta_analysis_country_jackknife.

Neither taxonomic resolution nor season had strong directional effects on trend estimates, with error bars generally overlapping, but the sign (positive/negative) of some estimates

changed depending on taxonomic resolution or season. Some caution is advised when inferring conclusions from a dataset including different levels of taxonomic resolution or different seasons. However, intra-site sampling was consistently within one season or taxonomic resolution, so within-site trends were not affected by these differences. Patterns across taxonomic resolutions and sampling seasons were generally similar to those presented in Fig.2 (Extended Data Figs. 9–10).

Other general points

Functional diversity needs some more explanation in the main text, both in terms of technical aspects (e.g. what ‘functional space’ is) and giving an overview of what the different functional measures tell us (+ how they complement one another)

Author response:

We thank the reviewer for their suggestions to make the manuscript more accessible. In response, we have revised and expanded the main text to explain functional diversity in more detail (lines 41-44):

The biological traits of freshwater invertebrates are well characterised, allowing assessment of functional diversity – defined as the range of functional traits of the organisms in a given ecosystem¹⁴ – an important facet of biodiversity that can be used as a proxy for ecosystem functioning^{15,16}.

Furthermore, we have added definitions of “functional diversity” and “functional niche” (the latter including definition of “functional space”) and the sources informing our definitions to Supplementary Table 4, which now provides definitions of all functional diversity related terms used in the manuscript. Functional diversity has been defined as “*the value and range of functional traits of the organisms in a given ecosystem*” (Tilman 2001 Encyclopedia of Biodiversity), and functional niche as “*Represents an n-dimensional hypervolume in functional space, whereby the axes of the functional space are functions or processes associated with different functional attributes (i.e. traits)*” (Rosenfeld 2002 Oikos).

I couldn’t find any mention of potential biases arising from the sampling strategies used in the constituent data sets e.g. were sampling locations selected randomly, focused on particular stream orders, purposefully chosen to monitor the effects of wastewater treatment outfalls, etc? I think it would be extremely difficult to account for such factors in the analysis, but important to provide some consideration of this issue and its potential impacts.

Author response:

The reviewer is correct to note that variability in sampling strategies had potential to introduce bias into our analyses. We describe the collation of time series at the beginning of the Methods section and provide an overview of the sampling locations in Figure 1. Sampling locations were not selected randomly nor restricted to a single stream order, reflecting variability among datasets meeting our six criteria (as stated in the Time series section in the Methods). As further detailed in the Methods, we reduced potential biases arising from some of this variability by accounting for temporal autocorrelation, sampling date, and variation in sampling methods/strategies across studies and countries in our model (lines 70-74):

We use hierarchical Bayesian models to estimate trends and identify drivers of abundance and taxonomic and functional diversity of Europe’s freshwater invertebrate

communities, while accounting for temporal autocorrelation, sampling date, and sampling variation across studies and countries.

Regarding variability in, for example, stream order and the reason for the monitoring (e.g. “purposefully chosen to monitor the effects of wastewater treatment”), we have added information to the Methods section (lines 454-461):

Our compiled time series represent different stream types and stream orders from a large geographical area of Europe. Data were collected for purposes including research projects and regulatory biomonitoring, although detailed information on the purpose is unavailable for some time series. These data were not selected randomly but are collected from available studies meeting our six criteria. As these data were collected from sites exposed to varying and unqualified levels of anthropogenic impacts, we cannot rule out biases arising from unequal representation of sites exposed to different impact levels from severely impacted to least impacted.

Freshwater macroinvertebrates are often viewed as indicators for environmental quality as a whole, not just in river systems because – as you say – they integrate processes occurring across their catchments. Are you able to comment any more on this e.g. the extent to which the observed changes are telling us about the terrestrial environment in addition to the aquatic one?

Author response:

We quantified relationships between freshwater invertebrate change and multiple environmental drivers, two of which represent the terrestrial environment (i.e. cropland and urban land use in the upstream catchment) and four of which represent local climate (Figure 4 and Extended Data Figures 4 & 5). In the section “Environmental drivers of biodiversity”, we provided more detailed information on the linkages between land use and freshwater invertebrates, including evidence that pressures from the terrestrial environment affect freshwater communities. However, as we do not have any further terrestrial data, we are reluctant to speculate on this topic in our manuscript.

Specific minor points

Abstract: Are the net increases across the whole time series or just prior to the mid-2010s? It seems a bit strange say that increases stopped by the mid-2010s, then present the annual increases

Author response:

The overall net increases represent all time series, across all years. Net change across all sites is the most common overarching statistic included in large time-series analyses (e.g. Blowes et al. 2019 Science). In our moving window analyses, we go beyond net trends and examine variation in trends over time, showing increases in the 1990s and 2000s and decelerating trajectories in the 2010s. The description of the moving window results follows the results describing net trends. To better distinguish between the results from long-term mean trends and the moving window analysis, we have rephrased the corresponding sentence to refer to “overall increases” (lines 8-9):

We observed overall increases in taxon richness (0.73% y-1), functional richness (1.03% y-1), and abundance (0.51% y-1).

This revised sentence is immediately followed by (line 10) “*However, these increases primarily occurred before the 2010s, and have since plateaued.*”, and we are satisfied that these two sentences collectively support readers’ understanding of the observed patterns.

line 98, functional diversity: Needs a more accessible definition for a general audience. Also it would be useful to provide some more explanation of the meaning and benefits of the multiple metrics of this diversity e.g. for general readers, the interpretation and importance of functional evenness is unlikely to be obvious

Author response:

We thank the reviewer for identifying this opportunity to make the manuscript more accessible. In response, we have revised lines 106-109 to provide an accessible definition for a broad readership:

Functional richness, which quantifies the functional space filled by a community, increased on average by 1.03% y⁻¹ (0.99 probability of increase; Fig. 2e). Functional redundancy, a measure of overlap in functional trait space, had no strong trend (+0.03% y⁻¹, 0.64 probability of increase; Fig. 2f).

We have also revised Supplementary Table 4, as detailed above in response to the reviewer’s first “other general point”. We selected these metrics both as informative metrics in their own right and also “*to align with the taxonomic diversity metrics*”, which we now clarify at lines 506-507. We respectfully suggest that further consideration of the benefits of each metric is beyond our scope.

line 120, non-linear trajectories. Could non-linearity also be generated by falling recovery rates as recovery/gains from mitigation measures are exhausted?

Author response:

The reviewer is right to infer that falling recovery rates may partly reflect the diminishing returns of mitigation measures, leading to non-linearity. This is one of our core arguments when interpreting our results (see section “Biodiversity recovery has come to a halt”). We slightly rephrased our second hypothesis to pick up this aspect. It now reads (lines 63-68):

We further hypothesise that freshwater invertebrate community recovery was strongest around the end of the previous century following onset of concerted efforts to mitigate and restore ecosystems, but has slowed in recent years due to diminishing returns of these actions in addition to remaining and new pressures including climate change, land use intensification, and emerging pollutants.

However, we suggest that mitigation measures have not been exhausted, as per our response to the reviewer’s first “Recovery” comment, above.

line 147 – urban areas are also likely to be a proxy for point source pollution (cf. diffuse)

Author response:

We agree and have revised this sentence as follows (lines 174-177):

Here, we show that the percent of upstream urban areas and cropland (both sources of pollution and causes of habitat degradation), climate, and dam impacts can all be linked to trends in taxonomic and functional metrics representing Europe's freshwater invertebrate communities (Fig. 4; Extended Data Figs. 4–5).

line 152, percent increases in temperatures will only be meaningful if expressing the change in Kelvins. Do you need to present it in this way? Degrees C would be more interesting and informative

Author response:

We thank the reviewer for this suggestion and have converted these values into degrees C and mm per year.

line 158, lower recovery rates in cooler areas due to less severe degradation. Good point - it would be interesting to see how change was related to existing biodiversity e.g. smaller/no gains in functional richness in streams which were already rich

Author response:

We agree that this is an interesting topic for future research. Such an analysis would require stream-type-specific baseline information (i.e. the number and identities of taxa under reference conditions), which would be difficult, if not impossible, to retrieve and which would require analyses beyond the scope of the present study.

line 166. Precipitation could also play a role via varying water quality (runoff, dilution...)

Author response:

We agree with the reviewer and have revised this sentence as follows (lines 193-195):

Precipitation can influence invertebrate communities and their functioning by altering flow regimes (and thus water quality and temperature, via changes in runoff, discharge, and dilution) and food availability⁶.

line 215, positive effects of rising temperatures at high latitudes. Why not test that directly here?

Author response:

In this sentence we are not referring to variation within our study but comparing our results (trends in Europe) to previous work showing freshwater macroinvertebrate declines in a warmer (subtropical) region (Romero et al. 2017 Biology Letters). We have revised this sentence (lines 259-263):

The positive effects of higher mean temperatures on invertebrate richness are likely a result of the lower initial degradation in northern European countries but may also reflect the overall cooler temperatures in European countries on average, whereas decreases are currently expected in freshwaters of warmer bioregions, such as tropical regions, which are not represented in our study⁴⁴.

line 505 – say why you used gls in this instance

Author response:

We have revised this section to use a similar analytic framework (i.e., Bayesian regression models) to calculate the climate trends, for consistency with other analytical methods.

lines 592+ – more details needed to describe the sensitivity analyses

Author response:

We have added additional details to this section as suggested by the reviewer. The sensitivity section now reads (lines 718-746):

To check the robustness of our results to analytical decisions, we ran multiple sensitivity analyses for all biodiversity metrics. We tested the effects on trend estimates of: (a) taxonomic resolution, by rerunning meta-analysis models with resolution (family, mixed, and species) as an additional fixed factor, (b) sampling season, by rerunning meta-analysis models (described above in section (a) Trend analysis) with season (winter, spring, summer, and fall) as an additional fixed factor, and (c) country, using a jackknife resampling analysis in which the meta-analysis was rerun after sequentially removing countries. Each sensitivity model included metrics as response variables weighted by the inverse of their uncertainty. Model predictors included an intercept, and random effects of study identity and country. Scripts for sensitivity analyses are available at: <https://github.com/Ewelti/EuroAquaticMacroInverts> in the folder R and named HPC_Sensitivity_analysis.R and HPC_Meta_analysis_country_jackknife.

Neither taxonomic resolution nor season had strong directional effects on trend estimates, with error bars generally overlapping, but the sign (positive/negative) of some estimates changed depending on taxonomic resolution or season. Some caution is advised when inferring conclusions from a dataset including different levels of taxonomic resolution or different seasons. However, intra-site sampling was consistently within one season or taxonomic resolution, so within-site trends were not affected by these differences. Patterns across taxonomic resolutions and sampling seasons were generally similar to those presented in Fig.2 (Extended Data Figs. 9–10). Trends of taxonomic richness were robust to one-country removal but abundance trends became more strongly positive on removal of data from some countries, suggesting geographical variability in abundance trends (<https://github.com/Ewelti/EuroAquaticMacroInverts>, plots/Online Figures.docx: Fig. 17).

We analysed the effect of the number of sampling years in a time series on observed trends using simple linear regression. The number of sampling years did not affect trend estimates of taxon richness ($R^2 < 0.001$), abundance ($R^2 < 0.001$), functional richness ($R^2 = 0.004$), or functional redundancy ($R^2 < 0.001$). Further results from our sensitivity analyses are available at <https://github.com/Ewelti/EuroAquaticMacroInverts>, under plots/Online Figures.docx: Fig. 14.

Referee #3 (Remarks to the Author):

The extensive data set analysed in this paper is a considerable strength – I know of no other paper with equivalent depth of coverage of biodiversity data from a large biogeographic region. Its focus on invertebrates is laudable, and a welcome change from analyses that consider only vertebrates. The paper’s take home message – of initial recovery, subsequently halted – has important policy implications. As far as I am aware, this is an original finding.

Author response:

We thank the reviewer for their positive comments, and especially for highlighting our unique coverage of taxa and biogeographic regions.

With papers of this sort, the details of the analyses are paramount. I found it difficult, in places, to fully understand what was done, or what the raw data looked like. Line 370 states that the authors used time series that included the whole invertebrate community. But we are not told which groups were sampled at each site. This information should be included in Supplementary Table 2.

Author response:

All datasets reported samples of whole communities, i.e. including all sampled macroinvertebrates. Between 13 and 516 taxa from 47 (order level or higher) groups were sampled per site. Given the large number of times series (1816) adding the number of taxonomic groups for each time series in an accompanying table (such as Supplementary Table 2) would be an unfeasible length. Nonetheless, information on the individual taxa in each sample will be provided upon publication of the raw data if the manuscript is accepted. In response to the reviewer’s comment, to enable readers to better understand the taxa present in sampled assemblages, we have added the number of major taxonomic groups and have provided examples of the common groups included in our study:

Lines 448-451:

Between 13 and 516 taxa were sampled per site across all sampling years. Communities from 42% of sites were identified to species, 30% were identified to mixed (species-to-family) taxonomic levels, and 28% were identified primarily to family. In total, 2,648 taxa from 959 genera, 212 families, and 47 groups (primarily orders) were recorded.

Lines 433-435:

... (including all sampled macroinvertebrates, e.g. Coleoptera, Crustacea, Diptera, Ephemeroptera, Hirudinea, Mollusca, Odonata, Oligochaeta, Plecoptera, Trichoptera, Tricladida),...

Biodiversity analyses are notoriously vulnerable to variation in sampling effort. The authors note that they standardised the data by including only one sampling season (line 378). They further note (line 398) that they computed rarefied taxon richness. It is not clear whether they mean a sample-based rarefaction (as implied above) or some other form of rarefaction. This leads me to wonder to what extent change in the various metrics is underpinned by shifts in abundance. Clearly a more comprehensive description of data handling and analysis is needed here. At present there is insufficient information to replicate the analyses.

Author response:

All the data within each site were collected using standardised methods. Moreover, we only included sampling sites for which sampling effort was consistent through time; as such, we did not need to control for variation in sampling effort to estimate site-level trends. Consequently, our calculation of rarefied richness did not require standardisation of sampling effort. Instead, we present individual-based, not sample-based, rarefied richness, along with observed richness. This additional metric provides a more complete picture of freshwater invertebrate community change, as also shown by other analyses of standardised time-series data (e.g., Newbold et al. 2015 Nature, Blowes et al. 2020 Journal of Applied Ecology, McGlinn et al. 2019 Methods in Ecology and Evolution, and proposed as part of the Measurement of Biodiversity framework: <https://pubmed.ncbi.nlm.nih.gov/35869831/>). We used individual-based rarefied richness to help understand whether changes in richness reflect changes in abundance, and hence can be explained by simply the presence of more or fewer individuals. Alternatively, if richness trends and rarefied richness trends differ, then richness trends cannot be explained by changes in abundance and additional factors are at play. We revised the manuscript to clarify the calculation of rarefied richness (lines 470-473):

As sampling effort was standardised within time series prior to metric calculation, individual-based rarefied richness was used to estimate the number of taxa per given number of individuals, based on the lowest number of individuals per sampling year in each time series¹⁷.

With regard to functional diversity I wondered how the authors dealt with taxa at different developmental stages (given that the functional roles of aquatic invertebrates can shift markedly as they mature).

Author response:

The trait information in the databases that we used for our trait analyses describe the aquatic stage or stages of the respective species. In non-insects (e.g. crustaceans), both juveniles and adult life stages are aquatic, and the databases thus provide a single value to represent all stages. In insects, commonly only the juvenile stage is aquatic while most adults are terrestrial. The primary exception, at least within our dataset, is the Coleoptera, for which trait information is provided for both adults and larvae. For almost all Coleoptera in our dataset, the trait values for adults and larvae are identical. Nonetheless, we acknowledge that this approach does not account for potential differences in e.g. different juvenile instars or sexes, because such trait information is unavailable for most taxa, a constraint that also applies to many other marine and terrestrial taxa groups. We have added the following sentence to the manuscript to address this comment (lines 483-485):

For the taxa with multiple aquatic life stages (primarily beetles), whenever available from the trait databases, functional roles were assigned for each life stage, otherwise adult traits were used.

The authors used a variety of statistical methods to estimate diversity trends. A headline result is that gains in biodiversity have been reversed in the last decade. The moving window analysis appears to be key here, but as the authors mention, there are many caveats involved. A complicating factor (evident from Figure 1c) is the much higher number of sites post 2000. The authors note this, and have tried to deal with it by focussing the analysis on a subset of sites. Nonetheless, reassurance that change in sampling effort has not influenced the results is

needed, and that either increases in the types of sites included in the data set, or spatial biases, cannot account for the conclusions.

Author response:

Below, we show the number of sites per country in each window of the moving window analysis. This figure shows that our moving window analysis has good spatial representation over time. This figure is included in our public github repository

(<https://github.com/Ewelti/EuroAquaticMacroInverts>, plots/Online Figures.docx: Fig. 19).

In response to the reviewer’s comment (and comments made by reviewers 1 & 2), we have also rerun our moving window analyses using a more robust inclusion threshold. We include the results of these additional analyses on our public github repository: <https://github.com/Ewelti/EuroAquaticMacroInverts> under plots/Online Figures.docx: Figure 22.

In this additional sensitivity analysis, we included sites with ≥ 20 sampling years during our full main period of 1990-2020 but removed the first two windows (resulting in windows running from 1992-2001 to 2011-2020) because they included < 200 sites. The results of this analysis are provided below.

These higher-threshold moving window analyses include fewer sites, and thus lower spatial coverage than the moving window analysis presented in the main manuscript. Hence, there is more uncertainty. However, the mean taxon richness trend still declines over time. The key result of a decline in the mean trend of taxon richness – positive mean trends during the early years and close to zero mean trends during the later years – are robust.

From a conceptual standpoint, Figure 3 is interesting. It shows 4 metrics for which the earlier values are higher than the later ones (but with the credible intervals overlapping zero to a large extent). (Extended data Figure 3 is the equivalent plot for 4 more metrics, but – the final data point apart – are much less convincing of a shift in diversity). Should we expect gains in diversity to be sustained continuously? Surely these systems have some ‘natural’ baseline level of diversity around which levels will fluctuate but not necessarily show directional

change? To what extent are the observed trends indicative of a flattening off in recovery as it reaches some equilibrium, as opposed to a reversal of gains? Is the shift a consequence of climate change? - the paper notes that warming is associated with an increase in diversity (a result that tallies with other work in the field).

Author response:

Regarding the 1st question (“Should we expect gains in diversity to be sustained continuously?”), we recognise that biodiversity metrics tend to fluctuate around a baseline even if environmental conditions are stable. In our case, we observed increases in taxon richness and abundance by 0.7% and 0.5% annually. Considering the length of this time period (>20 years) and such levels of increase, we suggest that our results should not be regarded as fluctuations but rather indicate directional changes. While gains in biodiversity cannot continue indefinitely in any system, more than 90% of Europe’s rivers and streams have not reached high ecological status under the Water Framework Directive and 60% have not reached a good ecological status. As such, we do not interpret our results as evidence of full recovery with fluctuations in diversity around a natural baseline.

Regarding the reviewer’s 2nd and 3rd questions (“Surely these systems have some ‘natural’ baseline level of diversity around which levels will fluctuate but not necessarily show directional change? To what extent are the observed trends indicative of a flattening off in recovery as it reaches some equilibrium, as opposed to a reversal of gains?”). The final results from our moving window analysis do not show flattening of recovery of biodiversity within individual sites but rather a shift in the biodiversity trends across sites to have a mean response that is closer to zero. The estimated mean response could either reflect a larger proportion of sites having declines in biodiversity or a greater proportion of sites having trends closer to zero. To further investigate these changes, we examined the proportion of positive moving window trends. The proportion dropped to near or below 0.5 around 2012 for both taxon richness and abundance, supporting the findings presented in Figure 3. The proportions for taxon richness and functional richness peaked between 2000 and 2005, suggesting that more sites had positive trends than negative trends during this period. As another way to validate these results, we also examined the proportion of sites with positive trends in the biodiversity metrics. We now include this analysis in our revised Fig. 3, which depicts the results of the moving window analysis. The revised figure is shown below (panels e-h show proportion of positive trends):

In response to the 5th question (“Is the shift a consequence of climate change?”): Multiple concurrent, interacting factors are likely to be responsible for the observed increases in taxonomic and functional metrics, including area-wide installations and upgrades of wastewater treatment plants and climate warming. As shown in Figure 4, increasing temperature (i.e. tmax slope) is negatively correlated with trends in taxon richness (Fig. 4a) and abundance (Fig. 4b). This could indicate that climate warming is limiting further recovery. While we caution against overinterpreting our data, we direct the reviewer to our consideration of this topic at lines 263-266:

However, the declines in taxon richness, abundance, and functional richness in communities experiencing greater rates of warming are worrying and are likely to worsen as temperatures continue to rise, and as climatic extremes including summer droughts and heatwaves become more common⁴⁵.

Figure 4 seems to be an analysis of overall trends in relation to various drivers, rather than one specifically focused on shifts in trends in the last decade. It left me wondering to what extent these drivers account for the reported downturn in diversity. Can these not be explicitly investigated?

Author response:

The reviewer is correct that our driver analysis seeks to characterise the variation in long-term trends among sites, and does not explain the temporal shifts in site-level trends in the last decade. Our analysis is limited by the availability of driver data, especially information on temporal changes in driver intensities. To characterise shifts in the last decade, we would need site-level information describing temporal changes in water quality. Such information isn’t available for most of the sites (and across time series durations) in our study. However, recent research indicates that water quality improved after the implementation of clean water and sewer legislation in the late 1980s (e.g. Vaughan and Ormerod 2012 Nature Communications, Bouraoui & Grizzetti 2011 Science of the Total Environment, European Parliament, Council of the European Union Common Implementation Strategy for the Water

Framework Directive (2000/60/EC), 2009, <https://www.eea.europa.eu/data-and-maps/indicators/freshwater-quality/freshwater-quality-assessment-published-may-2>), but the past decade has seen little improvement and potentially even declines in water quality (Wolfram et al. 2021 Environment International). This is a likely key driver for the change in trends identified by our analysis. We could seek to explain the shift in the last decade using other variables for which temporal data are available across all sites (e.g. temperature and precipitation) but we would not feel confident interpreting the effects of any regression model without accounting for the variation caused by water quality. This would lead to ‘omitted variable bias’ when making any attempt to infer causality. As data, both for biodiversity and environmental variables, become better organised in the future, we hope such an analysis will become possible.

A general observation I had is that the paper can’t quite decide if it is an analysis of overall trends, or is focussed mainly on the break point in trends. It contains many very interesting analyses and is potentially influential, but in places it needs to provide more information, and additional analyses, to support the conclusions it draws.

We thank the reviewer for highlighting this opportunity to clarify our overall approach. We intentionally used two different approaches. Our first approach (overall mean trends: including all time series covering the entire study period from 1968 to 2020) is very common and allows for comparisons with other studies (e.g. Blowes et al. 2019 Science, Pilotto et al. 2020 Nature Communications). Our second approach (moving window analysis) characterises the nonlinearity in temporal biodiversity change and has rarely been included in other large-scale synthesis studies, probably for the reason the reviewer highlights: because it complicates interpretation of results. Previous large-scale synthesis have been criticised for oversimplifying patterns of biodiversity change (e.g. Bellard et al. 2022 Nature Communications); hence, we aim to provide a more complete, if complex, picture in our analysis. In response to this comment, we have revised the text to clarify that we could not explain the break point in trends with our driver analysis (lines 705-708):

Our analysis of drivers includes all time series and, due to limitations caused by spatial and temporal heterogeneity in data coverage, cannot be used to infer causality in moving window analyses.

Referee #4 (Remarks to the Author):

This report concerns only the statistics aspect of the article, I do not comment on the other scientific aspects.

Overall, this paper uses statistical models to analyse time series data on riverine invertebrate communities. The statistical models they used was based on a two stage approach: first stage obtains a trend from time series separately for each time series, then the second stages use these trends as response to combine spatially separate time series to obtain an estimate of an overall trend. The approach does not account for all the uncertainty in the estimates, potentially leading to an over-confidence of changes in trend.

Author response:

In response to the reviewer's comment, we realise that we had not sufficiently explained our methods: we included the uncertainty of the trend estimates (specifically the standard deviation of the site-level trend posteriors) in our meta-analysis model. This is akin to a measurement error model using the brms package. The form of the model was (as described in Methods: Statistical analysis: (a) Trend analysis):

```
brm(estimate|se(sd_trend_estimate) ~ 1 + (1|study_id) + (1|country), data =  
response_stan, iter = 5000, inits = 0, chains = 4, prior = prior1, control =  
list(adapt_delta = 0.90, max_treedepth = 12))
```

where prior1 referred to:

```
prior1 = c(set_prior("normal(0,3)", class = "Intercept"))).
```

As we show above in response to reviewer 1's comment 6, we get almost identical results in a one-stage synthesis in which site-level error is implicitly propagated, which shows that our results are robust to how error is included, and also to whether studies are weighted or unweighted. Scripts for these analyses is provided in our github repository (<https://github.com/Ewelti/EuroAquaticMacroInverts>): folder: R, script: HPC_onestage_models.R & HPC_Meta_analysis.R. The figure is provided below and at plots/Online Figures.docx : Fig.16.

In contrast to the reviewer’s suggestion that our approach might lead to overconfidence in trends, our approach in fact tends to be conservative for many variables, because the confidence intervals are generally wider in a two-stage approach compared to a one-stage approach.

While we did initially consider both a one-stage and two-stage approach, we decided to continue with the two-stage approach, primarily to ensure that our overall synthetic mean did reflect a mean across all sites and not just a mean across the subset of sites that were especially well-sampled. We can see there are arguments both for and against a one-stage or two-stage synthesis, but both ultimately ask the same question and the biggest difference is simply computational. The above plot shows that the results are robust to this analytical choice.

Also, our paper is about both the mean trend and variation in trends across sites. While the mean trend has been more frequently highlighted by previous studies analysing temporal change in biodiversity, the mean trend becomes less informative as variance increases. Hence, in Fig. 2, we show the results of both stages of our analysis: the site-level trends from the first stage and the overall mean from the meta-analysis model used in the second stage.

To better explain our approach, we have added the following text to the Methods (lines 633-636):

We compared our weighted meta-analysis model of trends with unweighted meta-analysis models, and the one-stage synthesis approach. Overall, these models produced similar trend results (<https://github.com/Ewelti/EuroAquaticMacroInverts>, plots/Online Figures.docx: Fig. 16).

Overall, I find that the statistical writing can be improved, expressions (particularly conclusions) can be made more precise and the details of the statistical models should be provided. As the manuscript stands now, I found it difficult to follow the statistical techniques used. Below I give specific comments and some suggestions on how to make the statistics sections easier to read.

L77, when the authors say taxon richness increased by a net mean of 0.94%, what do they actually mean by this? Is it the differences between the first and last observed values, or the fitted estimate of trend? Why not provide confidence intervals to indicate whether this change is statistically significant?

Author response:

Here, we took the fitted estimate of the trend (i.e., the raw coefficient for the year effect) and transformed it into a % change to help communicate our results. Hence, this % value indicates that, on average across all sites and years, taxon richness increased on average by 0.73% per year (the value has changed from 0.94% to 0.73% due to a change in the error term used in our revised meta-analysis model). We calculated this overall mean % by dividing the fitted trend estimate for taxon richness by the average annual value of taxon richness and then multiplying by 100. Multiple credible intervals (80%, 90%, and 95%) and the full distribution of responses are provided in Figure 2. We have removed the confusing wording of “net mean” and now explain this in the Figure legend 2 as follows (lines 120-121):

Trend estimate values are provided as percent change per year on each panel.

L79-86 “posterior distributions revealed 99% and 99.3% probabilities of a net increase...”.

- firstly, percentages are not probabilities, the authors mean 0.99 probability of;
- secondly what is the meaning of this sentence? Posterior distributions of what? What is the confidence interval for these probabilities?

Author response:

We have changed each “% probability” to probability, as suggested by the reviewer.

The probability is calculated as the proportion of the posterior distribution above or below zero, i.e., the probability that the mean trend is increasing or decreasing. These probabilities do not have confidence intervals. We now state this as follows (lines 624-626):

For each response metric, we calculated the proportion of the posterior distribution of the mean trend estimate across all sites (i.e. the overall LMM intercept) above or below zero, i.e. the probability that the mean trend is increasing or decreasing.

L117, the use of a 10-year moving average window is not well justified, as we know, the width of the window has large impact on the results.

Author response:

To address this query, we have added the following justification (lines 641-643):

A 10-y window was chosen following current recommendations regarding times series length^{83,84} and six was chosen as the number of sampling years covering >50% of each 10-y period.

Trends for shorter-lived animals like freshwater invertebrates can be measured in shorter time series than those of long-lived animals or plants (Cusser et al. 2021, Ecology Letters), suggesting a shorter (e.g. 10-y) window is an appropriate choice for this study system.

L116, are the conclusions in this section based on visual observations of the 10-year moving average smoothing? Is there any statistical tests that were carried to test for statistically significant decline? e.g., in Fig 3a.

Author response:

In response to the reviewer's questions, we have added a direct statistical test of the linear change over time in the trends in the moving window analyses. We modeled the effect of year on moving window trends using a simple Bayesian model of:

$\text{Moving_window_trend} | \text{se}(\text{Moving_window_trend_error}) \sim \text{Moving_window_year}$

to test for a directional change with time in our trend analysis. The script for this analysis is included in our github repository (<https://github.com/Ewelti/EuroAquaticMacroInverts> in folder R: MetaMeta_movingwindowEsts_Yr.R). In these models, we found declines in taxon richness trends (Estimate = -0.0192, SD = 0.006) with 95% CI not overlapping zero. Functional richness trends also tended to decline (Estimate = -0.0004, SD = 0.0002), with 95% CI barely overlapping zero. No directional change was found for abundance trends (Estimate = 0.00009, SD = 0.0002) or functional richness trends (Estimate = -0.00003, SD = 0.00003) as 95% CIs overlapped zero. We now include estimates converted into change in trends per year in the main manuscript (lines 133-148):

Although trends in taxon richness were generally positive, indicating increases in local richness through time, this effect became weaker over the decades (mean change in trends of taxon richness = $-8.8\% \text{ y}^{-1}$, 95% CI range: -13.6% to $-3.8\% \text{ y}^{-1}$). Trends in taxon richness started declining around 2010 and then levelled off, reaching an average of net zero around 2013 (Fig. 3a), indicating an end to the preceding recovery period. When considering only the dominant pattern as measured by the proportion of positive trends, the number of sites with increasing taxon richness increased in the late 1990s, but then sharply declined after windows centred in the early 2000s (Fig. 3e). Functional richness trends were more variable, with the highest trends evident for windows centred on 2000 and 2010, and near net zero trends post-2010 (Fig. 3c). Functional richness trends had an overall tendency to decline (mean change in trends of functional richness = $-5.9\% \text{ y}^{-1}$, 95% CI range: -12% to $+0.1\% \text{ y}^{-1}$). Temporal changes in the proportion of sites with positive functional richness trends were similar to those reported for taxon richness (Fig. 3e, 3g). Trends in abundance (Fig. 3b, 3f) and functional redundancy (Fig. 3d, 3h) changed little over time (i.e. credible intervals overlapping zero in an analysis of the change in estimates over time), although abundance trends tended to decline from windows centred on 2010 until the end of the study period.

L151 – 167, the affects of climate on the invertebrate communities discussed here are based on which model? and when reporting the results, it is best to give 95% confidence interval and state which increases or decreases are statistically significant, SE alone does not tell you that.

Author response:

Climate variables are included in the models of driver effects on biodiversity metrics (see Methods: Statistical analysis: (c) Analysis of environmental drivers and Fig. 4). We have added reference to Figure 4 to this section to clarify the discussed results. Figure 4 provides three levels of credible intervals (the Bayesian equivalent of confidence intervals) including 95% intervals.

L416-428, it was described the method used to impute missing trait data. However, it seems to me that only 15.9% are available at the original identification level, the rest are proxies. Even if the genus-level trait data is a reasonable substitute, there's still more than 50% missing. What assurances can the authors demonstrate that their approach for imputing the missing data does not lead to potentially wrong conclusions?

Author response:

Trait data, in particular invertebrate community trait data, are often gap filled and imputation is required (e.g. Penone et al. 2014 *Methods in Ecology and Evolution*, Taugourdeau et al. 2014 *Ecology and Evolution*). Information available at other taxonomic levels are routinely used to fill gaps. Many selected traits are taxonomically (phylogenetically) conserved. For example, the trait “respiration type” describes whether the taxon has gills, a tegument, plastron, spiracles, etc. – trait modalities that tend to be conserved at family to order levels. Perhaps our least taxonomically conserved trait is “maximum potential body size”, but even this trait is expected to be correlated with taxonomy. While missing trait values are not ideal, taxonomic imputation is a sufficient gap-filling approach for the types of traits included in these analyses (Schmera et al. 2017). For example, Sotomajor et al. (2021 *Diversity and Distributions*) explicitly tested genus-level versus aggregated family-level information and found 82% similarity of results.

L515, the authors should precisely write down (using equations) the statistical model used, both for the first stage and the second stage using LMM and the trend estimate from the first stage as response.

Author response:

We thank the reviewer for this suggestion which we believe will make our analysis more transparent. We have added the models to our Methods at lines 600, 606-607, 619-621, 654-656, 699-702, and at lines 753-754:

Annotated R code is available on GitHub:
<https://github.com/Ewelti/EuroAquaticMacroInverts>

L523, it was mentioned that an AR(1) model was used, can the authors justify this choice?

Author response:

Because we analyzed time series data, we wished to account for temporal autocorrelation. Modeling autocorrelation with an AR(1) component is often used in ecological trend models (e.g., Arkilanian et al. 2020, Turchin 1990 *Nature*) because most of the autocorrelation is explainable with this lag, especially with invertebrates with short generation times. We now provide a supporting reference (Ziebarth et al. 2010 *Ecology Letters*).

L536, it is stated that the probability of mean increase is calculated from the posterior distribution, how is it calculated?

Author response:

The probability of mean increase is calculated as the proportion of posterior distributions above or below zero. We have added this information (lines 85-86):

Posterior distributions (i.e. the probability of the mean trend being above or below zero) revealed....

L539, what priors were actually used? A more precise expression should be given here.

Author response:

We have revised the entire paragraph and added additional information about priors (lines 628-633):

We used default priors except for the trend estimates, for which we selected a narrower prior to diminish the influence of biologically unrealistic trend estimates. Specifically, we used normally distributed priors with a mean of zero and a standard deviation of 10 (for site-level trends) or 3 (for mean site-level trends). For the analysis of environmental drivers, we used regularising horseshoe priors on the environmental covariates to limit overfitting.

L541, moving-window analysis, as I mentioned before, why a 10 year width of window is chosen? Additionally, why not use something like the splines, which can mitigate some of the issues mentioned by the authors in terms of boundary bias.

Author response:

To justify our choice of a 10-y window, we have added the following (lines 641-643):

A 10-y window was chosen following current recommendations regarding times series length^{83,84} and six was chosen as the number of sampling years covering >50% of each 10-y period.

To eliminate boundary bias in the revised manuscript, we have removed the points in the final windows of the moving window analysis so that all windows are 10 years with a final window of 2011-2020.

L560 (and later), when the authors say, “weighted by their inverse error to account for uncertainty”, what do they mean by this? How is this done?

Author response:

We used the package brms for this analysis and included the standard deviations of the trend estimate posteriors as a form of measurement error on the trend estimates. The connection between measurement error models and meta-analysis models is discussed in the Stan User’s Guide and Reference Manual (Stan Development Team. 2021. Stan Modeling Language, https://mc-stan.org/docs/2_19/stan-users-guide/measurement-error-and-meta-analysis.html; see also: <https://vasishth.github.io/bayescogsci/book/ch-remame.html>). This means that the influence of trend estimates for individual sites/time series on the overall estimated mean trend depends on the level of certainty. This is typical in a meta-analysis model (Koricheva et al. 2013 Handbook of Meta-analysis in Ecology and Evolution). For another example, see

Doherty et al. (2021, Nature Ecology and Evolution). Scripts are available in our github repository (<https://github.com/Ewelti/EuroAquaticMacroInverts>).

The form of our meta-analysis model is:

```
fit1 <- brm(estimate|se(sd) ~ 1 + (1|study_id) + (1|country),  
  data = response_stan, iter=5000, inits = 0,  
  chains = 4, prior = prior1,  
  control = list(adapt_delta = 0.90,  
    max_tredepth = 12))
```

In addition, we have now repeated all our analyses as a one-stage model in which error was implicitly propagated and we obtained highly comparable results. A comparison of model estimates between model types is available at <https://github.com/Ewelti/EuroAquaticMacroInverts> in: plots/Online Figures.docx: Fig. 16.

L585, should R-hat value be around 1? (why <1.1?)

Author response:

R-hat is a measure of the overlap of the different MCMC chains, so a value of 1 would mean multiple chains converge to the same posterior estimates. While 1 is ideal, R-hat <1.1 is the traditional cutoff to indicate a model with sufficient convergence (Kéry and Schaub, 2012). We now include this reference in the manuscript (cited on line 711).

Reviewer Reports on the First Revision:

Referees' comments:

Referee #1 (Remarks to the Author):

This paper presents the results of an extensive dataset on freshwater invertebrate biodiversity and its change over the last several decades. They discovered that biodiversity has tended to increase, but that these increases have plateaued recently. Further, the analyses revealed that broad-scale drivers such as dams are associated with less positive trends in biodiversity. These patterns are likely driven by several phenomena—First, efforts to increase freshwater health have paid dividends and are likely plateauing. Second, mitigation efforts of more degraded systems are failing to continue to restore freshwater health and biodiversity.

Overall the authors did an extensive suite of additional analyses that helped strengthen the interpretation of the data. They assuaged my concerns given potential complications that are associated with the variable dataset in terms of coverage, resolution, etc. I appreciate the care and effort that was taken.

My main remaining substantive concern (should be easy to address) is with Figure 4 and its interpretation. My understanding of the estimates is that they are effect sizes on the trend, not effect sizes on biodiversity. So, I think a negative coefficient means that the trends are lower than average, but not necessarily that the trend is negative (biodiversity is decreasing). Presumably there is a positive intercept that would push these estimates up. These things seem conflated in several places in the manuscript. It is also quite possible that I am not interpreting this correctly, in which case a bit more information is needed with regards to the presentation of these findings.

- >Line 196-197. This topic sentence doesn't quite align with my understanding of the analysis—this result could come in part from lower positive trends in biodiversity (not just decreases in biodiversity). I would suggest that this sentence could be more precise.
- >Lines 253-255. Similar to line 196-197, it is not clear to me that the analysis actually shows biodiversity declines associated with dams/etc, the analysis shows that the biodiversity trend is lower than typical.
- >Other places where this conflation occurs include: 259-266; 269. The abstract wording is accurate in my read.
- >I am not sure I am thinking about this correctly, but one option might be to create prediction intervals that include the intercept. This might allow the paper to speak more directly to the pattern. It could help showcase whether, for example, dams are associated with biodiversity declines or not (rather than lower trends in biodiversity change).

- Figure 2 legend could use more clarity. I think it should state explicitly that the histogram is the site-level estimates. For the point estimate with error bars, I think some word (perhaps "aggregate"?) needs to be added to the legend.
- Lines 236-237. It wasn't clear exactly what this 60% statistic comes from.

Referee #2 (Remarks to the Author):

The authors have produced comprehensive responses to the reviewer comments (mine and others) and the revised manuscript presents a stronger case (e.g. with expanded sensitivity analyses) and does so more clearly. They are to be commended on the extent and care with which these have been done - the additional analyses in particular.

There are a couple of my original comments (Reviewer 2) and author responses that I'd like to pick up and then a few additional minor points to flag (line numbers relate to the clean version

without the changes highlighted):

Original comment "I was interested in your findings that (1) increases in diversity were smaller, or declines were evident, in catchments with cropland or urban land use".

I take your point about agricultural catchments and ongoing diffuse pollution. However, for urban areas, whilst diffuse pollution is also an issue, wouldn't you expect these catchments to be characterized by the majority of historical point source pollution and have been the major focus for installation of wastewater treatment plants/upgrading wastewater infrastructure? You highlight the improvement in wastewater treatment as a key driver of river recovery, so by that logic wouldn't you predict some of the largest recovery in and around urban centres?

Original comment "I think the first analysis of the paper – the overall mean trends through time – needs some more thought".

I think your response is well argued – and also the new phrasing in the Abstract: "However, these increases primarily occurred before the 2010s, and have since plateaued." is helpful here, being explicit about mean rates of change and nonlinearity.

Original comment "Similarly, it's good to see the inclusion of various sensitivity analysis. If I understand correctly, in the case of varying taxonomic resolution, different data sets are used for the different taxonomic levels..."

From your response, I don't think I expressed this clearly – it wasn't a criticism of the mixed resolution that you use, nor a suggestion to restrict the whole study to species-level data. Instead, it was a suggestion related to the sensitivity analysis: rather than comparing the results from different subsets of data where different taxonomic resolution was used (my understanding of your sensitivity analysis), a more effective way of carrying out the sensitivity analysis would surely be to use exactly the same data set throughout i.e. running the analysis with the species-level data (n = 762 sites, I think), then coarsening those same data to genus level and re-running the analysis, then repeating at family-level. This would separate any taxonomic resolution effect from potentially systematic differences in the subsets of data that were originally recorded at different resolutions e.g. different countries might use different resolution. From your extensive sensitivity analyses demonstrating the results to be robust to various sources of bias, I think it unlikely that this would reveal strong biases – so whilst I think this would be a better approach, it is perhaps not essential.

Additional, minor points:

Lines 172-3. "...we hypothesise that abundance, taxonomic diversity, and functional diversity have experienced a recovery...". A change in the wording here to something like "we hypothesise that abundance, taxonomic diversity, and functional diversity have all increased, consistent with a recovery..." would be more accurate and explicit about the changes you are expecting as part of biological recovery.

Lines 308-9. Should "reduced trends" be "declines"?

Lines 349-350. "However, gains in taxon richness started to decelerate around 2010, which may reflect recovery of some sites...". Something like "...may reflect full recovery of some sites..." would be clearer here.

Line 599 – just a point of clarification. Following the scaling of modalities between 0 and 1, was it the case that the sum of all modalities for a trait (e.g. the seven modalities for body size used in your illustration) = 1? Current wording is equivocal

Referee #4 (Remarks to the Author):

The authors have addressed my points sufficiently well, and I have no further comments.

Referee #5 (Remarks to the Author):

My brief comments on this manuscript are confined to the use of climate data in this analysis.

Lines 684-686: The authors extracted monthly climate data for this analysis from the TerraClimate dataset. TerraClimate development following a hybrid approach, which used high-resolution long-term average values from WorldClim 1.4 and 2 and a coarse-resolution time series from Climate Research Unit time series data version 4.0 (CRU Ts4.0). Climatologically-aided interpolation was used to superimpose monthly anomalies from the CRU time series on to the WorldClim 1971-2000 long-term averages to arrive at a high-resolution time series. The WorldClim interpolation formulation is fairly simple and can have difficulty capturing climatic variations in areas of complex terrain where rain shadows, coastal effects, or temperature inversions are present. However, there is no mention of the uncertainties in the climate data inputs, something that is often missing from reports on ecological analyses. It appears that most of the sampling sites are likely in areas with relatively good station coverage across the time period of interest and away from complex physiography. I also suspect that the sheer number of sampling points would override an occasionally poor climate estimate that might be made by TerraClimate, which is reassuring, but it would be useful to remind the reader that such uncertainties exist and should not be ignored when interpreting the results.

Lines 689-690: The authors report single slopes to describe trends in maximum temperature and precipitation for each site. These may not be completely representative, given that climate experiences cyclical variations over time, and rarely in the form of monotonic increases or decreases. More information is needed to clarify how the trends were calculated using the R package `brms80`. Were they calculated in the same manner as those of the biodiversity variables? Example plots of the maximum temperature and precipitation time series and the resultant calculated trends would help the reader visualize the data entering into these calculations.

Author Rebuttals to First Revision:

Nature manuscript 2022-03-04262A

Referees' comments:

Referee #1 (Remarks to the Author):

This paper presents the results of an extensive dataset on freshwater invertebrate biodiversity and its change over the last several decades. They discovered that biodiversity has tended to increase, but that these increases have plateaued recently. Further, the analyses revealed that broad-scale drivers such as dams are associated with less positive trends in biodiversity. These patterns are likely driven by several phenomena—First, efforts to increase freshwater health have paid dividends and are likely plateauing. Second, mitigation efforts of more degraded systems are failing to continue to restore freshwater health and biodiversity.

Overall the authors did an extensive suite of additional analyses that helped strengthen the interpretation of the data. They assuaged my concerns given potential complications that are associated with the variable dataset in terms of coverage, resolution, etc. I appreciate the care and effort that was taken.

Author response:

We thank the reviewer for their positive and constructive comments.

My main remaining substantive concern (should be easy to address) is with Figure 4 and its interpretation. My understanding of the estimates is that they are effect sizes on the trend, not effect sizes on biodiversity. So, I think a negative coefficient means that the trends are lower than average, but not necessarily that the trend is negative (biodiversity is decreasing). Presumably there is a positive intercept that would push these estimates up. These things seem conflated in several places in the manuscript. It is also quite possible that I am not interpreting this correctly, in which case a bit more information is needed with regards to the presentation of these findings.

Author response:

The reviewer is correct: the estimates shown in Figure 4 are effects of drivers on biodiversity trends. As the reviewer suggests, a negative coefficient means that at higher values of the driver, the trends are lower but not necessarily negative.

We took this approach of response variables always being the *temporal trend* of the biodiversity metrics because we used data from a variety of sources with variable sampling methods and effort, meaning raw values of biodiversity metrics are not directly comparable across the different studies included in our analysis. However, because methods did not vary within study, the study-level trends are more readily comparable. Moreover, our questions were primarily about biodiversity trends rather than the exact state of biodiversity itself.

We tried to use careful wording throughout the manuscript to reflect our analysis of drivers of trends (not raw abundance, richness, etc). This is stated in the caption of Fig. 4: “All response variables are site-level trends (i.e. change in biodiversity metric over time)” We have now extended the caption to better explain to the reader how to interpret the effect sizes:

Lines 229-233 (Fig. 4 caption): *A positive coefficient means that sites with higher values of the driver, tended to have higher trends, although not necessarily positive trends, compared*

to sites with lower values of the driver. For example, trends in taxon richness were higher at sites with higher maximum mean temperatures (tmax mean) but lower at sites with higher rates of temperature increase (tmax sl.);

In response to the reviewer's comment, we have identified places to alter our phrasing to clarify that the focus of our analysis was explaining variation in trends. In addition to the lines mentioned by the reviewer below (previous Lines 196-197, 253-255, 259-266, & 269), to clarify these results and their interpretation, we have rephrased our wording in the following relevant places throughout the manuscript and use only language referring to “positive” and “negative effects” of drivers on biodiversity trends:

Lines 189-192: *“Mean precipitation had a positive effect on functional richness long-term trends but a negative effect on abundance and functional redundancy long-term trends, indicating the addition of functionally unique taxa at wet sites.”*

Lines 201-204: *“Contrastingly, high dam impacts had a positive effect on both taxonomic and functional evenness long-term trends, suggesting that dominant species declined in abundance in communities downstream of dams, while richness declines were more pronounced for rare species.”*

Lines 210-212: *“A greater percentage of upstream urban area had negative effects on taxon richness long-term trends (Fig. 4) but positive effects on non-native richness long-term trends (Extended Data Fig. 5a) suggesting losses of rare and sensitive native species.”*

>Line 196-197. This topic sentence doesn't quite align with my understanding of the analysis—this result could come in part from lower positive trends in biodiversity (not just decreases in biodiversity). I would suggest that this sentence could be more precise.

Author response:

We agree with the reviewer and have changed the sentence as follows:

Lines 196-197: *“Biodiversity trends were generally lower at sites downstream of dams and in catchments with a high percentage of urban areas or cropland.”*

>Lines 253-255. Similar to line 196-197, it is not clear to me that the analysis actually shows biodiversity declines associated with dams/etc, the analysis shows that the biodiversity trend is lower than typical.

Author response:

We agree with the reviewer and have changed the sentence as follows:

Lines 255-258: *“Negative effects of poor water quality on biodiversity are supported by our findings that freshwater invertebrate communities downstream of dams, urban areas, and cropland were less likely to experience biodiversity recovery.”*

>Other places where this conflation occurs include: 259-266; 269. The abstract wording is accurate in my read.

Author response:

We agree with the reviewer and have changed the sentences as follows:

Lines 263-276: *“The positive effects of higher mean temperatures on long-term trends in invertebrate richness likely reflect the lower initial degradation in northern European countries. This may also reflect the overall cooler temperatures in European countries on*

average, whereas decreases are currently expected in freshwaters of warmer bioregions, such as tropical regions, which are not represented in our study⁴⁴. However, the negative effects on long-term trends of taxon richness, abundance, and functional richness in communities experiencing greater rates of warming are worrisome and are likely to worsen as temperatures continue to rise, and as climatic extremes including summer droughts and heatwaves become more common⁴⁵.

Considering that environmental legislation and policy have insufficiently addressed ongoing and emerging stressors⁸, the stalled recovery is unsurprising. Further management actions to revive the recovery should target sites at greater risk of biodiversity loss, such as those downstream of urban areas, cropland, and dams, while maintaining and strengthening protection of the least impacted systems, which are refuges of biodiversity.”

>I am not sure I am thinking about this correctly, but one option might be to create prediction intervals that include the intercept. This might allow the paper to speak more directly to the pattern. It could help showcase whether, for example, dams are associated with biodiversity declines or not (rather than lower trends in biodiversity change).

Author response:

We focus our interpretation on the model coefficients (i.e. effects of the drivers on biodiversity trends) rather than on the model predictions (i.e., biodiversity trend values) for a number of reasons:

- We used multiple regression to simultaneously model the effects of multiple drivers. We also standardized each driver covariate to units of standard deviation. This means that the regression coefficients can be interpreted as standardized effect sizes of the impact of each driver, indicating their relative importance.
- Equally, we appreciate the reviewer’s point: it would be interesting to understand whether the predicted trends are actually positive or negative under a specific set of conditions. However, various complexities arise because we are considering multiple drivers. For instance, whether trends are actually negative with higher dam impacts will also depend on the values of the other drivers.

However, there are some standard options to better visualize the model outputs to provide the information that the reviewer suggests. A common technique is to make predictions over the range of values for one covariate while holding all other covariate values at their median. The caveat with this approach is that this is just one scenario and this “median” scenario may not exist in the real world. However, this approach can be used to determine whether under this median scenario, biodiversity trends are negative, positive, or stable in response to environmental drivers. For example, using this approach and plotting predicted biodiversity trends (with 95% credible intervals shown as grey polygons and trends=0 depicted with red dashed lines) we find that at high dam impacts, taxon richness trends are negative:

This contrasts with, for example, crop cover, which has a negative effect on taxon richness trends, but even at sites with high mean % upstream crop cover, taxon richness trends are still predicted to be positive/stable (just less positive than at areas with low mean % upstream crop cover):

The direction of these predicted trends corroborates our standardized effect sizes in Fig. 4. We now include these plots for our eight main biodiversity metrics and seven main environmental drivers in our Online Figures (<https://github.com/Ewelti/EuroAquaticMacroInverts>, plots/Online Figures.docx: Figs. 28-34). Script for this analysis is at <https://github.com/Ewelti/EuroAquaticMacroInverts>, R/HPC_Meta_analysis_drivers.R. We prefer not to place these plots in the main text because we have insufficient space to properly explain them and ensure their correct interpretation.

•Figure 2 legend could use more clarity. I think it should state explicitly that the histogram is the site-level estimates. For the point estimate with error bars, I think some word (perhaps “aggregate”?) needs to be added to the legend.

Author response:

We agree with the reviewer regarding adding an explicit statement about the histograms. We have incorporated the 'aggregate' nature of these estimates by adding the term 'meta-analysis' to describe estimates. We have changed the figure legend as follows:

Lines 117-123: “**Fig. 2: Averages and distributions of trends in taxonomic and functional diversity metrics.** Overall meta-analysis estimates and distributions of site-level trends for taxonomic metrics of a, taxon richness, b, abundance, c, Shannon’s evenness, d, turnover, and functional metrics of e, richness, f, redundancy, g, evenness and h, turnover across all 1,816 sites. Histograms depict all site-level estimates. Black error bars and text on each panel show the mean estimates (percent change per year). Error bars indicate 80%, 90%, and 95% credible intervals.”

•Lines 236-237. It wasn’t clear exactly what this 60% statistic comes from.

Author response:

We agree with the reviewer and have found the 60% statistic to have been an error in calculation. We have changed the sentence as follows:

Lines 240-243: “*The biodiversity gains observed across 70% (1,269/1,816) of time series are concurrent with the widespread implementation of mitigation measures⁸, particularly improvements in wastewater treatment motivated by the EU Urban Waste Water Directive from 1991.*”

Referee #2 (Remarks to the Author):

The authors have produced comprehensive responses to the reviewer comments (mine and others) and the revised manuscript presents a stronger case (e.g. with expanded sensitivity analyses) and does so more clearly. They are to be commended on the extent and care with which these have been done - the additional analyses in particular.

Author response: We thank the reviewer for their comments which have improved the manuscript.

There are a couple of my original comments (Reviewer 2) and author responses that I’d like to pick up and then a few additional minor points to flag (line numbers relate to the clean version without the changes highlighted):

Original comment “I was interested in your findings that (1) increases in diversity were smaller, or declines were evident, in catchments with cropland or urban land use”.

I take your point about agricultural catchments and ongoing diffuse pollution. However, for urban areas, whilst diffuse pollution is also an issue, wouldn’t you expect these catchments to be characterized by the majority of historical point source pollution and have been the major focus for installation of wastewater treatment plants/upgrading wastewater infrastructure? You highlight the improvement in wastewater treatment as a key driver of river recovery, so by that logic wouldn’t you predict some of the largest recovery in and around urban centres?

Author response:

We agree with the reviewer that streams in urban areas have benefitted from improved wastewater treatment the most. However, at the same time, urban areas have expanded, further worsening stream hydromorphological conditions. Today, many streams in urban

areas are canalized and diked to prevent buildings from flooding. While water quality may have indeed been improved in urban areas, the poor hydromorphological conditions as well as other factors like increasing light pollution or impervious areas appear to remain a hindrance to biodiversity recovery.

We have rephrased the following sentence in our discussion to clarify this issue:

Lines 261-263: *“Most European rivers bear a substantial legacy of human impacts on their hydromorphology^{8,37}, with urban areas being most affected, despite considerable river restoration in recent decades⁴³.”*

Original comment “I think the first analysis of the paper – the overall mean trends through time – needs some more thought”.

I think your response is well argued – and also the new phrasing in the Abstract: “However, these increases primarily occurred before the 2010s, and have since plateaued.” is helpful here, being explicit about mean rates of change and nonlinearity.

Author response: We thank the reviewer for their appreciation of our revision.

Original comment “Similarly, it’s good to see the inclusion of various sensitivity analysis. If I understand correctly, in the case of varying taxonomic resolution, different data sets are used for the different taxonomic levels...”

From your response, I don’t think I expressed this clearly – it wasn’t a criticism of the mixed resolution that you use, nor a suggestion to restrict the whole study to species-level data. Instead, it was a suggestion related to the sensitivity analysis: rather than comparing the results from different subsets of data where different taxonomic resolution was used (my understanding of your sensitivity analysis), a more effective way of carrying out the sensitivity analysis would surely be to use exactly the same data set throughout i.e. running the analysis with the species-level data (n = 762 sites, I think), then coarsening those same data to genus level and re-running the analysis, then repeating at family-level. This would separate any taxonomic resolution effect from potentially systematic differences in the subsets of data that were originally recorded at different resolutions e.g. different countries might use different resolution. From your extensive sensitivity analyses demonstrating the results to be robust to various sources of bias, I think it unlikely that this would reveal strong biases – so whilst I think this would be a better approach, it is perhaps not essential.

Author response:

Thank you for clarifying your original suggestion. Our sensitivity analysis to test the effect of taxonomic resolution by a subset analysis suggested no cause for concern; hence, our findings appear robust. In particular, analysis of the dataset subset to those resolved to species (the largest group of studies), produced the same results as our presented analysis of the full dataset. Hence, our results would be the same if we only cherry-picked the best studies.

Additional, minor points:

Lines 172-3. “...we hypothesise that abundance, taxonomic diversity, and functional diversity have experienced a recovery...”. A change in the wording here to something like “we hypothesise that abundance, taxonomic diversity, and functional diversity have all increased, consistent with a recovery...” would be more accurate and explicit about the changes you are expecting as part of biological recovery.

Author response:

We have changed the sentence as follows:

Lines 62-64: “...we hypothesise that abundance, taxonomic diversity, and functional diversity have increased, consistent with a recovery...”

Lines 308-9. Should “reduced trends” be “declines”?

Author response:

Our response variables in the analysis of Fig. 4 are trends, not raw biodiversity metrics. In response to this comment and those of Reviewer 1, we have greatly reworded all text relating to the interpretation of the Fig. 4 analysis. For further clarification of the interpretation of Fig. 4, see the comments and our responses to Reviewer 1.

Lines 349-350. “However, gains in taxon richness started to decelerate around 2010, which may reflect recovery of some sites...”. Something like “...may reflect full recovery of some sites...” would be clearer here.

Author response:

We agree with the reviewer and have changed the sentence as follows:

Lines 243-246: “*However, gains in taxon richness started to decelerate around 2010, which may indicate that progress toward recovery has come to a halt at many sites, while remaining sites may reflect either predominant recovery or ongoing degradation towards the end of the study period.*”

Line 599 – just a point of clarification. Following the scaling of modalities between 0 and 1, was it the case that the sum of all modalities for a trait (e.g. the seven modalities for body size used in your illustration) = 1? Current wording is equivocal

Author response: The reviewer is correct that the sum of the modalities within a trait for a given taxon equals to one after scaling. We have revised the following sentence for clarity:

Lines 494-496: “*Within each trait, we scaled the affinities to different component modalities between 0 and 1, (summing to 1 across modalities for each taxon) so that each taxon was assigned an affinity score for each modality⁶⁴, to recognise potential trait plasticity.*”

Referee #4 (Remarks to the Author):

The authors have addressed my points sufficiently well, and I have no further comments.

Author response: We thank the reviewer for their previous comments, which have aided us in improving this work.

Referee #5 (Remarks to the Author):

My brief comments on this manuscript are confined to the use of climate data in this analysis.

Lines 684-686: The authors extracted monthly climate data for this analysis from the TerraClimate dataset. TerraClimate development following a hybrid approach, which used high-resolution long-term average values from WorlClim 1.4 and 2 and a coarse-resolution time series from Climate Research Unit time series data version 4.0 (CRU Ts4.0).

Climatologically-aided interpolation was used to superimpose monthly anomalies from the CRU time series on to the WorldClim 1971-2000 long-term averages to arrive at a high-resolution time series. The WorldClim interpolation formulation is fairly simple and can have difficulty capturing climatic variations in areas of complex terrain where rain shadows, coastal effects, or temperature inversions are present. However, there is no mention of the uncertainties in the climate data inputs, something that is often missing from reports on ecological analyses. It appears that most of the sampling sites are likely in areas with relatively good station coverage across the time period of interest and away from complex physiography. I also suspect that the sheer number of sampling points would override an occasionally poor climate estimate that might be made by TerraClimate, which is reassuring, but it would be useful to remind the reader that such uncertainties exist and should not be ignored when interpreting the results.

Author response:

We thank the reviewer for pointing us towards some uncertainties related to the climate data used in our analyses. We have now picked up this issue and added the following sentence to our method section:

Lines 587-590: *“The TerraClimate dataset is associated with uncertainties in areas of complex terrain, but our large number of sampling sites, relatively good station coverage, and the low physiographical complexity of most site locations should have minimised error in our analyses.”*

Lines 689-690: The authors report single slopes to describe trends in maximum temperature and precipitation for each site. These may not be completely representative, given that climate experiences cyclical variations over time, and rarely in the form of monotonic increases or decreases. More information is needed to clarify how the trends were calculated using the R package `brms80`. Were they calculated in the same manner as those of the biodiversity variables? Example plots of the maximum temperature and precipitation time series and the resultant calculated trends would help the reader visualize the data entering into these calculations.

Author response:

We agree with the reviewer that changes in climate are not expected to be linear at all sites and climatic cycles can be important drivers of biotic change in some systems. However, in our large meta-analysis, with multiple drivers simultaneously affecting the focal ecosystems, we are unlikely to capture these complexities. We are interested in the long-term trends of biodiversity metrics (e.g. taxon richness) and how these trends varied among locations. Therefore, we needed to define climatic metrics in composite values for each site and decided to characterize both the spatial and temporal dimensions (spatial - as a simple site-level mean and temporal - as a site-level linear slope).

Visualizations of climatic variation across sites are provided in the Online Figures (<https://github.com/Ewelti/EuroAquaticMacroInverts>, plots/Online Figures.docx: Fig. 12).

The calculation of climatic slopes is identical to those of biodiversity metrics. To clarify this point, we have added the following lines to our method section:

Lines 586-587: *“These models were similar to those used to calculate site-level biodiversity metric trends, in which a trend was estimated as the coefficient of a continuous year effect.”*

Reviewer Reports on the Second Revision:

Referees' comments:

Referee #1 (Remarks to the Author):

I appreciate the responses to my comments and to the other reviewers.

I still see that there are a few places where it appears that the paper appears to incorrectly interpret the data. I have voiced this before, and I will voice it again—it is very important the authors actually make sure that the data support the conclusions.

Line 267-268. I don't think this statement is correct. A change from positive change in functional richness to no change does not mean ecosystem function is decreasing, it means that ecosystem function has plateaued.

Line 268-270. Again, the paper conflates no change with a decrease. Functional redundancy is not changing, which does NOT mean lost resilience.

Line 292, 293. I think it should be "y-1".

Referee #2 (Remarks to the Author):

The authors have done a good job of addressing my comments (and those of the other reviewers) from the previous version of the manuscript.

I have just one further, minor point:

lines 265-268 "A switch from primarily positive trends in functional richness in the late 1990s and early 2000s to near-zero trends starting around 2012 (Fig. 3c) may suggest a reduction in ecosystem functioning."

Wouldn't a near-zero trend estimate suggest no change in ecosystem functioning, rather than a reduction in function? If so, this wording just needs updating to clarify this point.

Referee #5 (Remarks to the Author):

Thank you for your responses to my comments – I feel they are appropriate. One of my concerns was that linearly calculated climate trends could be misleading, since they are rarely linear (this is often an issue in climatological studies). However, I appreciate that climate trends were just one of several possible drivers of freshwater biodiversity considered in this study, and that the biodiversity data would not likely be sufficient to capture more nuanced climatic variations. Thus, the choice to keep the driving metrics simple.

Author Rebuttals to Second Revision:

Nature manuscript 2022-03-04262B

Referees' comments:

Referee #1 (Remarks to the Author):

I appreciate the responses to my comments and to the other reviewers.

Author response: We thank the reviewer for their positive and constructive comments.

I still see that there are a few places where it appears that the paper appears to incorrectly interpret the data. I have voiced this before, and I will voice it again—it is very important the authors actually make sure that the data support the conclusions.

Line 267-268. I don't think this statement is correct. A change from positive change in functional richness to no change does not mean ecosystem function is decreasing, it means that ecosystem function has plateaued.

Author response: The reviewer is right and we have changed this and the next sentence accordingly. Please see our changes in the response to your next comment.

Line 268-270. Again, the paper conflates no change with a decrease. Functional redundancy is not changing, which does NOT mean lost resilience.

We have rephrased the lines 277-282:

“A switch from primarily positive trends in functional richness in the late 1990s and early 2000s to near-zero trends starting around 2012 (Fig. 3c) may suggest no further improvements in ecosystem functioning. The concurrent limited change in functional redundancy (Fig. 3d) indicates that the increase in functional richness provided new traits to the communities rather than adding traits that were already present.”

Line 292, 293. I think it should be “y-1”.

Author response: Yes, many thanks. Changed as suggested.

Referee #2 (Remarks to the Author):

The authors have done a good job of addressing my comments (and those of the other reviewers) from the previous version of the manuscript.

Author response: We thank the reviewer for their positive and constructive comments.

I have just one further, minor point:

lines 265-268 "A switch from primarily positive trends in functional richness in the late 1990s and early 2000s to near-zero trends starting around 2012 (Fig. 3c) may suggest a reduction in ecosystem functioning."

Wouldn't a near-zero trend estimate suggest no change in ecosystem functioning, rather than a reduction in function? If so, this wording just needs updating to clarify this point.

Author response: The reviewer is correct and we have rephrased the sentence as follows (lines 277-280):

“A switch from primarily positive trends in functional richness in the late 1990s and early 2000s to near-zero trends starting around 2012 (Fig. 3c) may suggest no further improvements in ecosystem functioning.”

Referee #5 (Remarks to the Author):

Thank you for your responses to my comments – I feel they are appropriate. One of my concerns was that linearly calculated climate trends could be misleading, since they are rarely linear (this is often an issue in climatological studies). However, I appreciate that climate trends were just one of several possible drivers of freshwater biodiversity considered in this study, and that the biodiversity data would not likely be sufficient to capture more nuanced climatic variations. Thus, the choice to keep the driving metrics simple.

Author response: We thank the reviewer for their positive and constructive comment.